# Multiplicative priming of the correct response can explain the interaction between Simon and flanker congruency

**Alodie Rey-Mermet**[1,2]*, **Miriam Gade**[1,3], **Marco Steinhauser**[1]

**1** Department of Psychology, General Psychology, Catholic University of Eichstätt-Ingolstadt, Eichstätt, Germany, **2** Faculty of Psychology, Swiss Distance University Institute, Brig, Switzerland, **3** Department of Sciences, Medical School Berlin, Berlin, Germany

* alodie.rey-mermet@fernuni.ch

## Abstract

In the Simon task, participants perform a decision on non-spatial features (e.g., stimulus color) by responding with a left or right key-press to a stimulus presented on the left or right side of the screen. In the flanker task, they classify the central character while ignoring the flanking characters. In each task, there is a conflict between the response-relevant features and the response-irrelevant features (i.e., the location on the screen for the Simon task, and the flankers for the flanker task). Thus, in both tasks, resolving conflict requires to inhibit irrelevant features and to focus on relevant features. When both tasks were combined within the same trial (e.g., when the row of characters was presented on the left or right side of the screen), most previous research has shown an interaction. In the present study, we investigated whether this interaction is affected by a multiplicative priming of the correct response occurring when both Simon and flanker irrelevant features co-activate the correct response (Exp. 1), a spatial overlap between Simon and flanker features (Exp. 2), and the learning of stimulus-response pairings (Exp. 3). The results only show an impact of multiplicative priming.

## Introduction

In a Simon task, participants are asked to perform a decision on non-spatial features for stimuli presented on the left and right side of the screen [e.g., 1–3]. For example, they can be instructed to perform a color decision by pressing a left key for one response (e.g., the response "red") and a right key for the other response (e.g., the response "blue"). In a flanker task, participants are asked to classify the central character of a row (e.g., the color of the middle letter from a row of X's) while ignoring the flanking characters [e.g., 4]. When both tasks are combined within the same trial (e.g., by presenting a colored row of X's on the left or right side of the screen), previous research has revealed an interaction [5–9 but see 8, 10]. The purpose of the present study was to shed more light on the source of this interaction. More specifically, we asked whether this interaction is affected by a multiplicative priming effect, a spatial overlap between Simon and flanker features, or the learning of stimulus-response pairings.

**Data Availability Statement:** The experiment scripts, data, scripts analyses and results presented in this work can be found at https://osf.io/dxp58.

**Funding:** This work was supported by a grant from the Swiss National Science Foundation (www.snf.ch) to ARM (Grant P300P1_164598). The funder had no role in study design, data collection and analysis, decision to publish, or preparation of the manuscript.

**Competing interests:** The authors have declared that no competing interests exist.

## Simon task, flanker task, and their interaction

In both Simon and flanker tasks, trials are labelled incongruent when they contain features pointing to two different response alternatives. For example, a trial is incongruent in the Simon task when the stimulus location on the screen activates a response that is different from the correct response (e.g., a red letter presented on the right side of the screen but requiring a left key-press). In the flanker task, a trial is incongruent when the response activated by the color of the central letter is different from the response activated by the color of the flanking letters (e.g., a red central letter flanked by blue letters). In contrast, trials are congruent when the response-relevant and the response-irrelevant features activate the same response. For example, a trial is congruent in the Simon task when the stimulus location on the screen activates the correct response (e.g., a red letter presented on the left side on the screen and requiring a left-key press). In the flanker task, a trial is congruent when the response activated by the color of the central letter is the same as the response activated by the color of the flanking letters (e.g., a red central letter flanked by red letters). In both tasks, the results showed slower and more error-prone performance on incongruent trials than on congruent trials. This difference in performance is called "congruency effect".

Theoretically, resolving the conflict in the flanker or Simon task requires to focus on relevant features and/or to inhibit irrelevant features, suggesting at least a similarity or a relation between the processes assumed to be involved in each task. Findings from correlational studies showed, however, small to moderate correlations between both congruency effects (-.01 to .21), suggesting different processes across the tasks [11–16]. In contrast to the correlational approach, the experimental approach suggests a relation between both tasks by showing an interacting pattern. Table 1 shows an overview about the previous studies combining both tasks experimentally. Typically, both tasks were combined by presenting the row of characters for the flanker task on the left or right side of the screen. Most results showed an under-additive interaction, indicating that one congruency effect (e.g., the flanker congruency effect) was smaller in trials that were incongruent with respect to the other congruency variable (in this case, Simon incongruent trials).

## Accounts used to explain the interaction between Simon and flanker congruency

Previous research has used different theoretical accounts to explain the interaction between congruency variables. A first attempt is based on the conflict monitoring framework [19], which assumes that a conflict monitoring system estimates the current levels of conflict and adjusts control accordingly. While conflict monitoring has originally been used to account for sequential modulations of control across trials [see 20, 21, for reviews], it has also been applied to explain the within-trial interactions between the different congruency variables [7, 22, 23]. In particular, to account for the interaction between Simon and flanker congruency, one could assume that if the first irrelevant feature being processed (e.g., the irrelevant flanker) is incongruent with the relevant one and thus associated with a high degree of conflict, this leads to a shift of control signal. As a consequence, the response-relevant feature is activated while all irrelevant features–including the irrelevant stimulus location–are inhibited. Thus, the strength of control is also adjusted for the second irrelevant feature to be ignored. This reduces the impact of the irrelevant features if the second irrelevant feature is also incongruent, thus resulting in a smaller congruency effect. According to this account, control processes generalize across all irrelevant features, explaining the under-additive interaction observed in most previous studies.

**Table 1. Previous studies combining Simon and flanker congruency within the same trial.**

| Study | Sample size | Under-additive interaction? | Decision | Number of stimulus values per congruency variable | Simon orientation | Flanker orientation |
|---|---|---|---|---|---|---|
| Akçay and Hazeltine [10]– Exp. 1 | 18 | no | color | 4 locations, 4 colors | horizontal | vertical |
| Akçay and Hazeltine [10]– Exp. 2 | 24 | no | color | 5 locations, 5 colors[a] | horizontal | vertical |
| Frühholz and colleagues [5]–EEG session | 20 | yes | color | 2 locations, 2 colors | horizontal | horizontal and vertical (i.e., cross-like around the target) |
| Frühholz and colleagues [5]–fMRI session | 24 | yes | color | 2 locations, 2 colors | horizontal | horizontal and vertical (i.e., cross-like around the target) |
| Hommel [6]–Exp. 2 (horizontal) | 16 | yes | letter | 2 locations, 2 letters | horizontal | horizontal |
| Hommel [6]–Exp. 2 (vertical) | 16 | | | | vertical | vertical |
| Rey-Mermet & Gade [7]– Exp. 3 | 24 | yes | color | 4 locations, 4 colors | quadrant[b] | horizontal |
| Stoffels and Molen [8]– Exp. 1[c] | 20 | no | letter | 3 locations, 3 letters[d] | horizontal | horizontal |
| Stoffels and Molen [8]– Exp. 2[c] | 16 | yes | arrow | 3 locations, 3 arrows[d] | horizontal | horizontal |
| Treccani and colleagues [17]–Exp. 1[e] | 26 | yes | colors | 2 locations2 colors | horizontal | horizontal |
| Wendt and colleagues [9]– Exp. 2a | 15 | yes | letter | 2 locations, 2 letters | horizontal | horizontal and vertical (i.e., cross-like around the target) |

EEG = electro-encephalogram. fMRI = functional magnetic resonance imaging. The studies listed here tested young healthy adults (see [18] for a study including a neglect patient).

[a]All locations and colors were response relevant.

[b]Stimuli were presented in one of the four quadrants limited by a cross hair.

[c]Simon and flanker congruency variables were so combined that the stimulus for the flanker congruency (either congruent, incongruent or neutral) was accompanied by auditory noise presented to the ear ipsilateral to the correct response (Simon congruent), to the ear contralateral to the correct response (Simon incongruent), or to both ears (neutral).

[d]One location and one letter/arrow was included to create neutral trials (i.e., not all locations and letters/arrows were response relevant).

[e]A unilateral flanker stimulus was combined with an accessory Simon stimulus. That is, a central colored square was presented in the middle of the screen, which was flanked by another colored square either to the left or to the right. Thus, the flanker square was a distractor for the flanker congruency variable because it was presented in a congruent or incongruent color. At the same time, it was also a distractor for the Simon congruency variable because it conveyed an irrelevant location, which was assumed to be congruent or incongruent with the response location.

A second account is based on the spontaneous decay of response-irrelevant features [6]. According to this "temporal overlap" account, whereas the representations of the response-relevant features are actively maintained, the representations of response-irrelevant features are assumed to decay rapidly. Thus, when the trial is Simon and flanker incongruent, incongruent flankers slow down trial processing, which thus provides a longer opportunity for the irrelevant location code to decay. This decreases the impact of the irrelevant location code on response selection. In contrast, when the trial is Simon incongruent but flanker congruent, the time to decay for the representation of the irrelevant location would be shorter, and its impact would be greater. Together, this results in a smaller Simon congruency effect for flanker incongruent trials than for flanker congruent trials, thus explaining the interaction between Simon and flanker congruency.

A third account used to explain the interaction between Simon and flanker congruency is the multiplicative priming of the correct response. According to this account, when trials are

congruent for both congruency variables, the correct response is not only activated by the target feature, but also primed by the irrelevant stimulus features (i.e., the color of the flanking character for the flanker congruency and the irrelevant location for the Simon congruency). Multiplicative effects of priming can result if each of the irrelevant features increases response activation proportional to its activation. Once the response is primed by one of the irrelevant features, the priming effect of the other feature would have a stronger effect on the activation level of this response. First empirical support for such a priming of irrelevant features was provided by Treccani and colleagues [17]. In their study, both Simon and flanker congruency variables were so combined that a colored square was presented in the middle of the screen, which was flanked by one colored square either to the left or to the right (see Fig 1). Thus, the flanker square conveyed the irrelevant flanker color that causes the flanker congruency effect. At the same time, it also conveyed the irrelevant location that causes the Simon congruency effect. The reason is that the square location was either congruent or incongruent with the location of the correct response. Thus, in congruent-congruent trials, the irrelevant square triggers the correct response twice (i.e., by means of its color and its position; see the top-left panel of Fig 1), thus facilitating performance. Moreover, in this particular design, the irrelevant square also triggers the incorrect response in incongruent-incongruent trials twice (again by means of its color and its position; see the bottom-right panel of Fig 1). This allows a faster rejection of this response. Together, this accounts for the interaction between Simon and flanker congruency by explaining the faster and more correct response in congruent-congruent trials and incongruent-incongruent trials compared to the trials mixing incongruent and congruent features.

The first two accounts–that is, the within-trial conflict monitoring and the spontaneous decay of the representations of response-irrelevant features–have been questioned by empirical findings. First, it has been shown that the interaction between Simon and flanker congruency may occur throughout the RT distribution [7], questioning the temporal overlap account, in particular the assumption that the slow processing of incongruent flankers provides a longer opportunity for the irrelevant location code to decay. However, the design used by Rey-Mermet and Gade [7] differed from the design used, for example, by Hommel [6] (see Table 1), suggesting that the decay of the response codes could explain the observed interaction in some experiments but not in others. Second, the central assumption put forward in both accounts–that is, resolving the first conflict affects the processing of the second conflict–has been put into question by a study using event-related potentials (ERPs) [5, see also 23, 24]. In that study, ERPs were used to track the neural time course of conflict processing. The results showed an interaction in RTs, but a sequential conflict processing in the ERPs. That is, the processing of the flanker conflict was associated with an early component (i.e., N2), whereas the processing of the Simon conflict was associated with a later component (i.e., P3b). However, no interaction was observed in the ERPs, in particular on the later component. If the first conflict affected the processing of the second conflict, this should have resulted in an interaction in the later component because this later component is associated to the processing of the second conflict. In contrast, the results suggest that there is little influence of the first conflict on the processing of the second conflict.

The multiplicative priming effect of the correct response has been so far tested only by Treccani and colleagues [17, 18]. In comparison to previous studies (see Table 1), they used a very particular design in which the same object–that is, the irrelevant square–conveyed the irrelevant features for both congruency variables. This may have changed the weight for processing the irrelevant features so that the irrelevant features could trigger the associated response. Thus, it is possible that in all other studies reported in Table 1 in which the Simon and flanker congruency variables were combined in a more standard way–that is, with different objects conveying the irrelevant features for both congruency variables–the priming effect cannot

## Flanker congruent
## Simon congruent

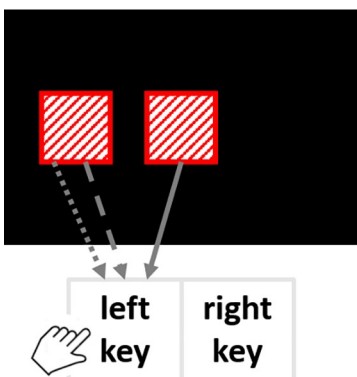

## Flanker congruent
## Simon incongruent

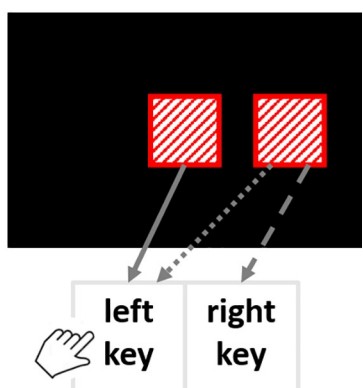

## Flanker incongruent
## Simon congruent

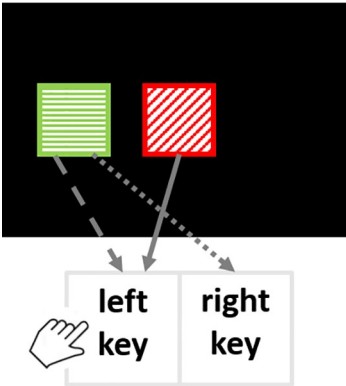

## Flanker incongruent
## Simon incongruent

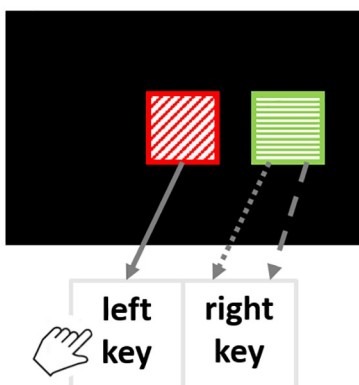

**Fig 1. Schematic depiction of the design used by Treccani and colleagues [17] in which both Simon and flanker irrelevant features are conveyed by the same object for all trial types (flanker congruent–Simon congruent, flanker congruent–Simon incongruent, flanker incongruent–Simon congruent, flanker incongruent–Simon incongruent).** The target feature associated to the correct response is the color of the central square, which was presented either in red or green. These two colors are additionally presented in oblique and horizontal striations, respectively. The red color was mapped to the left key, whereas the green color was mapped to the right key. Squares presented on the left and right side of the screen were the flanker squares. Solid arrows represent the priming of the target feature (i.e., the color of the central square) to the correct response. Dotted arrows represent the priming of the flanker irrelevant feature (i.e., the color of the flanker square) to the response. Dashed arrows represent the priming of the Simon irrelevant feature (i.e., the left-right location of the screen) to the response. The figure was adapted from the Fig 3 of Treccani and colleagues [17].

account for the interaction between Simon and flanker congruency. This might be particularly the case because there is some debate in previous research whether in standard Simon and flanker tasks, the irrelevant stimulus feature can co-activate the correct response in congruent trials at all [2, 4, 25–29].

Together, the above-mentioned accounts can so far explain some but not all empirical results. Accordingly, more than one process could account for the under-additive interaction between flanker and Simon congruency.

## Why was the interaction between Simon and flanker congruency sometimes not observed?

Table 1 highlights that whereas most studies report an interaction between Simon and flanker congruency [5–9, 17], a few studies did not [8, 10]. A closer inspection of these studies suggest two possible candidates explaining the discrepancy in findings: (1) the feature overlap between both Simon and flanker congruency variables, and (2) the learning of stimulus-response pairings. We next discuss each of these candidates in detail.

**Feature overlap between Simon and flanker congruency.** First, most studies included some spatial overlap in the features of the Simon and flanker congruency variables [see 10, for an exception]. For example, in Treccani and colleagues [17], the irrelevant features for both congruency variables were conveyed by the same distractor object (i.e., a colored square presented to the left or right of the screen). In Hommel [6], the irrelevant location leading to the Simon congruency effect was the horizontal position of the stimulus display (flankers + target), and the flankers were also presented horizontally (i.e., left and right to the target). In Rey-Mermet and Gade [7], there was also a partial overlap. That is, the stimulus display (flankers + target) were presented in one of the four quadrants so that the irrelevant locations leading to the Simon congruency effect were the horizontal and vertical position of the stimulus display. Thus, there was a partial overlap with the flanker congruency variable as the flanking characters were presented horizontally.

In most studies including some spatial overlap, the interaction between Simon and flanker congruency was observed (see the first experiment in [8] for an exception), whereas no interaction was reported when there was no overlap [10]. This suggests that certain aspects of the task structure–such as an overlap in the Simon and flanker features–may favor an integration of representations of Simon and flanker features [30], thus creating a basis for the interaction between Simon and flanker congruency.

**Learning of stimulus-response pairings.** A second issue is the question whether the interaction is affected by episodic memory effects, in particular the learning of stimulus-response pairings [31]. According to this account, the interaction would be affected by the different numbers of stimulus-response pairings in the four trial types (i.e., flanker congruent–Simon congruent, flanker incongruent–Simon congruent, flanker congruent–Simon incongruent, flanker incongruent–Simon incongruent). That is, in case all four trial types are presented equally often but there are different numbers of stimulus-response pairings pro trial type, some stimulus-pairings are presented more frequently than others. In this case, the stimulus-pairings that are presented more frequently are more practiced, explaining better performance for these stimulus-pairings and thus the trial types to which these pairings belong. It is important to note that this difference in learning stimulus-pairings cannot affect the interaction when the stimulus set is small (i.e., when there are two stimulus values per congruency variable, such as the left and right location for the Simon congruency variable, and the colors blue and green for the flanker congruency variable). In this case, the number of stimulus-response pairings is similar across all trial types (see Fig 2, top part), and thus the learning of these stimulus-response pairings should be similar in all trial types.

In contrast, when the stimulus set consists of four stimulus values (i.e., four locations for the Simon congruency variable and four colors for the flanker congruency variable), the number of stimulus-response pairings differed across trial types (see Fig 2, bottom part). There are four stimulus-response pairings when both congruency variables are congruent (i.e., for each of the four congruent pairings of stimulus and response locations, there is one congruent paring of target and flanker colors), twelve stimulus-response pairings when one congruency

| Stimulus set size | Flanker congruent Simon congruent | Flanker congruent Simon incongruent | Flanker incongruent Simon congruent | Flanker incongruent Simon incongruent |
|---|---|---|---|---|
| 2 | 2  | 2  | 2  | 2  |
| 4 | 4 (= 4 congruent target/flankers X 1 congruent location) | 12 (= 4 congruent target/flankers x 3 incongruent locations) | 12 (= 3 incongruent target/flankers x 4 congruent locations) | 36 (= 4 targets x 3 flankers x 3 locations) |

**Fig 2. Number of stimulus-response pairings for each trial type (flanker congruent–Simon congruent, flanker congruent–Simon incongruent, flanker incongruent–Simon congruent, flanker incongruent–Simon incongruent) in set size 2 and 4.** For the set size 2, the target feature associated to the correct response is the color of the central square, which was presented either in red or green. These two colors are additionally presented in oblique and horizontal striations, respectively. The red color was mapped to the left key, whereas the green color was mapped to the right key.

variable is congruent and the other is incongruent (i.e., 4 congruent target/flanking colors x 3 incongruent locations, or 3 incongruent target/flanking colors x 4 congruent locations), and thirty-six stimulus-response pairings when both congruency variables are incongruent (i.e., 4 target colors x 3 flanking colors x 3 locations). Thus, presenting equally often each trial type results in different frequencies for each stimulus-response pairing. This may result in a better learning for the stimulus-response pairings which are presented more frequently and thus practiced more often. It is so far unknown how this learning might shape the interaction between Simon and flanker congruency. For example, it is possible that more practice does not lead, at first, to better performance (e.g., going from the few presentations of each of the stimulus-response pairings for incongruent-incongruent trials to the relatively more frequent presentations of the stimulus-response pairings for trials mixing incongruent and congruent features). However, a significant improvement in performance may be found with much more practice, such as when the stimulus-response pairings are presented very frequently in congruent-congruent trials. Therefore, if the learning of stimulus-response pairings has an impact on the interaction between Simon and flanker congruency, it should be best observed in congruent-congruent trials. This would explain why responses were the fastest and the most correct for the congruent-congruent trial type. Critically, explaining performance on this trial type may be sufficient to account for the interaction between Simon and flanker congruency observed in Rey-Mermet and Gade [7]. In that study, all possible stimulus-response pairings were presented in each trial type (please note that this information was not provided in Akçay and Hazeltine [10], the unique other study including a large stimulus set but showing no interaction between both congruency variables; see Table 1). In particular, Rey-Mermet and Gade's [7] results showed that the interaction was driven by fast RTs and high rates of correct responses in the congruent-congruent trials. There was no decrease in RTs (i.e., faster RTs) or increase in correct responses (i.e., higher rates of correct response) in the incongruent-incongruent trials.

### The present study

The purpose of the present study was to shed more lights on the variables influencing the under-additive interaction observed between Simon and flanker congruency. More precisely, Experiments 1a and 1b were designed to investigate the role of multiplicative priming in the interaction when using standard administration of the Simon and flanker congruency variables, that is, when different objects convey the irrelevant features for both Simon and flanker congruency variables. Experiment 2 was conducted to test for the possibility that an overlap in the spatial configuration of the Simon and flanker features is necessary to find an interaction between Simon and flanker congruency. Experiment 3 was intended to determine the impact of the learning of stimulus-response pairings on the under-additive interaction. In all experiments, we opted for a design with a large stimulus set as this design was investigated only in a few studies and resulted in mixed evidence (see Table 1). Moreover, the data were analyzed using null-hypothesis significance testing (NHST) with analyses of variance (ANOVAs) and Bayesian hypothesis testing with Bayesian ANOVA.

## Experiment 1

Experiments 1a and 1b were conducted to determine whether the under-additive interaction between Simon and flanker congruency is affected by the multiplicative priming of the correct response in a design in which–contrary to the design used by Treccani and colleagues [17, 18]–the irrelevant features for both Simon and flanker congruency variables are conveyed by different objects (see Fig 3). With such a design, in congruent-congruent trials, the correct response is activated not only by one of the irrelevant features (e.g., the position on the screen) but also by the other irrelevant feature (e.g., the flanker; see the top-left panel of Fig 3). This may result in multiplicative effects of priming of the correct response due to co-activation [17, 32]. Thus, this would accelerate trial processing for the congruent-congruent trials not only compared to when the response is primed by only one irrelevant feature (see all other panels of Fig 3), but its facilitative effect would be also stronger than the sum of the single priming effects. This over-additive priming would result in an under-additive RT pattern because the processing of congruent-congruent trials is selectively accelerated.

To assess the impact of this priming, we included neutral trials–that is, trials with only one response-relevant feature–in each congruency variable. For the flanker congruency, the neutral trials were the trials in which the flanking letters were printed in a response-irrelevant color (e.g., dark pink or green). For the Simon congruency, previous research has used two types of neutral trials [33]: (1) trials in which the stimulus was presented in the middle of the screen, and (2) trials in which the stimulus was also presented in the center but in the top or bottom part of the screen. Both types of Simon neutral trials were used in Experiments 1a and 1b, respectively. In Experiment 1b, we opted for two stimulus exemplars (i.e., the top and bottom part of the screen) for Simon neutral trials. Thus, all stimulus-response pairings for both Simon and flanker neutral trials were presented equally often.

With such a design, we had three trial types for each Simon and flanker congruency variable in each experiment: *incongruent* trials, which are assumed to trigger interference, *congruent* trials, which are assumed to trigger positive priming or facilitation, and *neutral* trials, which are assumed to trigger neither interference nor facilitation. This allows us to distinguish the interference effect from the facilitation effect [e.g., 34–36]. By computing the difference between incongruent and neutral trials, we can measure the interference processes induced by incongruent trials. By computing the difference between congruent and neutral trials, we can measure the priming induced by congruent trials.

In both experiments, we hypothesized that if the interaction between Simon and flanker congruency is affected by the co-activation of the correct response when the trial is both

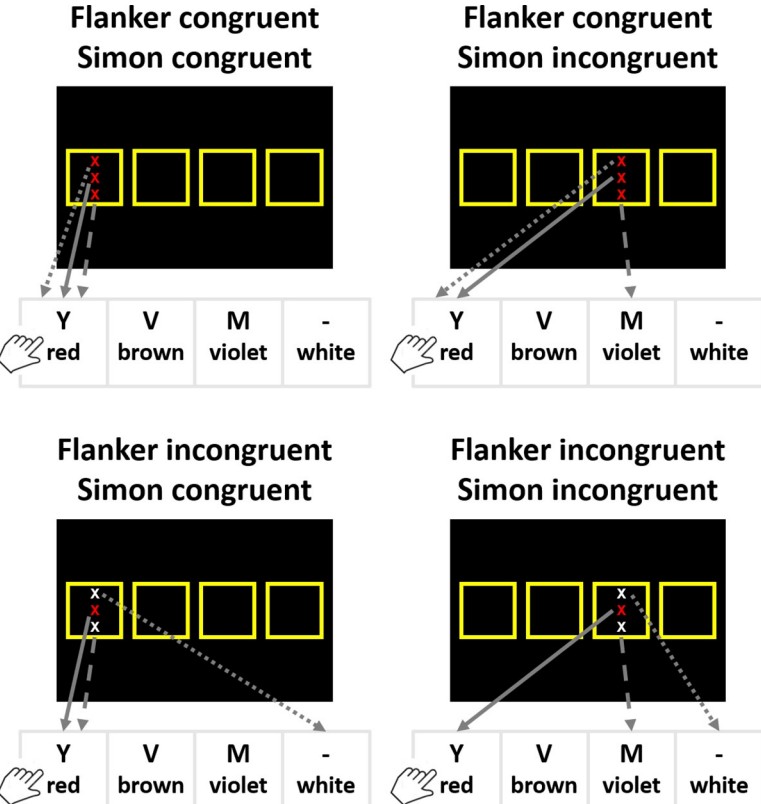

**Fig 3. Schematic depiction of the design used in Experiment 1 in which Simon and flanker irrelevant features were conveyed by different objects for all trial types (flanker congruent–Simon congruent, flanker congruent–Simon incongruent, flanker incongruent–Simon congruent, flanker incongruent–Simon incongruent).** The target feature associated to the correct response is the color of the central X, which was presented either in red, brown, violet, or white. X's presented above and below the central X were the flanker stimuli. Solid arrows represent the priming of the target feature (i.e., the color of the central X) to the correct response. Dotted arrows represent the priming of the flanker irrelevant feature (i.e., the color of the flanker X) to the response. Dashed arrows represent the priming of the Simon irrelevant feature (i.e., the location of the screen) to the response.

Simon and flanker congruent [17, 32], the interaction should be observed when the analyses focused on the facilitation effect. The reason is that the facilitation effect is computed as the difference between congruent and neutral trials, and congruent trials are assumed to induce some facilitative priming. In contrast, the interaction should not be found when the analyses focused on the interference effect because this difference involved incongruent and neutral trials (i.e., trials in which no facilitative priming is assumed to occur). However, these hypotheses are based on the assumption that the irrelevant stimulus feature for each Simon and flanker congruency co-activate the correct response in congruent trials. This assumption has been debated in previous research [2, 4, 25–29], raising the possibility that both the irrelevant location and flanking letters do not co-activate the correct response. In this case, the co-activation of the correct response would not occur when the trial is both Simon and flanker congruent. If so, no interaction should be observed when the analyses focused on the facilitation effect.

## Method

**Participants.** For all experiments, we aimed at a minimum sample size of 24 participants per congruency variable in which the interaction between Simon and flanker congruency is expected to occur. This belongs the largest sample sizes tested in previous research (see

Table 1). Twenty-nine participants took part in Experiment 1a (28 women, 26 right-handed, $\text{mean}_{\text{age}}$ = 21.5 years, $\text{SD}_{\text{age}}$ = 3.0). Thirty new participants took part in Experiment 1b (27 women, 29 right-handed, $\text{mean}_{\text{age}}$ = 21.8 years, $\text{SD}_{\text{age}}$ = 2.9).

All participants had normal or corrected to normal vision. They received payment (8 € per hour) or course credit for participating in the experiment. The study was approved by the ethics committee of the Catholic University of Eichstätt-Ingolstadt (approval number: 2016/18), and written informed consent was acquired from all participants.

**Material.** The present experiment consisted of three types of blocks: (1) a stimulus-response mapping block, which was included as a practice block to learn the stimulus-response mapping; (2) pure Simon and flanker blocks, which served to familiarize participants with each congruency separately and to control for the presence of congruency effects within each congruency variable; and (3) mixed blocks in which both congruency variables were combined. The stimuli for each block type are described separately.

In the *stimulus-response mapping block*, a stimulus consisted of a row of four asterisks colored in either red, brown, violet, or white (the same colors we used in [7]). All stimuli were presented centrally on a black background in 40-point Arial Bold font. They comprised a height of 0.38˚ visual angle and a width of 1.81˚ visual angle at a viewing distance of 60 cm (screen size: 38 x 30 cm, and screen resolution: 1280 x 1024 px). The stimulus color was determined randomly for each trial.

In the *pure Simon blocks*, a stimulus consisted of a row of four asterisks displayed in the colors red, brown, violet, and white. All stimuli were presented on a black background in 28-point bold Arial font. In Experiment 1a, each stimulus was displayed in one of five locations, which were presented horizontally in the center of the screen (see Fig 4A). At each location, a square comprising a side length of 3.25˚ visual angle was presented in yellow. The distance between the squares was 1.81˚ visual angle. In Experiment 1b, each stimulus was displayed in one of six locations: four of them deviated horizontally from the center of the screen (two left, two right), and two of them deviated vertically from the center of the screen (up, down; see Fig 4B). Similar to Experiment 1a, a square comprising a side length of 3.25˚ visual angle was presented in yellow at each location. For the four squares with a horizontal deviation, their location was the same as in Experiment 1a. For the two squares with a vertical deviation, the distance between them was 8.50˚ visual angle. The stimulus–displayed in the center of the square–was determined randomly for each trial. A trial was congruent when the location of the asterisks on the screen corresponds to the location of the response key. In contrast, it was incongruent when the location of the asterisks on the computer screen does not correspond to the location of the response key. Furthermore, in Experiment 1a, a trial was neutral when the stimulus was presented in the middle of screen. In Experiment 1b, a trial was neutral when the stimulus was presented at a location that deviated vertically from the center of the screen. Congruent, incongruent, and neutral trials were presented equally often.

In the *pure flanker* blocks, the stimulus consisted of the letter X presented three times, displayed either in red, brown, violet, white, dark pink or green. All stimuli were presented centrally (without a yellow square) on a black background in 28-point Arial Bold font. In Experiment 1a, the flanker X's were presented above and below the target so that all three letters comprised a height of 1.62˚ visual angle and a width of 0.57˚ visual angle. In Experiment 1b, to avoid increasing the impact of flanking letters when moving the eyes to the top or bottom on the screen in mixed blocks (see below), flankers were presented on the diagonal. The two diagonals (i.e., from the bottom left to the top right and from the top left to the bottom right) were presented equally often. The stimulus was determined randomly for each trial. A trial was congruent when the color of the central letter was the same as the color of the flanking letters. A trial was incongruent when the color of the central letter was different from the color

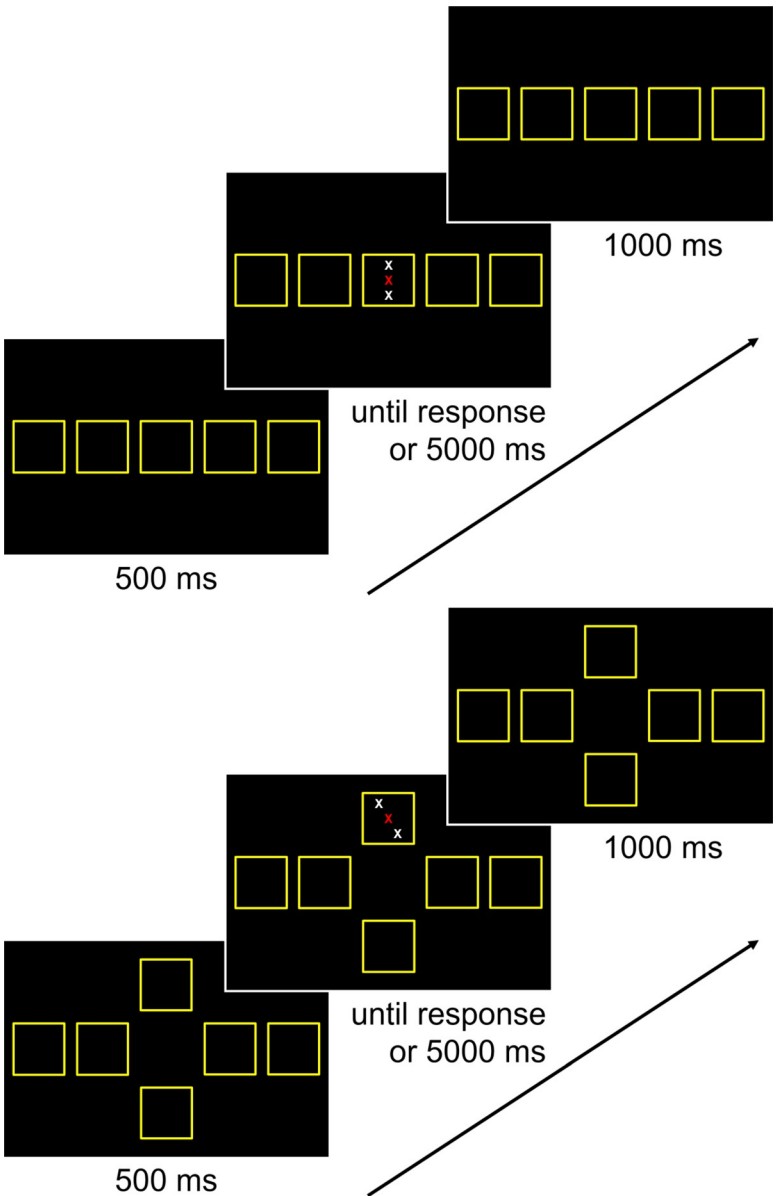

**Fig 4. Experiment 1: Example of one trial sequence.** Participants were asked to indicate the color of the central letter while ignoring the flanking letters and their location on the screen. They used the four response keys *y*, *v*, *m*, and -, which were mapped to the colors red, brown, violet, and white, respectively. Top part: Experiment 1a. Bottom part: Experiment 1b.

of the flanking letters. Furthermore, a trial was neutral when the flanking letters were presented in dark pink and green, that is, colors which were assigned to no response. Congruent, incongruent, and neutral trials were presented equally often.

In the *mixed blocks*, the material for the mixed blocks was the same as in the pure flanker blocks, except that colored X's were displayed as in the pure Simon blocks. All trial types (i.e., Simon incongruent and flanker incongruent, Simon incongruent and flanker congruent, Simon incongruent and flanker neutral, Simon congruent and flanker incongruent, Simon congruent and flanker congruent, Simon congruent and flanker neutral, Simon neutral and

flanker incongruent, Simon neutral and flanker congruent, Simon neutral and flanker neutral) occurred equally often.

**Procedure.** A trial sequence is illustrated in Fig 4. Each trial started with a 500-ms foreperiod. For the pure Simon and mixed blocks, yellow squares were presented in each of the locations during this foreperiod. For the stimulus-response mapping block and the pure flanker blocks, a yellow fixation cross was presented centrally during this foreperiod. Then, the stimulus was presented until a response or 5000 ms elapsed. To respond, participants used the four response keys *y*, *v*, *m*, and—of a standard German QWERTZ keyboard. These keys were mapped to the colors red, brown, violet, and white, respectively. In case of an error, the German word "Fehler" (*engl.* error) was displayed in yellow for 500 ms. This feedback was presented in the middle of each yellow square for the pure Simon and mixed blocks, and in the middle of the screen for the stimulus-response mapping block and the pure flanker blocks. Finally, a blank was presented during an intertrial interval of 1000 ms. In case pure Simon and mixed blocks, the yellow squares were displayed in each location during this blank.

All participants first performed the stimulus-response mapping block for practice. After this block, half of the participants started with a pure flanker block, followed by a pure Simon block, whereas the other half started with a pure Simon block followed by a pure flanker block. After the first two blocks, all participants performed ten mixed blocks. After these critical blocks, we again presented pure blocks of each congruency variable in order to have enough observations to compute the congruency effects. That is, participants performed a pure flanker block and a pure Simon block in the same order as in the first two pure blocks (e.g., participants who started with a pure flanker block, followed by a pure Simon block, had then a pure flanker block, followed by a pure Simon block). The stimulus-response block included 96 trials, pure blocks included 180 trials each, and mixed blocks included 144 trials each. An overall feedback, which consisted of mean reaction time (RT) and error rate, was displayed at the end of each block. Participants could take brief rests after each block.

At the beginning, after being informed, participants signed the informed consent sheet. Then, they were given short instructions about the task they had to carry out. Before each change of block type (e.g., from the pure flanker block to the mixed blocks), a short instruction for the now relevant task was also given. For the stimulus-response mapping block, participants were instructed to indicate the color of the stimulus. For the pure Simon blocks, they were instructed to indicate the color of the stimulus while ignoring the location of the stimulus on the screen. For the pure flanker blocks, they were instructed to indicate the color of the central letter while ignoring the flanking letters. For the mixed blocks, they were instructed to indicate the color of the central letter while ignoring the location on the screen and the color of the letters.

Participants were tested in group up to six during one session of approximately 2 hours. The experiment was programmed using Tscope5 [37] and run on IBM compatible computer.

**Data preparation.** The stimulus-response mapping block, which served as a practice block, was not analyzed. The first trial of each block was considered as a warm-up trial and thus was excluded. The dependent variables were reaction times (RTs) and error rates. We applied an arcsine square root transformation to the error rates for statistical analysis. For RTs, to exclude error-related cognitive processes [38], we additionally removed errors and one trial following an error from the raw data set. In Experiment 1a, this dismissed 13.50% of the trials for the mixed blocks, 12.22% of the trials for the pure Simon blocks, and 12.03% of the trials for the pure flanker blocks. In Experiment 1b, this dismissed 14.10% of the trials for the mixed blocks, 12.38% of the trials for the pure Simon blocks, and 14.67% of the trials for the pure flanker blocks.

We also excluded RTs faster than 3 standard deviations (SD) from the mean and slower than 3 SD from the mean for each trial type and participant. In Experiment 1a, this further dismissed 1.97% of the trials for the mixed blocks, 2.03% of the trials for the pure Simon blocks, and 1.96% of the trials for the pure flanker blocks. In Experiment 1b, this further dismissed 1.61% of the trials for the mixed blocks, 1.63% of the trials for the pure Simon blocks, and 1.78% of the trials for the pure flanker blocks

**Data analysis.** Analyses for the pure blocks are presented in S1 File (see S1a and S1b Table as well as S1 Fig). In the mixed blocks, we investigated whether or not an interaction between Simon and flanker congruency occurred. To this end, we first used a NHST approach by carrying out three two-way repeated-measures ANOVA with Simon congruency and flanker congruency. The focus in the first ANOVA was on the *congruency effect*, and thus each congruency variable included the level *incongruent* and *congruent*. In the second ANOVA, the focus was on the *interference effect*, and thus each congruency variable included the level *incongruent* and *neutral*. In the third ANOVA, the focus was on the *facilitation effect*, and thus each congruency variable included the level *neutral* and *congruent*. These ANOVAs were implemented in R [39] with the afex package [40]. An alpha level of 0.05 was used for all these tests. Effect sizes are expressed as $\eta_g^2$ values (generalized eta square) [41]. In case of significant two-way interactions, follow-up two-tailed *t*-tests were performed and effect sizes are expressed Cohen's *d*.

In addition, to assess not only the strength of evidence for the alternative hypothesis (e.g., the presence of the interaction) but also the strength of evidence for the null hypothesis (e.g., the absence of the interaction), we used a Bayesian approach by computing Bayesian ANOVAs with default prior scales. These were implemented in R with the BayesFactor package [42]. To compute the Bayes Factor (BF) in favor of the alternative hypothesis (i.e., $BF_{10}$) for the main effect model including either the Simon or flanker congruency, we compared the main effect model against the null model. To compute the $BF_{10}$ for the two-way interaction model, we compared the interaction model (i.e., the model with both main effects and the interaction) with the main-effects model (i.e., the model with both main effects but no interaction) by calculating the ratio $BF_{Interaction\ Model}$ / $BF_{Main\ Effects\ Model}$ [43]. For each model comparison, the Bayes Factor in favor of the null hypothesis ($BF_{01}$) was computed as $1/BF_{10}$. Following Raftery's [44] classification scheme, we considered a BF between 1–3 as weak evidence, between 3–20 as positive evidence, between 20–150 as strong evidence, and larger than 150 as very strong evidence.

## Results

In the present experiments, the most relevant results are the interaction between both congruency variables, that is, the modulation of the congruency effect (i.e., incongruent vs. congruent trials), the interference effect (i.e., incongruent vs. neutral trials), and the facilitation effect (i.e., neutral vs. congruent trials) of one congruency variable by the congruency of the other variable. The descriptive results for RTs and error rates are presented in Figs 5 and 6, for Experiment 1a and 1b, respectively. The results from the NHST and Bayesian analyses are presented in Tables 2 and 3 for Experiment 1a and 1b, respectively. The results are first presented for Experiment 1a, and then for Experiment 1b.

**Experiment 1a: Congruency effect (incongruent vs. congruent).** For RTs, the ANOVA including the variables Simon congruency (incongruent, congruent) and flanker congruency (incongruent, congruent) showed a significant interaction (see Table 2, left part). In line with the NHST analysis, the Bayesian analysis revealed small evidence in favor of the interaction model (see Table 2). Thus, the flanker congruency effect was smaller, but still significant, when

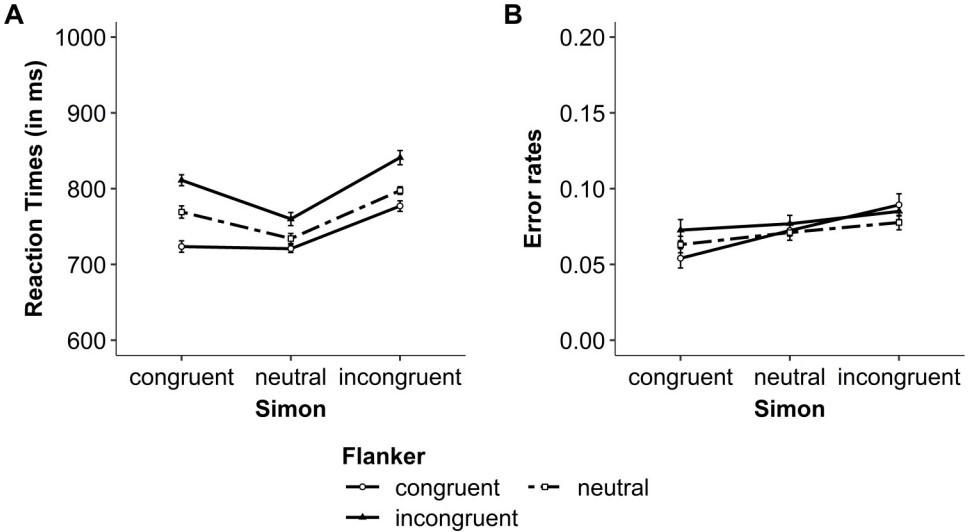

**Fig 5. Experiment 1a: Modulation of the flanker congruency effect by Simon congruency within a trial.** (A) Mean reaction times (RT) and (B) mean raw error rates for both flanker and Simon congruency. Error bars represent within-subject confidence intervals [see 45, 46].

the trials were Simon incongruent (64 ms), $t(28) = 8.89$, $p < .001$, $d = 1.65$, $BF_{10} = 9.38 \times 10^{6}$, $BF_{01} = 1.07 \times 10^{-7}$, compared to when they were Simon congruent (87 ms), $t(28) = 16.81$, $p < .001$, $d = 3.12$, $BF_{10} = 1.52 \times 10^{13}$, $BF_{01} = 6.60 \times 10^{-14}$ (see Fig 5A). Similarly, the Simon congruency effect was smaller, but still significant, when the trials were flanker incongruent (30 ms), $t(28) = 3.08$, $p = .005$, $d = 0.57$, $BF_{10} = 8.91$, $BF_{01} = 0.11$, compared to when they were flanker congruent (53 ms), $t(28) = 7.24$, $p < .001$, $d = 1.34$, $BF_{10} = 2.11 \times 10^{5}$, $BF_{01} = 4.74 \times 10^{-6}$.

For error rates, the ANOVA from the NHST approach showed a significant interaction between Simon and flanker congruency, and there was weak evidence in favor of the

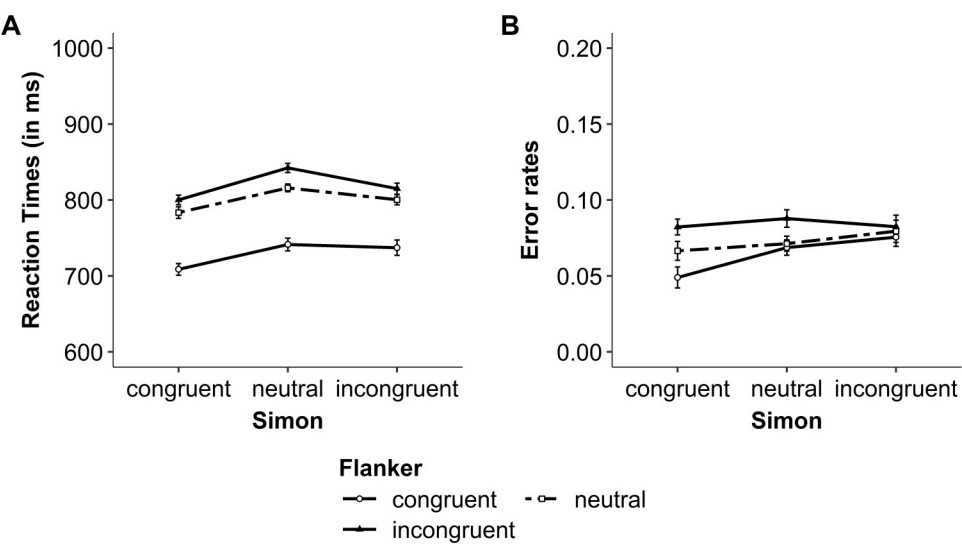

**Fig 6. Experiment 1b: Modulation of the flanker congruency effect by Simon congruency within a trial.** (A) Mean reaction times (RT) and (B) mean raw error rates for both flanker and Simon congruency. Error bars represent within-subject confidence intervals [see 45, 46].

**Table 2. Experiment 1a: Inferential statistical values for the two-way repeated-measures analysis of variance (ANOVA) with the variables Simon congruency and flanker congruency, and Bayes Factors (BF) from model comparisons.**

| Effect–Model | Reaction times | | | | | Arcsine transformed error rates | | | | |
|---|---|---|---|---|---|---|---|---|---|---|
| | ANOVA | | | Bayesian analysis | | ANOVA | | | Bayesian analysis | |
| | $F$ | $p$ | $\eta_g^2$ | $BF_{10}$ | $BF_{01}$ | $F$ | $p$ | $\eta_g^2$ | $BF_{10}$ | $BF_{01}$ |
| Congruency effect (incongruent vs. congruent) | | | | | | | | | | |
| Simon congruency | 33.00 | < .001 | .04 | 305.51 | $3.27 \times 10^{-3}$ | 18.41 | < .001 | .03 | $1.41 \times 10^3$ | $7.09 \times 10^{-4}$ |
| Flanker congruency | 322.79 | < .001 | .11 | $7.25 \times 10^{13}$ | $1.38 \times 10^{-14}$ | 5.48 | .027 | .007 | 1.08 | 0.92 |
| Two-way interaction | 6.35 | .018 | .003 | 1.65 | 0.61 | 7.00 | .013 | .007 | 2.36 | 0.42 |
| Interference effect (incongruent vs. neutral) | | | | | | | | | | |
| Simon congruency | 155.82 | < .001 | .10 | $3.89 \times 10^{15}$ | $2.57 \times 10^{-16}$ | 3.73 | .064 | .003 | 0.94 | 1.07 |
| Flanker congruency | 57.67 | < .001 | .03 | 66.56 | 0.02 | 4.54 | .042 | .004 | 1.60 | 0.63 |
| Two-way interaction | 3.43 | .074 | .002 | 0.92 | 1.09 | 0.00 | .980 | < .001 | 0.28 | 3.59 |
| Facilitation effect (neutral vs. congruent) | | | | | | | | | | |
| Simon congruency | 11.14 | .002 | .008 | 11.85 | 0.08 | 10.73 | .003 | .01 | 11.52 | 0.09 |
| Flanker congruency | 49.22 | < .001 | .02 | $1.17 \times 10^4$ | $8.52 \times 10^{-5}$ | 0.91 | .348 | .001 | 0.30 | 3.33 |
| Two-way interaction | 10.26 | .003 | .006 | 16.99 | 0.06 | 2.01 | .167 | .002 | 0.56 | 1.77 |

$df$s = (1, 28). Effect sizes are expressed as generalized $\eta^2$ values. $BF_{10}$ = Bayes Factor in favor of the alternative hypothesis (i.e., in favor of the effect). $BF_{01}$ = Bayes Factor in favor of the null hypothesis (i.e., in favor of the absence of the effect).

interaction model in the Bayesian analysis. Thus, the flanker congruency effect on error rates was small and not significant when the trials were Simon incongruent (-.004), $t(28)$ = -0.02, $p$ = .981, $d$ = 0.004, $BF_{10}$ = 0.20, $BF_{01}$ = 5.07. In contrast, it was slightly larger and significant

**Table 3. Experiment 1b: Inferential statistical values for the two-way repeated-measures analysis of variance (ANOVA) with the variables Simon congruency and flanker congruency, and Bayes Factors (BF) from model comparisons.**

| Effect–Model | Reaction times | | | | | Arcsine transformed error rates | | | | |
|---|---|---|---|---|---|---|---|---|---|---|
| | ANOVA | | | Bayesian analysis | | ANOVA | | | Bayesian analysis | |
| | $F$ | $p$ | $\eta_g^2$ | $BF_{10}$ | $BF_{01}$ | $F$ | $p$ | $\eta_g^2$ | $BF_{10}$ | $BF_{01}$ |
| Congruency effect (incongruent vs. congruent) | | | | | | | | | | |
| Simon congruency | 15.27 | < .001 | .01 | 1.23 | 0.82 | 7.66 | .010 | .02 | 6.62 | 0.15 |
| Flanker congruency | 231.32 | < .001 | .15 | $4.48 \times 10^{20}$ | $2.23 \times 10^{-21}$ | 39.05 | < .001 | .05 | 967.18 | $1.03 \times 10^{-3}$ |
| Two-way interaction | 1.44 | .240 | .001 | 0.48 | 2.07 | 12.26 | .002 | .02 | 21.53 | 0.05 |
| Interference effect (incongruent vs. neutral) | | | | | | | | | | |
| Simon congruency | 25.59 | < .001 | .01 | $1.03 \times 10^3$ | $9.70 \times 10^{-4}$ | 0.24 | .628 | < .001 | 0.21 | 4.84 |
| Flanker congruency | 28.50 | < .001 | .009 | 436.06 | $2.29 \times 10^{-3}$ | 5.71 | .024 | .01 | 2.48 | 0.40 |
| Two-way interaction | 1.75 | .196 | < .001 | 0.59 | 1.69 | 2.60 | .118 | .007 | 1.06 | 0.94 |
| Facilitation effect (neutral vs. congruent) | | | | | | | | | | |
| Simon congruency | 27.96 | < .001 | .02 | 34.48 | 0.03 | 4.68 | .039 | .02 | 4.10 | 0.24 |
| Flanker congruency | 171.79 | < .001 | .11 | $1.87 \times 10^{17}$ | $5.35 \times 10^{-18}$ | 15.09 | < .001 | .02 | 6.90 | 0.14 |
| Two-way interaction | 0.001 | .978 | < .001 | 0.27 | 3.66 | 10.25 | .003 | .01 | 3.84 | 0.26 |

$df$s = (1, 29). Effect sizes are expressed as generalized $\eta^2$ values. $BF_{10}$ = Bayes Factor in favor of the alternative hypothesis (i.e., in favor of the effect). $BF_{01}$ = Bayes Factor in favor of the null hypothesis (i.e., in favor of the absence of the effect).

when the trials were Simon congruent (.02), $t(28) = 4.06$, $p < .001$, $d = 0.75$, $BF_{10} = 83.61$, $BF_{01} = 0.01$ (see Fig 5B). Similarly, the Simon congruency effect was small and not significant when the trials were flanker incongruent (.01), $t(28) = 1.84$, $p = .076$, $d = 0.34$, $BF_{10} = 0.87$, $BF_{01} = 1.14$. In contrast, it was slightly larger and significant when the trials were flanker congruent (.04), $t(28) = 4.90$, $p < .001$, $d = 0.91$, $BF_{10} = 653.35$, $BF_{01} = 1.53 \times 10^{-3}$. Together, when the focus is on the congruency effect, the results showed an interaction between Simon and flanker congruency for both RTs and error rates.

**Experiment 1a: Interference effect (incongruent vs. neutral).** The ANOVA including the variables Simon congruency (incongruent, neutral) and flanker congruency (incongruent, neutral) showed no significant interaction, although the $p$-value approached the significance level (see Table 2, left part). Furthermore, the Bayesian analysis revealed weak evidence against the interaction model (see Table 2). The descriptive results suggest an over-additive interaction, rather than an under-additive interaction (see Fig 5A). That is, the flanker interference effect was observed in both Simon incongruent and neutral trials, but it was larger–at least descriptively–when the trials were Simon incongruent (43 ms), $t(28) = 6.11$, $p < .001$, $d = 1.14$, $BF_{10} = 1.34 \times 104$, $BF_{01} = 7.45 \times 10^{-5}$, compared to when they were Simon neutral (26 ms), $t(28) = 4.24$, $p < .001$, $d = 0.79$, $BF_{10} = 128.00$, $BF_{01} = 0.01$ (see Fig 5A). Similarly, the Simon interference effect was significant for both flanker incongruent and neutral trials, but it was larger–at least descriptively–when the trials were flanker incongruent (81 ms), $t(28) = 9.07$, $p < .001$, $d = 1.68$, $BF_{10} = 1.41 \times 10^7$, $BF_{01} = 7.07 \times 10^{-8}$, compared when they were flanker neutral (63 ms), $t(28) = 11.04$, $p < .001$, $d = 2.05$, $BF_{10} = 8.60 \times 10^8$, $BF_{01} = 1.16 \times 10^{-9}$. These results speak against an under-additive interaction between Simon and flanker congruency.

For error rates, the ANOVA from the NHST approach showed no significant interaction, and the Bayesian analysis revealed positive evidence against the interaction model. Thus, the flanker interference effect on error rates was similar for both Simon incongruent and neutral trials (.01), and the Simon interference effect was similar for both flanker incongruent and neutral trials (.01; see Fig 5B). Together, when the focus is on the interference effect, the results showed no under-additive interaction between Simon and flanker congruency for both RTs and error rates.

**Experiment 1a: Facilitation effect (neutral vs. congruent).** The ANOVA including the variables Simon congruency (neutral, congruent) and flanker congruency (neutral, congruent) showed a significant interaction (see Table 2, left part). In line with the NHST analysis, the Bayesian analysis reveals positive evidence in favor of the interaction model (see Table 2). Thus, the flanker facilitation effect was smaller, but still significant, when the trials were Simon neutral (13 ms), $t(28) = 2.36$, $p = .025$, $d = 0.44$, $BF_{10} = 2.11$, $BF_{01} = 0.47$, compared to when they were Simon congruent (46 ms), $t(28) = 6.28$, $p < .001$, $d = 1.16$, $BF_{10} = 1.86 \times 10^4$, $BF_{01} = 5.38 \times 10^{-5}$ (see Fig 5A). More surprisingly, a reversed Simon facilitation effect was observed in both flanker neutral and congruent trials. This was significant when the trials were flanker neutral (-35 ms), $t(28) = -4.35$, $p < .001$, $d = 0.81$, $BF_{10} = 168.11$, $BF_{01} = 0.01$, but it was not significant when the trials were flanker congruent (-3 ms), $t(28) = -0.41$, $p = .686$, $d = 0.08$, $BF_{10} = 0.21$, $BF_{01} = 4.69$.

For error rates, the ANOVA from the NHST approach showed no significant interaction, and the Bayesian analysis revealed weak evidence against the interaction model. Nevertheless, at the descriptive level, the results for the error rates mirror the pattern of results observed when the analyses focused on the congruency effect. That is, although the flanker facilitation effect was not significant in both Simon neutral and congruent trials, it was smaller when the trials were Simon neutral (-.001), $t(28) = -0.25$, $p = .808$, $d = 0.05$, $BF_{10} = 0.20$, $BF_{01} = 4.93$, compared to when they were Simon congruent (.009), $t(28) = 1.49$, $p = .148$, $d = 0.28$, $BF_{10} = 0.53$, $BF_{01} = 1.89$ (see Fig 5B). Furthermore, the Simon facilitation effect was small and not

significant when the trials were flanker neutral (.01), $t(28) = 1.07$, $p = .294$, $d = 0.20$, $BF_{10} = 0.33$, $BF_{01} = 3.01$. In contrast, it was slightly larger and significant when the trials were flanker congruent (.02), $t(28) = 4.49$, $p < .001$, $d = 0.83$, $BF_{10} = 239.45$, $BF_{01} = 4.18$ x $10^{-3}$.

**Experiment 1a: Discussion.** The results of Experiment 1a showed an under-additive interaction when the analyses focused on the congruency effect (i.e., incongruent vs. congruent). Critically, this interaction was not found when the focus was on the interference effect (i.e., incongruent vs. neutral). In contrast, the under-additive interaction was observed when the focus was on the facilitation effect (i.e., congruent vs. neutral). That is, the facilitation effect of one congruency variable was smaller when the trials of the other congruency variable were neutral compared to when they were congruent. Together, these findings suggest that the interaction between Simon and flanker congruency can be found even with a paradigm involving neutral trials and that the modulation of one congruency effect by the other congruency effect mainly involve priming or facilitation.

For the Simon congruency, we also observed a reverse facilitation effect (i.e., slower performance on congruent trials than on neutral trials; see [33] for a similar finding). This finding may result from the fact that our Simon neutral stimuli were presented at the center of the display where the visual acuity is higher and where no attentional selection process is required [36]. In contrast, in Simon congruent trials, stimuli were presented to the left or right side of the screen, thus requiring moving your attention to the location (left or right) on the screen. This additional attentional process may explain why performance was slower on Simon congruent trials than on neutral trials.

**Experiment 1b: Congruency effect (incongruent vs. congruent).** For RTs, the ANOVA including the variables Simon congruency (incongruent, congruent) and flanker congruency (incongruent, congruent) showed no significant interaction (see Table 3, left part). The Bayesian analysis revealed weak evidence against the interaction model (see Table 3). Nevertheless, at the descriptive level, the results mirror the pattern of results observed in the previous experiment. That is, the flanker congruency effect was smaller, but still significant, when the trials were Simon incongruent (78 ms), $t(29) = 8.79$, $p < .001$, $d = 1.60$, $BF_{10} = 1.01$ x $10^7$, $BF_{01} = 9.90$ x $10^{-8}$, compared to when they were Simon congruent (91 ms), $t(29) = 13.01$, $p < .001$, $d = 2.37$, $BF_{10} = 5.81$ x $10^{10}$, $BF_{01} = 1.72$ x $10^{-11}$ (see Fig 6A). Similarly, the Simon congruency effect was smaller, but still significant, when the trials were flanker incongruent (15 ms), $t(29) = 2.44$, $p = .021$, $d = 0.45$, $BF_{10} = 2.41$, $BF_{01} = 0.42$, compared to when they were flanker congruent (29 ms), $t(29) = 3.00$, $p = .005$, $d = 0.55$, $BF_{10} = 7.54$, $BF_{01} = 0.13$.

For error rates, the ANOVA from the NHST approach showed a significant interaction between Simon and flanker congruency, and there was strong evidence in favor of the interaction model in the Bayesian analysis. Thus, the flanker congruency effect on error rates was small and not significant when the trials were Simon incongruent (.01), $t(29) = 1.48$, $p = .148$, $d = 0.27$, $BF_{10} = 0.52$, $BF_{01} = 1.92$. In contrast, it was slightly larger and significant when the trials were Simon congruent (.03), $t(29) = 6.37$, $p < .001$, $d = 1.16$, $BF_{10} = 2.89$ x $10^4$, $BF_{01} = 3.46$ x $10^{-5}$ (see Fig 6B). Similarly, the Simon congruency effect was small and not significant when the trials were flanker incongruent (.0002), $t(29) = 0.04$, $p = .969$, $d = 0.01$, $BF_{10} = 0.19$, $BF_{01} = 5.14$. In contrast, it was slightly larger and significant when the trials were flanker congruent (.03), $t(29) = 4.12$, $p < .001$, $d = 0.75$, $BF_{10} = 101.27$, $BF_{01} = 0.01$. Together, the results showed that when the focus was on the congruency effect, the under-additive interaction between flanker and Simon congruency was significant for error rates, and only observed at the descriptive level for RTs.

**Experiment 1b: Interference effect (incongruent vs. neutral).** For both RTs and error rates, the ANOVA including the variables Simon congruency (incongruent, neutral) and flanker congruency (incongruent, neutral) showed no significant interaction and the Bayesian

analysis revealed inconclusive evidence regarding the interaction model (see Table 3, left part). Nevertheless, at the descriptive level, the RTs results mirror the pattern of results observed when the analyses focused on the congruency effect. That is, the flanker interference effect was smaller, but still significant, when the trials were Simon incongruent (15 ms), $t(29) = 2.29$, $p = .030$, $d = 0.41$, $BF_{10} = 1.82$, $BF_{01} = 0.55$, compared to when they were Simon neutral (26 ms), $t(29) = 5.00$, $p < .001$, $d = 0.91$, $BF_{10} = 910.54$, $BF_{01} = 1.10 \times 10^{-3}$ (see Fig 6A). Similarly, although the Simon interference effect was reversed in RTs (i.e., slower RTs for neutral trials than for incongruent trials), the Simon interference effect was smaller, but still significant, when the trials were flanker incongruent (-27 ms), $t(29) = -4.46$, $p < .001$, $d = 0.81$, $BF_{10} = 231.27$, $BF_{01} = 4.32 \times 10^{-3}$, compared to when they were flanker neutral (-16 ms), $t(29) = -2.55$, $p = .016$, $d = 0.46$, $BF_{10} = 2.96$, $BF_{01} = 0.34$.

For the error rates (see Fig 6B), the flanker interference effect was relatively similar for both Simon incongruent and neutral trials (.003 and .02, respectively). Similarly, the Simon interference effect was relatively similar for both flanker incongruent and neutral trials (-.01 and .01, respectively). Together, when the focus was on the interference effect, the results showed no interaction between flanker and Simon congruency in error rates, and an interaction in RTs but only at the descriptive level.

**Experiment 1b: Facilitation effect (neutral vs. congruent).** For RTs, the ANOVA including the variables Simon congruency (neutral, congruent) and flanker congruency (neutral, congruent) showed no significant interaction (see Table 3, left part). In line with the NHST analysis, the Bayesian analysis revealed positive evidence against the interaction model (see Table 3). Thus, the flanker facilitation effect was exactly similar for Simon neutral and Simon congruent trials (i.e., 33 ms for both trial types). Similarly, the Simon facilitation effect was exactly similar for both flanker neutral and congruent trials (i.e., 75 ms for both trial types; see Fig 6A).

For error rates, the ANOVA from the NHST approach showed a significant interaction between Simon and flanker congruency, and the Bayesian analysis revealed positive evidence in favor of the interaction model (see Table 3, right part). Thus, the flanker facilitation effect was smaller when the trials were Simon neutral (.003), $t(29) = 0.46$, $p = .649$, $d = 0.08$, $BF_{10} = 0.21$, $BF_{01} = 4.67$, compared to when they were Simon congruent (.02), $t(29) = 4.96$, $p < .001$, $d = 0.91$, $BF_{10} = 821.02$, $BF_{01} = 1.22 \times 10^{-3}$ (see Fig 6B). Similarly, the Simon facilitation effect was smaller when the trials were flanker neutral (.005), $t(29) = 0.18$, $p = .860$, $d = 0.03$, $BF_{10} = 0.20$, $BF_{01} = 5.07$, compared to when the trials were flanker congruent (.02), $t(29) = 3.71$, $p = .001$, $d = 0.68$, $BF_{10} = 36.93$, $BF_{01} = 0.03$. These results show that when the analyses focused on the facilitation effect, the interaction between Simon and flanker congruency was not significant for RTs, but significant for error rates.

**Experiment 1b: Discussion.** The results of Experiment 1b were more puzzling than those of the previous experiment. First, when error rates were taken into consideration, the results replicated previous findings. That is, overall error rates increased linearly across Simon congruent, neutral and incongruent trials (see Fig 6B), confirming that performance on Simon neutral trials represent an appropriate baseline for error rates [33]. More critically, the results of the present study showed an under-additive interaction for error rates when the analyses focused on the congruency effect (i.e., incongruent vs. congruent) and the facilitation effect (i.e., congruent vs. neutral). However, no such interaction was observed when the focus was on the interference effect (i.e., incongruent vs. neutral). In line with Experiment 1a, these findings emphasize priming or facilitation in the interaction between Simon and flanker congruency.

In contrast, for the RTs, no significant interaction between Simon and flanker congruency was found in any of the three effects (congruency, interference, and facilitation). Only at the

descriptive level, the interaction was observed when the analyses focused on the congruency effect and the interference effect. However, contrary to previous research in which performance increased linearly across Simon congruent, neutral and incongruent trials [33], the present results showed the slowest performance on the Simon neutral trials (see Fig 6A and S1 File for the same pattern of results in the pure blocks). This challenges the view that these Simon neutral trials represent a good baseline for RT performance [33], thus emphasizing the necessity to interpret with caution the RT findings of this experiment.

## Experiment 2

Experiment 2 was designed to determine whether the interaction between Simon and flanker congruency is affected by the overlap in spatial configuration. To this end, we tested two groups of participants in which the spatial configuration was manipulated orthogonally between both congruency variables. In the first group ("Simon horizontal–flanker vertical"), the overlap in spatial configuration was similar to Experiment 1a. That is, the four locations for the Simon congruency were presented *horizontally*, and the flanking characters for the flanker congruency were presented *vertically* (i.e., above and below the target). In the second group ("Simon vertical–flanker horizontal"), the four locations for the Simon congruency were presented *vertically*, and the flanking characters for the flanker congruency were presented *horizontally* (i.e., on the left and right side of the target). We hypothesized that if the interaction between Simon and flanker congruency is affected by the spatial overlap in both congruency variables [30], it should be observed in none of the groups.

## Method

**Participants.**  Forty-eight new participants (24 in each group) took part in the experiment (42 women, 47 right-handed, mean$_{age}$ = 21.7 years, SD$_{age}$ = 3.9).

**Material.**  The material was the same as in Experiment 1 except for the following modifications. First, there were only four trial types, occurring with an equal probability of 25%: Simon incongruent and flanker incongruent, Simon incongruent and flanker congruent, Simon congruent and flanker incongruent, Simon congruent and flanker congruent.

Second, each stimulus was displayed in one of four squares. A square comprised a side length of 3.72˚ visual angle. For the group of participants "Simon horizontal–flanker vertical", the four squares were presented horizontally in the center of the screen. For the group "Simon vertical–flanker horizontal", the four squares were presented vertically in the center of the screen. For both groups, the distance between the squares was 2.77˚ visual angle.

In order to have an orthogonal manipulation between Simon and flanker congruency variables, the flanker X's were presented above and below the target for the "Simon horizontal–flanker vertical" group (see Fig 7). Thus, all three letters comprised a height of 1.62˚ visual angle and a width of 0.57˚ visual angle. For the "Simon vertical–flanker horizontal" group, the flanker X's were presented left and right to the target. Thus, all three letters comprised a height of 0.57˚ visual angle and a width of 2.01˚ visual angle.

**Procedure.**  The procedure was similar to Experiment 1, except that participants used the four response keys *y*, *v*, *m*, and—in the "Simon horizontal–flanker vertical" group, and 5, *t*, *h*, and *b* in the "Simon vertical–flanker horizontal" group. These keys were used to give rise to the horizontal and vertical Simon congruency, respectively. The keys *y*, *v*, *m*, and—were mapped to the colors red, brown, violet, and white, respectively. The keys 5, *t*, *h*, and *b* were mapped to the colors red, brown, violet, and white, respectively. Furthermore, we asked participants to perform six mixed blocks with 144 trials each and two pure blocks for each Simon

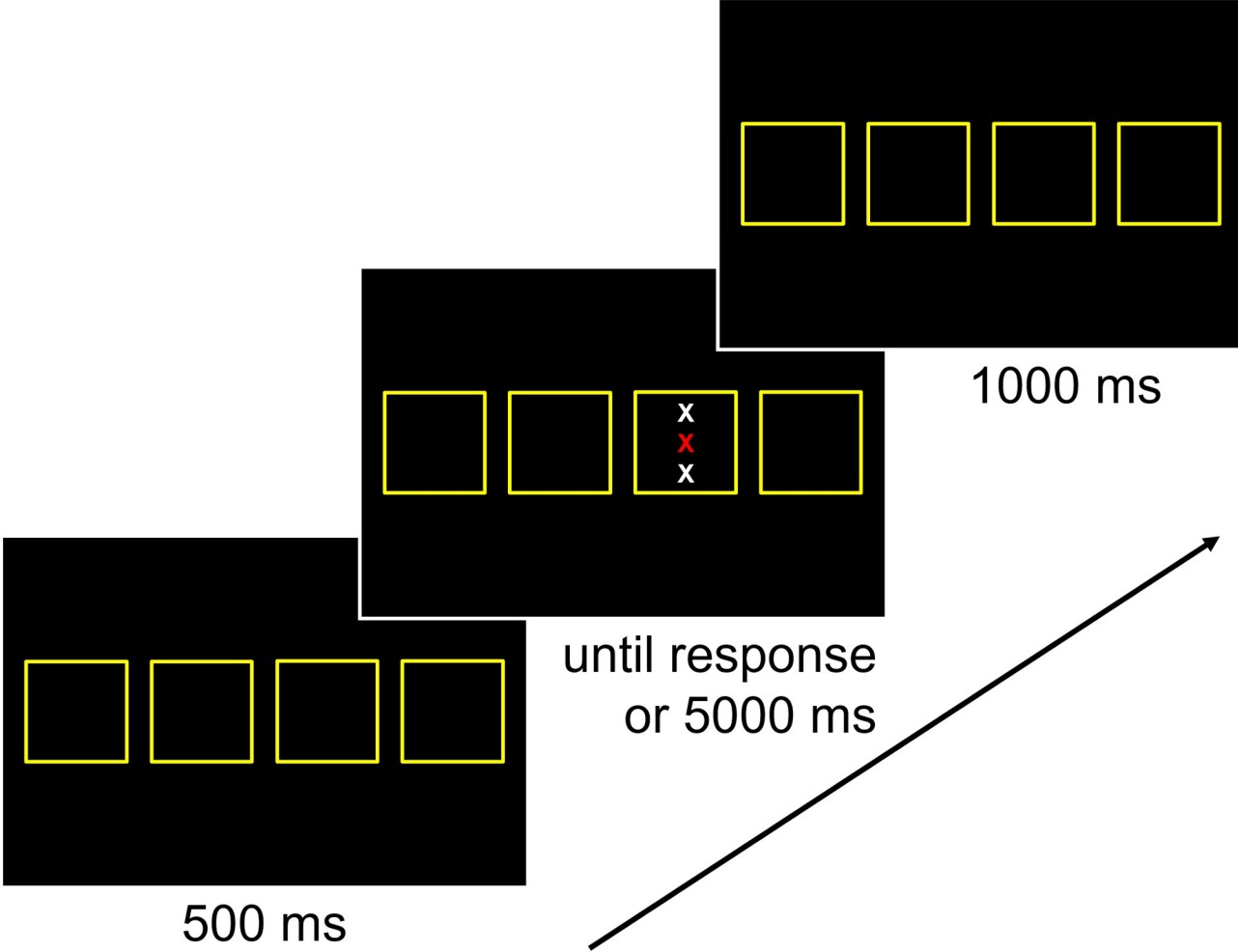

**Fig 7. Experiment 2: Example of one trial sequence in the position "Simon horizontal–flanker vertical".** Participants were asked to indicate the color of the central letter while ignoring the flanking letters and their location on the screen. They used the four response keys *y*, *v*, *m*, and -, which were mapped to the colors red, brown, violet, and white, respectively.

and flanker congruency with 192 trials each. Participants were tested during one session of approximately 90 min.

**Data preparation.** The data preparation was similar to Experiment 1. Removing errors and one trial following an error from the raw data set dismissed 10.75% of the trials for the mixed blocks, 10.72% of the trials for the pure Simon blocks, and 11.39% of the trials for the pure flanker blocks. Excluding RTs faster and slower than 3 SDs from the mean further dismissed 1.79% of the trials for the mixed blocks, 2% of the trials for the pure Simon blocks, and 2.01% of the trials for the pure flanker blocks

**Data analysis.** The data analysis was the same as in Experiment 1, except that a three-way ANOVA with Simon congruency (incongruent, congruent) and flanker congruency (incongruent, congruent) as within-subject variables and position (Simon horizontal–flanker vertical, Simon vertical–flanker horizontal) as a between-subjects variable was carried out. Furthermore, the $BF_{10}$ for the three-way interaction model was computed by comparing the three-way interaction model (i.e., the model with both main effects, all two-way interactions, and the

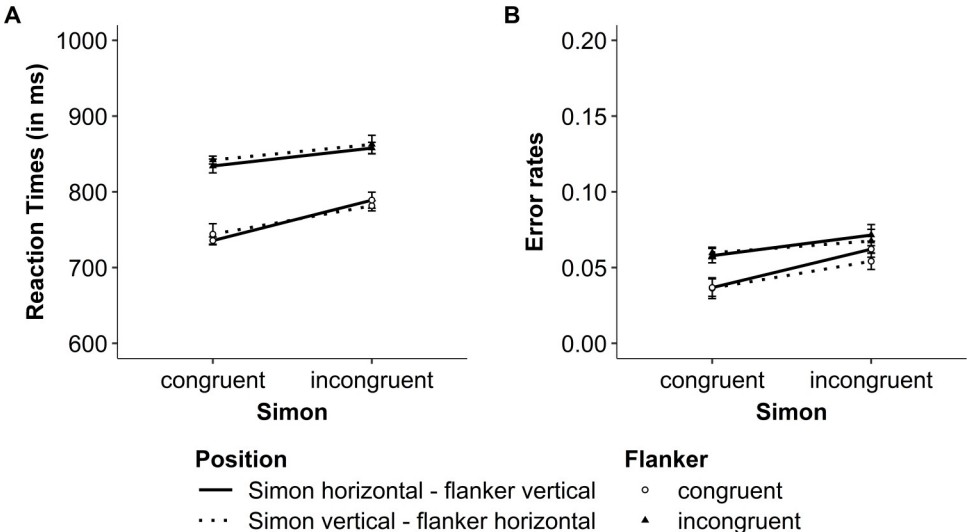

**Fig 8. Experiment 2: Modulation of the flanker congruency effect by Simon congruency within a trial.** (A) Mean reaction times (RT) and (B) mean raw error rates for both flanker and Simon congruency. Error bars represent within-subject confidence intervals [see 45, 46].

three-way interaction) with the model including the main effects and the two-way interactions by calculating the ratio $BF_{Three-way\ Interaction\ Model}$ / $BF_{Main\ Effects\ and\ Two-way\ Interactions\ Model}$.
Analyses for the pure blocks are presented in S1 File (see S1c Table and S1 Fig).

## Results

In this experiment, we investigated whether participants in both positions (i.e., Simon horizontal–flanker vertical and Simon vertical–flanker horizontal) show the interaction between Simon and flanker congruency. The descriptive results for RTs and error rates are presented in Fig 8. The results from the NHST and Bayesian analyses are presented in Table 4.

For RTs, the ANOVA showed a significant two-way interaction between Simon congruency and flanker congruency (see Table 4, left part). In line with the NHST analysis, the Bayesian

**Table 4. Experiment 2: Inferential statistical values for the three-way analysis of variance (ANOVA) with Simon congruency (incongruent, congruent) and flanker congruency (incongruent, congruent) as within-subject variables, and position (simon horizontal–flanker vertical, Simon vertical–flanker horizontal) as a between-subjects variable, and Bayes Factors (BF) from model comparisons.**

| Effect–Model | Reaction times | | | | | Arcsine transformed error rates | | | | |
| --- | --- | --- | --- | --- | --- | --- | --- | --- | --- | --- |
| | ANOVA | | | Bayesian analysis | | ANOVA | | | Bayesian analysis | |
| | $F$ | $p$ | $\eta_g^2$ | $BF_{10}$ | $BF_{01}$ | $F$ | $p$ | $\eta_g^2$ | $BF_{10}$ | $BF_{01}$ |
| Simon congruency | 44.45 | < .001 | .02 | 167.31 | 0.01 | 25.23 | < .001 | .03 | $9.21 \times 10^3$ | $1.09 \times 10^{-4}$ |
| Flanker congruency | 314.38 | < .001 | .10 | $2.91 \times 10^{30}$ | $3.44 \times 10^{-31}$ | 45.33 | < .001 | .04 | $2.25 \times 10^5$ | $4.44 \times 10^{-6}$ |
| Position | 0.01 | .927 | < .001 | 0.47 | 2.13 | 0.13 | .722 | .002 | 0.43 | 2.32 |
| Simon congruency x flanker congruency | 12.53 | < .001 | .002 | 3.86 | 0.26 | 8.60 | .005 | .007 | 3.61 | 0.28 |
| Simon congruency x position | 0.87 | .357 | < .001 | 0.25 | 4.04 | 1.82 | .184 | .003 | 0.47 | 2.14 |
| Flanker congruency x position | 0.36 | .551 | < .001 | 0.24 | 4.20 | 0.32 | .577 | < .001 | 0.25 | 4.08 |
| Three-way interaction | 0.94 | .336 | < .001 | 0.35 | 2.84 | 0.02 | .879 | < .001 | 0.28 | 3.58 |

$df$s = (1, 46) for the $F$ statistic. Effect sizes are expressed as generalized $\eta^2$ values. $BF_{10}$ = Bayes Factor in favor of the alternative hypothesis (i.e., in favor of the effect). $BF_{01}$ = Bayes Factor in favor of the null hypothesis (i.e., in favor of the absence of the effect).

analysis revealed positive evidence in favor of the two-way interaction model (see Table 4). In contrast, the three-way interaction between Simon congruency, flanker congruency and position was not significant, and the Bayesian analysis rather suggested weak evidence against the three-way interaction model (see Table 4). Thus, for both positions, the flanker congruency effect was smaller, but still significant, when the trials were Simon incongruent (75 ms), $t(47)$ = 12.24, $p < .001$, $d = 1.77$, $BF_{10} = 1.94$ x $10^{13}$, $BF_{01} = 5.16$ x $10^{-14}$, compared to when they were Simon congruent (98 ms), $t(47) = 17.57$, $p < .001$, $d = 2.54$, $BF_{10} = 1.51$ x $10^{19}$, $BF_{01} = 6.60$ x $10^{-20}$ (see Fig 8A). Similarly, the Simon congruency effect was smaller, but still significant, when the trials were flanker incongruent (22 ms), $t(47) = 4.00$, $p < .001$, $d = 0.58$, $BF_{10} = 111.52$, $BF_{01} = 0.01$, compared to when they were flanker congruent (45 ms), $t(47) = 6.99$, $p < .001$, $d = 1.01$, $BF_{10} = 1.48$ x $10^6$, $BF_{01} = 6.76$ x $10^{-7}$.

A similar pattern was observed in error rates (see Table 4, right part). The two-way interaction between Simon congruency and flanker congruency was significant and there was positive evidence in favor of the two-way interaction model in the Bayesian analysis. In contrast, the three-way interaction between Simon congruency, flanker congruency and position was not significant, and the Bayesian analysis suggested positive evidence against the three-way interaction model (see Table 4). Thus, for both positions, the flanker congruency effect in error rates was smaller, but still significant, when the trials were Simon incongruent (.01), $t(47)$ = 2.70, $p = .010$, $d = 0.39$, $BF_{10} = 3.92$, $BF_{01} = 0.26$, compared to when they were Simon congruent (.02), $t(47) = 8.44$, $p < .001$, $d = 1.22$, $BF_{10} = 1.75$ x $10^8$, $BF_{01} = 5.72$ x $10^{-9}$ (see Fig 8B). Similarly, the Simon congruency effect in error rates was smaller, but still significant, when the trials were flanker incongruent (.01), $t(47) = 2.22$, $p = .032$, $d = 0.32$, $BF_{10} = 1.44$, $BF_{01} = 0.69$, compared to when they were flanker congruent (.02), $t(47) = 5.87$, $p < .001$, $d = 0.85$, $BF_{10} = 3.70$ x $10^4$, $BF_{01} = 2.70$ x $10^{-5}$ (see Fig 8B).

## Discussion

The results showed an under-additive interaction between Simon and flanker congruency in both dependent measures (RTs and error rates) for both the "Simon horizontal–flanker vertical" group and the "Simon vertical–flanker horizontal" group. Therefore, this interaction occurred irrespective of the spatial overlap between both congruency variables.

## Experiment 3

Experiment 3 was designed to investigate whether the interaction observed between Simon and flanker congruency is affected by the learning of stimulus-response pairings when a large stimulus set is used. To this end, in contrast to Rey-Mermet and Gade [7] and the previous experiments of the present study, the number of stimulus-response pairings was equated across all trial types. That is, four stimulus-response pairings were selected randomly for each participant (i.e., four stimulus stimulus-response pairings for congruent-congruent, congruent-incongruent, incongruent-congruent, and incongruent-incongruent trial types).

We hypothesized that if the learning of stimulus-response pairings caused by the unequal numbers of stimulus stimulus-response pairings affects the interaction between Simon and flanker congruency in the previous experiments as well as in Rey-Mermet and Gade [7], its impact would be controlled in Experiment 3 because all stimulus-response pairings were presented equally often. Accordingly, performance on the congruent-congruent trial type would not be the fastest and the most accurate, and one congruency effect (e.g., the flanker congruency effect) should no longer differ depending on the congruency of the other variable (e.g., the Simon congruency).

## Method

**Participants.** Thirty-six new participants took part in the experiment (28 women, 32 right-handed, $mean_{age}$ = 23.1 years, $SD_{age}$ = 3.8).

**Material.** The material was the same as in Experiment 2, except that four stimulus-response pairings were selected randomly for each participant and each trial type. This concerned the incongruent trials in the pure blocks as well as the incongruent-congruent, congruent-incongruent, and incongruent-incongruent trials in the mixed blocks.

**Procedure.** The procedure was similar to Experiment 2.

**Data preparation.** The data preparation was similar to Experiments 1 and 2. Removing errors and one trial following an error from the raw data set dismissed 10.75% of the trials for the mixed blocks, 11.97% of the trials for the pure Simon blocks, and 14.3% of the trials for the pure flanker blocks. Excluding RTs faster and slower than 3 SDs from the mean further dismissed 1.82% of the trials for the mixed blocks, 1.73% of the trials for the pure Simon blocks, and 1.97% of the trials for the pure flanker blocks.

**Data analysis.** The data analysis was the same as in Experiments 1 and 2, except for the following modifications. First, a two-way repeated-measures ANOVA with the variables Simon congruency (incongruent, congruent) and flanker congruency (incongruent, congruent) was carried out. Then, we also determined the specific contribution of the learning of stimulus-response pairings by comparing performance in Experiment 3 with the corresponding condition of Experiment 2 (i.e., the "Simon horizontal–flanker vertical" group). To this end, we computed a three-way ANOVA with Simon congruency (incongruent, congruent) and flanker congruency (incongruent, congruent) as within-subject variables and experiment (2, 3) as a between-subjects variable. Analyses for the pure blocks are presented in S1 File (see S1d Table and S1 Fig).

## Results

In this experiment, we first focused on the modulation of the congruency effect of one variable depending on whether the trials of the other variable are congruent or incongruent (i.e., the interaction between both congruency variables). The descriptive results for RTs and error rates are presented in Fig 9. The results from the NHST and Bayesian analyses are presented in Table 5. In a second step, we compared performance in Experiment 3 with the corresponding condition of Experiment 2 (i.e., the "Simon horizontal–flanker vertical" group). For this analysis, the results are presented in Table 6.

**Interaction between Simon and flanker congruency.** For RTs, the ANOVA showed a significant interaction between Simon congruency and flanker congruency (see Table 5, left part). In line with the NHST analysis, the Bayesian analysis revealed positive evidence in favor of the interaction model (see Table 5). Thus, the flanker congruency effect was smaller, but still significant, when the trials were Simon incongruent (58 ms), $t(35)$ = 11.65, $p$ < .001, $d$ = 1.94, $BF_{10}$ = 5.36 x $10^{10}$, $BF_{01}$ = 1.78 x $10^{-11}$, compared to when they were Simon congruent (85 ms), $t(35)$ = 11.74, $p$ < .001, $d$ = 1.96, $BF_{10}$ = 6.82 x $10^{10}$, $BF_{01}$ = 1.47 x $10^{-11}$ (see Fig 9A). Similarly, the Simon congruency effect was smaller–but not significant–when the trials were flanker incongruent (10 ms), $t(35)$ = 1.73, $p$ = .092, $d$ = 0.29, $BF_{10}$ = 0.69, $BF_{01}$ = 1.45, compared to when they were flanker congruent (37 ms), $t(35)$ = 6.39, $p$ < .001, $d$ = 1.06, $BF_{10}$ = 6.52 x $10^4$, $BF_{01}$ = 1.53 x $10^{-5}$.

For error rates, although the Bayesian analysis revealed weak evidence against the interaction model, the ANOVA from the NHST approach showed a significant interaction between Simon congruency and flanker congruency (see Table 5, right part). Thus, the flanker congruency effect in error rates was smaller, but still significant, when the trials were Simon

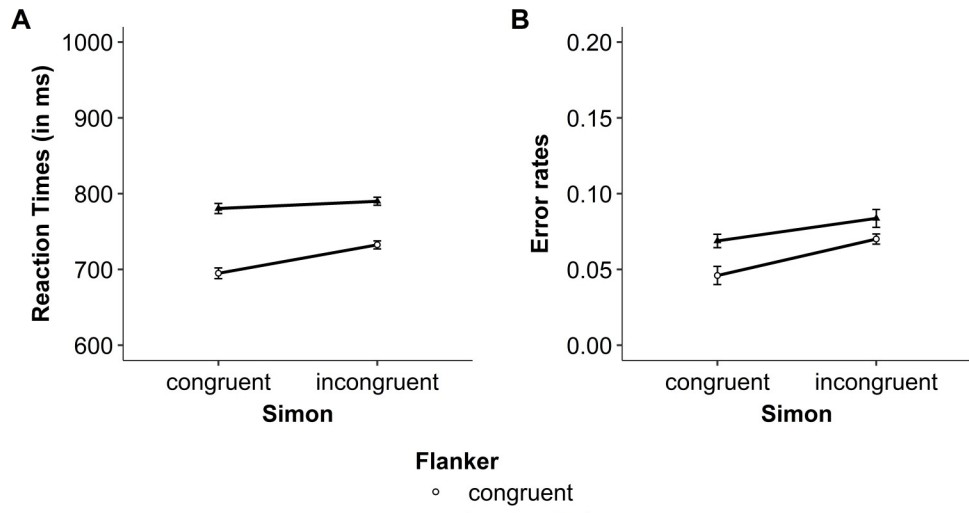

**Fig 9. Experiment 3: Modulation of the flanker congruency effect by Simon congruency within a trial.** (A) Mean reaction times (RT) and (B) mean raw error rates for both flanker and Simon congruency. Error bars represent within-subject confidence intervals [see 45, 46].

incongruent (.01), $t(35) = 3.09$, $p = .004$, $d = 0.52$, $BF_{10} = 9.54$, $BF_{01} = 0.10$, compared to when they were Simon congruent (.02), $t(35) = 4.52$, $p < .001$, $d = 0.75$, $BF_{10} = 351.48$, $BF_{01} = 2.85 \times 10^{-3}$ (see Fig 9B). Similarly, the Simon congruency effect was smaller, but still significant when the trials were flanker incongruent (.01), $t(35) = 3.47$, $p = .001$, $d = 0.58$, $BF_{10} = 23.26$, $BF_{01} = 0.04$, compared to when they were flanker congruent (.02), $t(35) = 6.18$, $p < .001$, $d = 1.03$, $BF_{10} = 3.62 \times 10^{4}$, $BF_{01} = 2.76 \times 10^{-5}$.

**Comparison of Experiment 2 and Experiment 3.** For RTs, the ANOVA comparing performance in Experiment 3 with the corresponding condition of Experiment 2 (i.e., the "Simon horizontal–flanker vertical" group) showed a significant two-way interaction between Simon and flanker congruency, but no three-way significant interaction between Simon congruency, flanker congruency, and experiment (see Table 6, left part). In line with the NHST analysis, the Bayesian analysis revealed very strong evidence in favor of the model including the interaction between Simon and flanker congruency, but positive evidence against the three-way interaction model (see Table 6).

For error rates, the pattern of results was similar. That is, the interaction between Simon and flanker congruency was significant, and there was positive evidence in favor of the two-

**Table 5. Experiment 3: Inferential statistical values for the two-way repeated-measures analysis of variance (ANOVA) with the variables Simon congruency (incongruent, congruent) and flanker congruency (incongruent, congruent), and Bayes Factors (BF) from model comparisons.**

| Effect–Model | Reaction times | | | | | Arcsine transformed error rates | | | | |
|---|---|---|---|---|---|---|---|---|---|---|
| | ANOVA | | | Bayesian analysis | | ANOVA | | | Bayesian analysis | |
| | *F* | *p* | $\eta_g^2$ | $BF_{10}$ | $BF_{01}$ | *F* | *p* | $\eta_g^2$ | $BF_{10}$ | $BF_{01}$ |
| Simon congruency | 34.10 | < .001 | .009 | 6.71 | 0.15 | 32.68 | < .001 | .05 | $3.84 \times 10^4$ | $2.60 \times 10^{-5}$ |
| Flanker congruency | 227.43 | < .001 | .08 | $1.14 \times 10^{23}$ | $8.77 \times 10^{-24}$ | 20.14 | < .001 | .03 | 978.97 | $1.02 \times 10^{-3}$ |
| Two-way interaction | 12.02 | .001 | .003 | 19.38 | 0.05 | 5.93 | .020 | .004 | 0.81 | 1.23 |

*df*s = (1, 35) for the *F* statistic. Effect sizes are expressed as generalized $\eta^2$ values. $BF_{10}$ = Bayes Factor in favor of the alternative hypothesis (i.e., in favor of the effect). $BF_{01}$ = Bayes Factor in favor of the null hypothesis (i.e., in favor of the absence of the effect).

**Table 6. Experiment 2 (Simon horizontal–flanker vertical) vs. Experiment 3: Inferential statistical values for the three-way analysis of variance (ANOVA) with Simon congruency (incongruent, congruent) and flanker congruency (incongruent, congruent) as within-subject variables, and experiment (Experiment 2, Experiment 3) as a between-subjects variable, and Bayes Factors (BF) from model comparisons.**

| Effect–Model | Reaction times | | | | | Arcsine transformed error rates | | | | |
| | ANOVA | | | Bayesian analysis | | ANOVA | | | Bayesian analysis | |
| | $F$ | $p$ | $\eta_g^2$ | $BF_{10}$ | $BF_{01}$ | $F$ | $p$ | $\eta_g^2$ | $BF_{10}$ | $BF_{01}$ |
|---|---|---|---|---|---|---|---|---|---|---|
| Simon congruency | 74.41 | $< .001$ | .01 | 774.48 | $1.29 \times 10^{-3}$ | 48.25 | $< .001$ | .05 | $4.11 \times 10^8$ | $2.43 \times 10^{-9}$ |
| Flanker congruency | 443.28 | $< .001$ | .08 | $2.29 \times 10^{38}$ | $4.36 \times 10^{-39}$ | 36.12 | $< .001$ | .03 | $3.16 \times 10^5$ | $3.17 \times 10^{-6}$ |
| Experiment | 2.54 | .117 | .04 | 0.97 | 1.03 | 0.61 | .438 | .009 | 0.48 | 2.08 |
| Simon congruency x flanker congruency | 20.40 | $< .001$ | .003 | 316.36 | $3.16 \times 10^{-3}$ | 12.52 | $< .001$ | .005 | 3.15 | 0.32 |
| Simon congruency x experiment | 4.35 | .041 | $< .001$ | 0.33 | 3.02 | 0.01 | .929 | $< .001$ | 0.20 | 5.11 |
| Flanker congruency x experiment | 2.72 | .105 | $< .001$ | 0.28 | 3.58 | 0.06 | .813 | $< .001$ | 0.21 | 4.84 |
| Three-way interaction | 0.02 | .888 | $< .001$ | 0.33 | 3.00 | 0.21 | .651 | $< .001$ | 0.27 | 3.76 |

$df$s = (1, 58) for the $F$ statistic. Effect sizes are expressed as generalized $\eta^2$ values. $BF_{10}$ = Bayes Factor in favor of the alternative hypothesis (i.e., in favor of the effect). $BF_{01}$ = Bayes Factor in favor of the null hypothesis (i.e., in favor of the absence of the effect).

way interaction model in the Bayesian analysis. In contrast, the three-way interaction was not significant, and there was positive evidence against the three-way interaction model (see Table 6, right part).

## Discussion

The results of Experiment 3 show that an under-additive interaction between Simon and flanker congruency was observed in both dependent measures (RTs and error rates) even when the number of stimulus-response pairings was the same across all trial types. Moreover, this interaction did not differ from the interaction observed in the corresponding condition of Experiment 2. This confirms the presence of the interaction when the learning of stimulus-response pairings was controlled for. Thus, the discrepancy of results observed in previous research using a large stimulus set (see Table 1) could not result from a difference in the number of stimulus-response pairings.

## Discussion

The purpose of the present study was to shed more light on the variables influencing the interaction between Simon and flanker congruency. In Experiments 1a and 1b, we investigated whether the interaction between Simon and flanker congruency is affected by the priming or facilitation occurring when the trials were both Simon and flanker congruent (i.e., when the correct response was co-activated by the irrelevant location on the screen as well as by the irrelevant color of the flanker X's). To this end, we included neutral trials for each congruency variable so that we could disentangle the interference caused by incongruent trials and the priming caused by congruent trials. Trials were flanker neutral when the flankers were presented in a color associated to none of the responses. Trials were Simon neutral when the stimulus was presented either in the middle of the screen (Experiment 1a) or at the top or bottom of the screen (Experiment 1b).

Using both a NHST and a Bayesian hypothesis testing approach, we found in Experiment 1a an interaction when the analyses focused on the facilitation effect (i.e., the difference between congruent and neutral trials), but not when they focused on the interference effect (i.e., the difference between incongruent and neutral trials). In Experiment 1b, these results were replicated for the error rates but not for RTs. Together, the RTs findings of Experiments

1a and 1b emphasized the difficulty of finding appropriate Simon neutral trials on which performance is slower than congruent trials but faster than incongruent trials. This seems to be specific to the present paradigm in which both Simon and flanker congruency variables were combined as most previous research showed appropriate neutral Simon conditions in standard Simon tasks [36, 47, 48].

In Experiments 2 and 3, we further tested whether the interaction between Simon and flanker congruency was affected by a spatial overlap in task features or by the learning of stimulus-response pairings. The results showed an interaction between Simon and flanker congruency in these experiments, ruling out that the interaction was influenced by spatial feature overlap and by the learning of stimulus-response pairings.

## Multiplicative priming of the correct response

Together, the results of the present study suggest that the multiplicative priming account is a plausible explanation for the interaction between Simon and flanker congruency [17, 32]. That is, when the trial was both Simon and flanker congruent (i.e., congruent-congruent trials), both irrelevant features–the irrelevant location on the screen and the irrelevant color of the flankers–prime the correct response. This multiple priming effect leads to overadditive facilitation provided that the effect of each congruent feature is proportional to the activation of the response. As a consequence, during trial processing, the different priming effects increase multiplicatively the activation level of the correct response. The information-accumulation is then accelerated towards the correct response so that the correct response is selected faster and more frequently. This selectively decreases the RTs and increases accuracy rates in congruent-congruent trials compared to mixed trials (i.e., incongruent-congruent or congruent-incongruent trials) in which the correct response is primed by only one irrelevant feature (i.e., either the irrelevant location on the screen for Simon congruent–flanker incongruent trials or the irrelevant color of the flankers for Simon incongruent–flanker congruent trials).

In previous research [17, 18], response priming was also used to account for the slightly faster RTs and more accurate responses observed sometimes in incongruent-incongruent trials. This was possible because in this research, the decision to perform was a two-choice decision. In this case, when the trials of both Simon and flanker congruency variables were incongruent, the incongruent response location and the incongruent flanker feature primed the same incorrect response (see Fig 1). This could have strengthened the activation of the incorrect response so that this response was faster classified as incorrect. In contrast, when the decision to perform is a four-choice decision, such as in the present study, the incongruent response location and the incongruent flanker feature may prime different responses (see Fig 3). Therefore, future work might test the assumption that there is some benefit or facilitation when two irrelevant features prime the same incorrect response, for example by comparing performance in incongruent-incongruent trials when both irrelevant features (i.e., irrelevant location and flankers) prime the same incorrect response or different incorrect responses.

## Impact of response speed

In some previous studies [5–7, 17], it was investigated whether response speed affected the interaction between Simon and flanker congruency. Whereas there is some evidence that response speed had no impact on the interaction [7], there is also some evidence suggesting an impact [5, 6, 17]. However, there are some inconsistencies. For example, Treccani and colleagues [17] observed a significant Simon congruency for flanker congruent trials at the end of the RT distributions (i.e., when the RTs were slow), and a reversed Simon congruency for flanker incongruent trials throughout the RT distribution (but only at the descriptive level). In

contrast, Hommel [6] found a significant Simon congruency effect for flanker congruent and incongruent trials in the first part of the RT distribution (i.e., when the RTs were fast to moderate). Only close to the end of the RT distribution, the Simon congruency effect reversed for flanker incongruent trials.

To determine the impact of response speed on our results, we computed delta plots for each experiment. Delta plots illustrate how the congruency effect vary as a function of response speed [see 49]. These are presented in S2 File. The delta plots demonstrate that the size of interaction effect increased as the RTs slowed down, suggesting an impact of response speed in the interactions observed in the present study. Moreover, across the experiments, the delta plots showed an inversed U-curved shape for the Simon congruency effect, in particular in flanker incongruent trials. That is, the Simon congruency effect initially increased while the RTs increased from fast to moderate. However, with further RT slowing, the Simon congruency effect diminished and sometimes reversed. Thus, these results are rather in line with Hommel [6] than with Treccani and colleagues [17].

There might be several reasons for this discrepancy. A first reason might be that in contrast to the present study and Hommel [6], Treccani and colleagues [17] used an accessory-stimulus Simon task. In this kind of task, because the target is presented at fixation, participants are forced to analyse the target first, and then they move their attention to the irrelevant accessory stimulus. Thus, the interfering information is not the position of the target, but the position of another object (i.e., the accessory stimulus). The irrelevant spatial feature causing the effect takes time to be coded, which can explain why the Simon congruency effect increased as the RT increased. Another reason for the discrepancy in results might be that in the present study as well as in Hommel [6], the Simon and flanker congruency variables were so combined that the irrelevant features for the Simon and flanker congruency variables were conveyed by different objects (i.e., the irrelevant locations on the screen and the irrelevant flanker colored letters, respectively). In contrast, in Treccani and colleagues [17], the same object–that is, an irrelevant colored square presented either to the left or right of the screen–was assumed to convey the irrelevant features for both Simon and flanker congruency variables. Thus, future studies could investigate whether combining both irrelevant features of the Simon and flanker congruency variables in the same object increases the impact of the irrelevant features in such a way that the Simon congruency effect reversed throughout the RT distribution.

## Towards an integrative explanation?

Taken together, previous research as well as the present results put forward the versatility of the interaction between Simon and flanker congruency, with some experiments showing an interaction, whereas other experiments showing no interaction. This emphasizes the necessity of an account in which the boundaries of when the interaction is assumed to occur and when it assumed to not occur are clearly formulated. Based on the additive factor logic [50, 51] according to which an interaction in RTs results from the fact that two variables influence the same processing stage, a starting point may be to assume that the interaction between Simon and flanker congruency should be observed when both Simon and flanker features are processed during the same stage. That is, the interaction should occur when there is an overlap in time (1) between the processing of Simon irrelevant feature and the processing of flanker irrelevant feature and (2) between the processing of these irrelevant features and the processing of the relevant target feature. Moreover, the overlap or the lack of thereof between these processes might not be fixed properties of a given paradigm (e.g., the Simon task or the flanker task). Rather, they seem to depend on the particular administration of the task combination and on the relevant and irrelevant stimulus features that the task combination involves. Thus, when the flanker and Simon

congruency variables are so administered that the flanker congruency effect increases as RTs increase but the Simon congruency effect decreases as RTs increase, the processing of both irrelevant and relevant features would not overlap in time, and both congruency variables should not interact. In contrast, when both congruency variables are so combined that the processing of Simon and flanker irrelevant features overlap in time with the processing of the relevant feature, the interaction should occur. Similar to Hommel [6], a temporal overlap is assumed in this account. However, there is at least one key difference. No spontaneous decay of the representation features is assumed. In this view, whereas the spontaneous decay of the representation features is still an option to explain the interaction between Simon and flanker congruency, the inhibition of irrelevant features (as derived in the conflict monitoring account) or the multiplicative priming of the correct response are also valid options.

Nevertheless, assuming an overlap in time when processing Simon and flanker features has two drawbacks. First, it is not really clear from the previous research as well as from the present study when an interaction should be expected to occur and when it should not be expected to occur. This makes difficult to formulate a priori the conditions under which there is an overlap in time resulting in an interaction, and the conditions under which there is no overlap in time and thus no interaction. Second, as mentioned in the introduction, the assumption that the temporal overlap in processing Simon and flanker irrelevant features results in an interaction in RTs is not directly compatible with the EEG findings [see 5, see also 23, 24]. The studies using EEG clearly showed an interaction in RTs but separate ERPs associated with the processing of the irrelevant features for both congruency variables. Nevertheless, the EEG evidence is so far limited as only three studies have been performed on paradigms combining different congruency variables [see 5, for a paradigm combining Simon and flanker congruency, 23, for a paradigm combining Stroop and flanker congruency, and 24, for a paradigm combining Stroop and semantic congruency]. This emphasizes the necessity of further research, in particular EEG research combining different congruency variables, in order to test for the robustness of these findings.

Although the EEG findings might be treated with caution due to the limited number of studies, we would like to mention that there is a way to integrate the EEG findings with the multiplicative priming account. This is possible if one assume that the flankers and the irrelevant location could activate their respective motor responses via direct associations already during early stimulus processing. This assumption is based on the model of selective attention and response selection put forward by Hübner and colleagues [52]. According to this model, in early stages, stimuli are processed by (incomplete) perceptual filtering. Thus, in addition to the pre-activation of the response associated to the relevant feature (i.e., the correct response), irrelevant features might still pre-activate their responses. Then, in a later stage, the selected feature (ideally the relevant feature) drives response selection. Thus, following this model and in line with the EEG results, one could assume in case of the interaction between flanker and Simon congruency variables that the flanker irrelevant and relevant features are first processed, pre-activating the correct response when trials are flanker congruent. Then, the Simon irrelevant and relevant features are processed, pre-activating the correct response when trials are Simon congruent. These pre-activations of motor responses then result in a co-activation of the correct response at the stage of response selection, which affects the duration of response selection or response execution, and in consequence performance. An illustration of this explanation is given in Fig 10.

It should be noted that according to this explanation, there is still an overlap in time. However, this temporal overlap is not between the processing of the relevant and irrelevant features, such as described above or proposed by Hommel [6]. Rather, it is between the effects of processing the irrelevant features. That is, the interaction between Simon and flanker congruency occurs only if *the effect of processing one irrelevant feature* (i.e., the response priming mediated

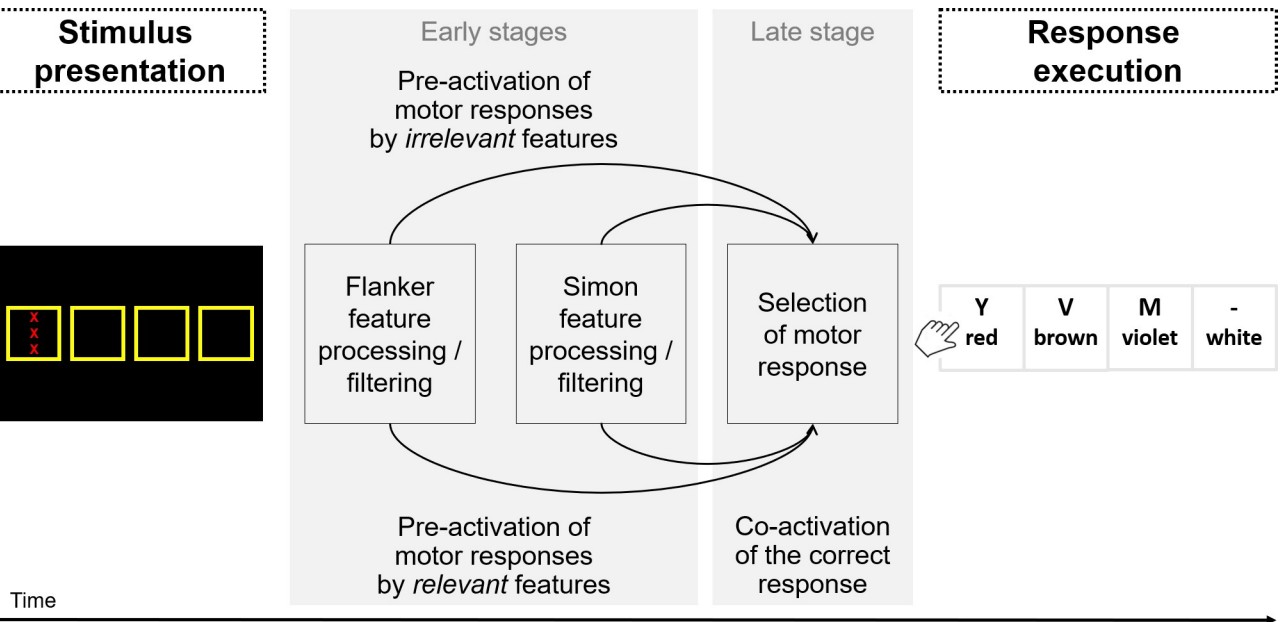

**Fig 10. An illustration of how multiplicative priming can be integrated with the different temporal conflict processing observed in the Simon and flanker tasks.** The figure presents the case when the trials are both flanker and Simon congruent. Thus, the flanker features are first processed, pre-activating the correct response. Then, the Simon features are processed, pre-activating the correct response again. As a consequence, at the stage of response selection, the correct response is co-activated, facilitating its selection. This impacts the duration of response selection or response execution, and accordingly performance.

by one irrelevant feature) overlaps in time with *the effect of processing the other irrelevant feature* (i.e. the response priming mediated by the other irrelevant feature). This assumption is also compatible with the EEG findings showing separate ERPs associated with the processing of the irrelevant features for both Simon and flanker congruency variables [5, 23, 24]. The reason is that the lack of a temporal overlap between the two ERP components reflecting the processing of relevant and irrelevant features is totally consistent with a temporal overlap in the effects of processing these features: One feature can be processed and its effect–that is, the response priming mediated by the processing of this feature–can last over time.

## Beyond response conflict

In the present study, we focused on the impact of response conflict. The reason is that incongruent trials are mainly assumed to induce a response conflict (in particular under no task-switching conditions, such as in the present study) [see 53]. Accordingly, in both Experiments 1a and 1b, we defined a trial as neutral when it includes only one response-relevant feature. Nevertheless, previous research has put forward that neutral stimuli, in particular neutral flanker trials, share similarities with the stimulus set, thus still involving a conflict but at the stimulus level [e.g., 26, 27]. For example, it is possible that the neutral color dark pink is associated for example to red, thus triggering a stimulus conflict. This possibility requires further research in which stimulus and response conflicts are disentangled either experimentally [e.g., 4] or using a computational modelling approach [see, e.g., 53].

## Conclusion

To summarize, the findings of the present study show that the interaction between Simon and flanker congruency is affected neither by a spatial feature overlap between congruency

variables nor by the learning of stimulus-response pairings. Rather, they are rather in line with the view that this interaction is affected by the multiplicative priming occurring when all irrelevant features co-activate the correct response.

## Supporting information

**S1 File. Performance on the pure Simon and flanker blocks.**
(PDF)

**S2 File. Delta plots.**
(PDF)

## Acknowledgments

The authors thank Rebekka Eilers, Eva-Maria Hartmann, and Maria Huber for their help in data collection.

## Author Contributions

**Conceptualization:** Alodie Rey-Mermet, Miriam Gade, Marco Steinhauser.

**Data curation:** Alodie Rey-Mermet.

**Formal analysis:** Alodie Rey-Mermet.

**Funding acquisition:** Alodie Rey-Mermet, Marco Steinhauser.

**Investigation:** Alodie Rey-Mermet.

**Methodology:** Alodie Rey-Mermet, Miriam Gade, Marco Steinhauser.

**Project administration:** Alodie Rey-Mermet.

**Resources:** Alodie Rey-Mermet, Marco Steinhauser.

**Software:** Alodie Rey-Mermet.

**Supervision:** Alodie Rey-Mermet.

**Validation:** Alodie Rey-Mermet.

**Visualization:** Alodie Rey-Mermet, Marco Steinhauser.

**Writing – original draft:** Alodie Rey-Mermet.

**Writing – review & editing:** Alodie Rey-Mermet, Miriam Gade, Marco Steinhauser.

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
