## [Decision Letter · Decision Letter 0]

28 Jul 2020

PONE-D-20-15659

Multiplicative priming of the correct response can explain the interaction between Simon and flanker congruency

PLOS ONE

Dear Dr. Rey-Mermet,

Thank you for submitting your manuscript to PLOS ONE. I have received two expert reviews and am ready to act. The reviews are excellent in both their thoroughness and thoughtfulness (thanks to both reviewers). As you will see below, the reviewers have raised some issues and I think your study can provide conclusive answers to the issues raised with a major revision. I will not spend time summarizing the reviews in any detail, you can see the detailed reviews below. Given these reviews, I invite you to submit a revised version of the manuscript that addresses the points raised during the review process. Please pay attention to the following comments and give them due consideration, as those changes are required for acceptance. I will send back the revision to both reviewers.

I look forward to receiving your revised manuscript.

Kind regards,

Ludovic Ferrand, Ph.D.

Academic Editor

PLOS ONE

Journal Requirements:

"The study was conducted in accordance with the guidelines of the ethics committee of the Catholic University of Eichstätt-Ingolstadt (approval number: 2016/18), and written informed consent was acquired from all participants.".

i) Please amend your current ethics statement to confirm that your named institutional review board or ethics committee specifically approved this study. 

ii) *Once you have amended this/these statement(s) in the Methods section of the manuscript, please add the same text to the “Ethics Statement” field of the submission form (via “Edit Submission”).*

* For additional information about PLOS ONE ethical requirements for human subjects research, please refer to *http://journals.plos.org/plosone/s/submission-guidelines#loc-human-subjects-research*. *

Reviewers' comments:

Reviewer's Responses to Questions

**Comments to the Author**

1. Is the manuscript technically sound, and do the data support the conclusions?

Reviewer #1: Partly

Reviewer #2: Partly

2. Has the statistical analysis been performed appropriately and rigorously? 

Reviewer #1: Yes

Reviewer #2: Yes

3. Have the authors made all data underlying the findings in their manuscript fully available?

Reviewer #1: Yes

Reviewer #2: Yes

4. Is the manuscript presented in an intelligible fashion and written in standard English?

Reviewer #1: Yes

Reviewer #2: Yes

5. Review Comments to the Author

Reviewer #1: The present work aims for a better understanding of the origin of the under-additive interaction between Simon and flanker congruency effect evidenced by previous work. Experiment 1 tests whether the interaction results from a spatial feature overlap between tasks by orthogonally manipulating the spatial configuration between both tasks (Simon horizontal/flanker vertical and Simon vertical/flanker horizontal). A robust interaction between Simon and flanker congruency effects was observed in the two groups, ruling out the spatial overlap hypothesis. Experiment 2 tests whether the interaction stems from a bias in the learning of the stimulus-response mappings. The interaction was still observed when equating the number of stimulus exemplars across all trial types, ruling out this hypothesis. Finally, Experiments 3a and 3b evaluate the hypothesis of a multiplicative priming account. In congruent trials, the correct response is activated by the irrelevant stimulus attribute of each task (the position on the screen and the flankers), which may result in multiplicative priming of the correct response (each irrelevant feature would increase response activation proportional to its activation, which would accelerate processing in congruent-congruent trials). This hypothesis was evaluated by incorporating neutral Simon and flanker trials. If the under-additive interaction results from multiplicative priming induced when the trial is both Simon and flanker congruent, the interaction should be observed when the facilitation effect (neutral minus congruent) is used as the dependent variable, but not when the interference effect (incongruent minus neutral) is used as the dependent variable. According to the authors, the results provide evidence for the multiplicative priming account.

The paper addresses a complex and interesting topic in the field of cognitive psychology. It is well written, and the experiments are rigorously conducted. I enjoyed the thorough theoretical approach to the problem, and found the results from Experiments 1 and 2 compelling. However, I believe that some important methodological and theoretical issues regarding the evaluation of the multiplicative priming account should be addressed before the paper is published. See attached file.

Reviewer #2: Results of this study are interesting but the way in which they are presented is not convincing. There are many logic inconsistencies and the rationale of the experiments is weak: the proposed accounts cannot explain the critical interaction and/or the proposed experiment is not appropriate to evaluate the account.

I suggest starting from the multiplicative priming account, which appears to be the most sensible, sound and grounded account based on previous studies, and presenting feature-overlap and frequency of S-R parings as factors they may have an impact on the interaction, i.e., they may be necessary for the multiplicative priming and, consequently, the interaction to occur (feature-overlap) or they may either act in favour or counteract the under-additive interaction predicted by the multiplicative priming account.

The study may be presented as an attempt to evaluate the impact of these factors.

MAJOR POINTS

1) Other accounts of the interaction (conflict-monitoring and spatial-code decay).

p.8. lines 117-125. I do not fully understand why the results of the ERP study described here rule out the conflict-monitor and spatial-code decay accounts. Please, give a thorough and clear explanation for that. Beyond this, the results of ONE experiment are not enough to rule out an account - especially because:

(a) as I understand it, the critical results challenging that account are ONLY the ERP data of that study ;

(b) the conflict-monitoring account is presented in the paper as one of the main accounts for the interaction.

I do not actually think that the conflict-monitoring account is the main account for the interaction. In my view it is a sensible explanation for sequential effects, but it is not a sound explanation for this type of interaction (see my second minor point).

In any case, among the studies that the authors mention, there may be other studies the results of which challenge these two accounts (conflict-monitor and spatial-code decay). For example, did other studies perform distributional analyses? Did other studies find additive effects instead of interactive effects? Did other studies find other types of interactions? Are these additive effects/other types of interactions accountable for by these two hypotheses?

From what I understand, the results of the present study do not rule out these alternative accounts (both accounts are compatible with the results), thus the authors should present convincing evidence that the multiplicative priming account is the most suitable explanation for the interaction.

2) On p. 8, the authors claim that “although flanker conflict and Simon conflict are resolved at separable stages, there appears to be little influence of the earlier stage on the later one.”

What are these different processing stages? That the flanker and Simon effects arise at different processing stage is indeed inconsistent with:

(a) the conclusions of the present study.

According to the additive factor logic, the presence of an interaction suggests that two effects arise at the same stage, or involve at least one processing stage in common (cf. Sternberg, 1969, 1984). Moreover, if, as suggested by the authors, the interaction is due to priming of the same vs. different response(s) by the two irrelevant features, then we can conclude that both effects emerge at the response-selection stage.

(b) main accounts of the flanker and Simon effects, according to which they both emerge at the response selection stage.

There are plenty of evidence suggesting that the locus of both effects is that stage (cf., e.g., literature reviews by Proctor and colleagues, Eriksen and colleagues, experimental studies like that by Hübnerand Töbel, 2019) and I think that the authors should explicitly state it. They should also underline that, given that both effects arise at the same processing sage, one may expect an interaction between the two compatibility factors.

3) The feature overlap account is NOT an account for the interaction, but only a possible account for the discrepancies between studies: according to this account, the interaction can only emerge when there is an overlap between the irrelevant spatial feature producing the Simon effect and a spatial feature of the flanker object(s), thus this overlap is a prerequisite for the interaction (and this would explain why in some studies no interaction emerged while it emerged in others), but it does not specify which is the mechanism producing the interaction. Accordingly, feature overlap should not be presented as an account for the interaction, but as a possible factor influencing it.

It is worth noticing that a paper mentioned by the authors on p. 5 and p. 45 (Treccani et al., 2012) proposes that the interaction only emerges when there is massive overlap between the two irrelevant features (i.e., they are borne by the very same object) and participants process such an overlap (i.e., they “bind” the two features). Thus the “feature binding” is the prerequisite. Yet, the mechanism proposed to account for the interaction is multiplicative priming.

4) pp. 9-11. The account based on the “learning of stimulus-response mappings” does not actually account for the critical interaction, at least as it is presented (see REASONS** below).

The authors may consider presenting the frequency of presentation of the different S-R pairings simply as a factor that may influence the performance in this kind of tasks (i.e., tasks combining more than one compatibility factor). They may present Experiment 2 as an experiment aimed to investigate the possible impact of this factor and its possible contribution to the interaction (together with OTHER factors) - in principle, at least, this factor may actually counteract, rather than act in favour of the interaction (see below).

If they decide to do so, they should specify in which previous studies this factor may have had an effect, that is, in which previous studies using flanker-Simon paradigms the S-R pairings were presented with a different frequency. Indeed, the rationale of Experiment 2 would become even weaker if in other previous studies the frequencies of different S-R parings were balanced. To make a significant contribution to the understanding of the factors influencing performance in this kind of tasks, I strongly suggest comparing (by means of an ANOVA) the results of Experiment 2 with that of the similar condition of Experiment 1 in which, however, frequencies were not balanced. In this way, the authors may evaluate the specific contribution of this factor.

**REASONS WHY THIS ACCOUNT DOES NOT WORK:

(a) The authors claim that “the stimulus-response mappings for the congruent-congruent trials are the easiest to learn and then to retrieve because there are only two mappings”

This does not make any sense to me : the congruent-congruent trials are not presented in isolation but with the other trials, thus each of these two (I would say) S-R “doubly” congruent pairings are simply one of 8 possible S-R pairings. They cannot be easier to learn or retrieve.

(b) The authors probably meant something else: They probably meant that the numbers of times with which the S-R pairings are presented can explain the interaction (i.e., some S-R pairings may be practiced more). However, I do not understand how.

- When the stimulus irrelevant features are two (e.g., two positions, two flanker identities), and the study provides for the same number of trials in all the four possible categories (a. flanker congruent-Simon congruent; b. flanker incongruent-Simon incongruent, c. flanker congruent-Simon incongruent; d. flanker incongruent-Simon congruent), the number of S-R pairings for each category is exactly the same (i.e, two S-R pairings for each category), thus one would not expect any interaction. To produce the interaction, a study should provide for a larger number of S-R pairings for the categories a and b than for the categories c and d. For example, a study that considers only three categories: totally congruent trials, totally incongruent trials, partially (either flanker or Simon) incongruent trials. This last category would include more S-R pairings (i.e, 4 exemplars), that would be presented less frequently To the best of my knowledge, a similar study has not been conducted, thus this is not an account for the interaction for any two-feature study (If –instead- I am wrong, the authors should explicitly mention this study and explain that in this specific case the interaction may emerge from the addition of two compatibilities effect and the effect of the number of exemplars/frequency)

- When the stimulus irrelevant features are four, and the study provides for the same number of trials in all the four possible categories, the number of S-R pairings for each category is as proposed by the authors: 4 exemplars of the category a (congruent congruent )< 12 exemplars of category c= 12 exemplars of category d< 36 exemplars of category b . The effect of this should be: RT (and/or errors) of a<c=d<b, which="">There is no apparent reason why there should be advantage for the 4-exemplars “a” category over the 12 -exemplars categories, while there should not be an advantage of the 12-exemplars categories (“c” or “d”) over the 36-exemplars category. Quite the opposite, these different frequencies may counteract a possible under-additive interaction produced by other factors (e.g., multiplicative priming). The comparison I suggested (see above) may help to understand the real impact of this factor.

(c) The authors also state that “Explaining performance on this trial type may be sufficient to account for the interaction …The reason is that although previous research has focused on the reduction of the congruency effect of one task when the other task was incongruent, the interaction between Simon and flanker congruency mainly resulted from an increase in the congruency effect of one task when the other task was congruent, in particular from the fast RTs and high rates of correct responses in the congruent-congruent trials”

As explained above, this pattern of performance cannot be explained by the different frequency of S-R pairings given that, based on that, one should expect larger effects of a compatibility factor in both levels of the other factor.

Moreover, I do not think that the interaction mainly results from an increase of the effect of a congruency factor in the congruent level of the other factor: in some studies a congruency effect was found in the congruent level of the other factor, while this effect was no longer significant in the incongruent level of the other factor (thus, a reduction/disappearance of the effect was observed in the other-factor incongruent conditions compared to standard conditions). The authors themselves found a disappearance of the effect of a compatibility factor in the incongruent level of the other factor. E.g.,: “Similarly, the Simon congruency effect was smaller – but not significant – when the trials were flanker incongruent”

5) The rationale of Experiments 3a and 3b is not clear. Do these two experiments simply aim to verify a prediction of the multiplicative priming account? The authors claim that “the interaction should not be found when the interference (i.e., the difference between incongruent and neutral trials) was used as dependent measure”. What are the predictions based on other accounts? Could we expect other outcomes based on the other factors mentioned by the authors?

I have other major concerns, but they might no longer be relevant if the authors decide to change the manuscript according to the comments and suggestions I have detailed above.

MINOR POINTS

1) From the manuscript: “Simon Task, flanker task, and their interaction” “Previous research has shown an interaction between the Simon and flanker tasks when both tasks were combined within the same trial”.

It is worth noticing that the interaction is not between two tasks. Indeed, the task is only one: participants had to indicate the colour of the central letter while ignoring the flanking letters and their location on the screen. This task had the characteristics of both typical Simon and flanker tasks.

I would say that the interaction is between the flanker-target identity congruence (the compatibility factor usually observed in flanker tasks) and the stimulus-response spatial correspondence (the compatibility factor usually observed in flanker tasks).

Speaking about two tasks may be confusing to the reader (one may think of a dual-task paradigm) and makes it harder to understand the proposed accounts for the interaction.

For example, that authors claim that “One can assume that the conflict monitoring system estimates the current levels of conflict for the task being processed first. If this task is incongruent and thus associated with a high degree of conflict, this leads to a shift of control signal. That is, response-relevant features are activated while all irrelevant features – including those for the task being processed second – are inhibited. Thus, the strength of control is also adjusted for the second task to be processed.. “

What are the tasks the authors are speaking about?

I guess they are speaking about irrelevant features, instead of tasks, which are not to be processed (given that they are task-irrelevant), but they are processed nevertheless.

I would rather write: “One can assume that the conflict monitoring system estimates the current levels of conflict. If the first irrelevant feature being processes (e.g., stimulus position) is incongruent with the relevant one, and thus associated with a high degree of conflict, response-relevant features are activated while all irrelevant features (not only that already processed) are inhibited. “

2) Also, what is exactly inhibited according to the conflict-monitor account? Following some versions of this account (e.g., the “gate” version), when a conflict between competing response codes is detected by the conflict monitoring mechanism, the weight of the route activated by the irrelevant feature(s) is attenuated. Do the different versions of the conflict-monitoring account have different predictions?

Actually, as underlined above, I find this account not very convincing. If the gate closes, it should close in the first place for the direct route involving the feature for which a conflict is detected. Moreover, it should take a while for the gate to close, thus the effect of this closure should be observed in the following trial

3) p.8 " Hommel [6] (see Table 1), suggesting that this effect could explain the observed interaction in some experiments but not in others".

What effect? Please be more explicit “suggesting that this phenomenon(i.e., the decay of the stimulus spatial code) could explain

4) Please fix inconsistencies in lexical choices. “Stimulus-response mapping” (e.g., p.9) usually means the instructed (i.e., contained in the instructions) associations between specific stimulus values and the correct responses(i.e., the responses required by these values). I think that the authors should use instead “S-R pairings” (e.g., S-R congruent pairings). Note that the authors themselves use “stimulus-response mapping” with the standard meaning on p. 14.

p. 14. “The stimulus was determined randomly for each trial” Maybe better (more clear) “the stimulus colour was…”.

5) p. 16. In contrast with the horizontal-key condition, the keys in the vertical-key condition are not perfectly aligned along one spatial axis, the vertical one (i.e., they are not perfectly one above the other). Can the authors justify this choice? (in many keyboards, there are numerical keys which are vertically aligned).

6) pp. 14-16. I did not understand whether stimuli were of different size in the different conditions (it is not clear from the description) and whether in the flanker pure blocks the asterisks appeared in a (central) yellow square as in the pure Simon blocks. These seem unnecessary differences between experimental conditions.</c=d<b,>

6. PLOS authors have the option to publish the peer review history of their article (what does this mean?). If published, this will include your full peer review and any attached files.

Reviewer #1: No

Reviewer #2: No

---

## [Author Response · Author response to Decision Letter 0]

28 Oct 2020

Editor

E1 “Please ensure that your manuscript meets PLOS ONE's style requirements, including those for file naming.”

RESPONSE: We checked our manuscript so that it meets PLOS ONE’s style requirements.

E2 “Thank you for including your ethics statement:

"The study was conducted in accordance with the guidelines of the ethics committee of the Catholic University of Eichstätt-Ingolstadt (approval number: 2016/18), and written informed consent was acquired from all participants.".

a) Please amend your current ethics statement to confirm that your named institutional review board or ethics committee specifically approved this study. 

b) Once you have amended this/these statement(s) in the Methods section of the manuscript, please add the same text to the “Ethics Statement” field of the submission form (via “Edit Submission”)”

RESPONSE: In the revised manuscript, we rewrote the ethics statement as follows (see p. 18): “The study was approved by the ethics committee of the Catholic University of Eichstätt-Ingolstadt (approval number: 2016/18), and written informed consent was acquired from all participants”. We also added this text to the “Ethics Statement” field of the submission form.

Reviewer A

A1 “Evaluating the multiplicative priming account requires a valid neutral condition for both Simon and flanker tasks. However, finding an appropriate neutral condition in these tasks has proven difficult, and has been the object of several studies. I was disappointed that these studies were not carefully reviewed in the present work, and some methodological features of Experiments 3a and 3b are potentially problematic.

In the flanker task, the authors use flanker stimuli that are neutral with respect to the response set. The issue here is that these neutral flanker stimuli can still share similarities with members of the stimulus set. This issue has been thoroughly investigated by Charles Eriksen and his colleagues (e.g., Yeh & Eriksen, 1984; Eriksen & St. James, 1986). Specifically, the authors used the colors red, brown, violet, and white as part of the stimulus set, and the colors dark pink or green for their neutral flankers. What evidence do we have that these ‘neutral’ colors do not share any similarity with members of the stimulus set? Dark pink could prime the response associated with violet (or red). 

For the Simon task, the authors used two types of neutral trials: stimuli presented at the center of the screen (Exp 3a) and stimuli presented above and below the center of the screen (Exp 3b). Presenting stimuli in the center of the screen gives a processing advantage compared to stimuli presented laterally (because the attentional focus does not need to be shifted), which is what happened in Exp 3a, resulting in a reversed facilitation effect. Exp 3b does not fully control for this issue because the spatial distance between neutral stimuli and the center of the screen is twice smaller than the spatial distance between the center of the screen and the outer lateral stimuli. In this respect, what was the rationale for using four stimulus locations (outer left, inner left, inner right, outer right) in the experiments? This design makes it impossible to have neutral and congruent/incongruent Simon trials presented at equidistance from the screen center.”

RESPONSE: In the present study, in particular in Experiment 1 (previously Experiment 3), we aimed to focus on the impact of response conflict. The reason is that incongruent trials are mainly assumed to induce a response conflict (in particular under no task-switching conditions, such as in the present study; Steinhauser & Hübner, 2009). Accordingly, we defined a trial as neutral when it includes only one response-relevant feature, such as flanker neutral trials in which the flankers were presented in color associated to none of the responses. Nevertheless, we agree with Reviewer A that the impact of similarity in stimulus set and thus the resulting stimulus conflict might be the topic of future work. This is acknowledged in the manuscript (see p. 52-53). 

In the present study, we opted for a Simon condition with four stimulus locations because the research using a large stimulus set has resulted in mixed evidence (see Table 1). Moreover, in Experiment 1b (previously Experiment 3b), we did not select a Simon condition with four neutral locations (i.e., outer above, inner above, inner below, outer below) because this would have resulted in an unbalanced number of stimulus-response pairings for Simon neutral trials in comparison to flanker neutral trials (2 stimulus-response pairings). Thus, with our design, we avoided stimulus-response pairings of flanker neutral trials to be presented more frequently and to be learned more efficiently because of their smaller number. This is now clarified in the manuscript (see p. 16). 

A2 “The theoretical foundations of the multiplicative priming hypothesis appear a bit weak. I do not see why facilitation effects in Simon and flanker tasks would interact. The authors mention the Treccani et al (2008) study as support for this multiplicative priming account, but I do not understand why. More problematic, the multiplicative priming hypothesis requires that the irrelevant stimulus attribute of each task generates facilitation in congruent trials. In the Simon task, although the bulk of studies generally found facilitation effects, some studies failed to observe facilitation, and attributed the Simon effect to interference only (Craft & Simon, 1970; Simon & Small, 1969). The picture is even less clear in the flanker task, where many studies incorporating neutral trials failed to find facilitation effects (e.g., Wild Wald et al., 2008; de Bruin & Della Salla, 2018), even when neutral flankers shared feature similarities with the target (Eriksen & Eriksen, 1974; Yeh & Eriksen, 1984; Eriksen & St. James, 1986). This literature must be considered by the authors”

RESPONSE: In the manuscript (see p. 8-9), we have now rewritten the paragraph explaining how Treccani et al. (2008) support the multiplicative priming account: “First empirical support for such a priming of irrelevant features was provided by Treccani and colleagues (2009). In their study, both Simon and flanker conditions were so combined that a colored square was presented in the middle of the screen, which was flanked by one colored square either to the left or to the right (see Fig 1). Thus, the flanker square conveyed the irrelevant flanker color for the flanker condition. At the same time, it also conveyed the irrelevant location for the Simon condition because it was assumed to trigger a location congruent or incongruent to the response location. Thus, in congruent-congruent trials, the irrelevant square triggers the correct response twice (i.e., by means of its color and its position; see the top-left panel of Fig 1), thus facilitating performance. Moreover, in this particular design, the irrelevant square also triggers the incorrect response in incongruent-incongruent trials twice (again by means of its color and its position; see the bottom-right panel of Fig 1). This allows a faster rejection of this response. Together, this accounts for the interaction between Simon and flanker congruency by explaining the faster and more correct response in congruent-congruent trials and incongruent-incongruent trials compared to the corresponding trials mixing incongruent and congruent features.” 

Moreover, we thank Reviewer A for putting our attention to the literature questioning whether irrelevant stimulus feature can generate facilitation in congruent trials. We have now integrated them in the following way (see p. 17): “However, the multiplicative priming account is based on the assumption that the irrelevant stimulus feature for each Simon and flanker condition co-activate the correct response in congruent trials. As there has been some debate in previous research whether this occurs (de Bruin & Sala, 2018; B. A. Eriksen & Eriksen, 1974; C. W. Eriksen & St. James, 1986; Simon & Craft, 1970; Simon & Small, 1969; Wild-Wall et al., 2008; Yeh & Eriksen, 1984), it is possible that both the irrelevant location and flanking letters do not co-activate the correct response. In this case, the interaction should not be observed in the facilitation effect. This implies that the interaction should be observed in the interference effect. The reason is that as the congruency effect is computed as the addition of the interference effect and the facilitation effect, finding an interaction in the congruency effect but not in the facilitation effect should result in finding an interaction in the interference effect.”

A3 “Another theoretical issue associated with the multiplicative priming hypothesis concerns the relative time-course of automatic response activations in Simon and flanker tasks. A large body of empirical and computational evidence (see Ulrich’s diffusion model for conflict tasks) that the location-based response activation in the Simon task is much faster than the flanker-based response activation. One may thus wonder how multiplicative priming can occur in this context, because the location-based response activation would have almost fully decayed before the flanker-based activation reaches the response activation stage. The authors provide a tentative explanation in the discussion section but I do not understand it. Could you please clarify?”

RESPONSE: In addition to mentioning Ulrich’s diffusion modeling approach (Ellinghaus et al., 2018; Ulrich et al., 2015), we clarified our explanation by illustrating it with a new figure (i.e., see p. 50-51 and Fig 10).

A4 “I noticed substantial variations in the evidence in favor of the interaction between Simon and flanker congruency effects on mean RTs between the first three experiments (Exp1: BF10 = 3.86; Exp2: 19.38; Exp 3a: 1.65). More problematic, Exp 3b reported weak evidence against the interaction model, though an interaction was found on error rates. These variations across experiments are puzzling in light of the minor methodological changes between experiments, and require some explanations. One possibility is that the experiments are underpowered. The authors performed a power analysis and concluded that the minimum sample size to observe the interaction of interest was 9. I do not understand how the minimum sample size for a mixed design ANOVA could be 9 given an effect size of η_p^2= .42 and a power of .99. Could you please clarify and specify the value of each parameter in G*Power? In addition, could you please clarify what this minimum sample size represent (minimum sample size for the whole experiment versus minimal sample size for each group)? Finally, the effect sizes for the interaction reported in the present study appear much smaller than the benchmark .42 partial eta-square value, though the use of different effect size statistics (η_p^2 versus η_g^2) prevents a direct comparison”

RESPONSE: The variation in Bayes factors (BF) across the experiments might be caused by differences in the number of participants, the number of trials, and the number of repetition of stimulus-response pairings per trial type. 

Experiment 1a (previously 3a):

- Number of participants: 29

- Number of trials in the mixed blocks per participant and trial type: 160

- Number of repetition of stimulus-response pairings per participant and per trial type:

 - Congruent-congruent trials: 40

 - Trials mixing incongruent and congruent stimuli: 13-14

 - Trials mixing incongruent and neutral stimuli: 53-54

 - Trials mixing neutral and congruent stimuli: 40

 - Neutral-neutral trials: 20

 - Incongruent-incongruent trials: 4-5

Experiment 1b (previously 3b):

- Number of participants: 30

- Number of trials in the mixed blocks per participant and trial type: 160

- Number of repetition of stimulus-response pairings per participant and per trial type:

 - Congruent-congruent trials: 40

 - Trials mixing incongruent and congruent stimuli: 13-14

 - Trials mixing incongruent and neutral stimuli: 26-27

 - Trials mixing neutral and congruent stimuli: 20

 - Neutral-neutral trials: 10

 - Incongruent-incongruent trials: 4-5

Experiment 2 (previously 1):

- Number of participants: 48 (i.e., 24 per group)

- Number of trials in the mixed blocks per participant and trial type: 216

- Number of repetition of stimulus-response pairings per participant and per trial type:

 - Congruent-congruent trials: 54

 - Trials mixing incongruent and congruent stimuli: 18

 - Incongruent-incongruent trials: 6

Experiment 3 (previously 2):

- Number of participants: 36

- Number of trials in the mixed blocks per participant and trial type: 216

- Number of repetition of stimulus-response pairings per participant and per trial type: 54

Thus, the larger BF observed in Experiment 3 (previously Experiment 2) might result from a certain consistency observed in the RT data due to the larger sample size and the large number of repetitions of each stimulus-response pairing for each trial type. 

Regarding the BF = 21.53 observed in error rates for the interaction between Simon and flanker conditions in the Experiment 1b (previously Experiment 3b), this seems to be associated to a slightly smaller range in the mean error rates in this experiment (i.e., CIs = 5%-8% compared to 5%-9% in Experiment 1a, for example). At the same time, the interaction – represented as the difference in the flanker congruency effect between Simon incongruent and congruent trials – is slightly larger in Experiment 1b (i.e., 3% in comparison to 2% in Experiment 1a). It is possible that both patterns – that is, smaller error-rates range but larger difference – explains this BF. 

In addition, we would like to thank Reviewer A for pointing out our error regarding the power analysis. We removed the power analysis because we misread G*Power requirements, which seems to occur very often (see Brysbaert, 2019). We also refrained us to compute a post-hoc power analysis because this is not appropriate (see http://daniellakens.blogspot.com/2014/12/observed-power-and-what-to-do-if-your.html). Moreover, if our experiments are underpowered, this means that all previous studies investigating the interaction between Simon and flanker conditions were underpowered. The reason is that we selected our minimal sample size (i.e., 24 participants in each group in which the interaction between Simon and flanker conditions was expected to occur) because it belongs the largest sample sizes tested in previous research (see Table 1).

MINOR POINTS

1. “The authors consider the possibility of multiple priming of the correct response in congruent-congruent trials. One may wonder if multiple priming of the incorrect response can occur as well, and if this phenomenon could contribute to explain the underadditive interaction.”

RESPONSE: Yes, the multiplicative priming was also used to explain performance in incongruent-incongruent trials. This is now clarified in the manuscript (see p. 8-9): “Moreover, in [the] particular design [used by Treccani and colleagues (2009)], the irrelevant square also triggers the incorrect response in incongruent-incongruent trials twice (again by means of its color and its position; see the bottom-right panel of Fig 1). This allows a faster rejection of this response. Together, this accounts for the interaction between Simon and flanker congruency by explaining the faster and more correct response in congruent-congruent trials and incongruent-incongruent trials compared to the corresponding trials mixing incongruent and congruent features.” 

2. “Table S1a: the authors report non-significant Simon and flanker effects for the pure Simon and flanker blocks. I hope it is simply a typo…”

RESPONSE: Yes, it was a typo. Thank you for pointing out this error. This has been changed (see Table S1c in the S1 file on p. 4-5).

3. “Analyses for the pure blocks of Experiment 2 are missing.”

RESPONSE: These analyses for Experiment 3 (previously Experiment 2) are presented in Table S1d in the S1file (see p. 6 in the S1 file). 

4. “Which error term was used for the follow-up t-tests analyses?”

RESPONSE: We used the pooled estimate for the error term.

5. “The authors should provide the mapping between each key and the corresponding hand/finger.”

RESPONSE: This has been now added (see p. 21 and 37). 

6. “It looks like the color-response mapping was fixed across participants for each experiment. What was the motivation for fixing this mapping?”

RESPONSE: We followed previous research (Akçay & Hazeltine, 2011; Rey-Mermet & Gade, 2016; Stoffels & Molen, 1988).

7. “Please check the syntax of the sentence lines 617-620.”

RESPONSE: This has been modified (see p. 24): “In the present experiments, the most relevant results are the interaction between both congruency variable, that is, the modulation of the congruency effect (i.e., incongruent vs. congruent trials), the interference effect (i.e., incongruent vs. neutral trials), and the facilitation effect (i.e., neutral vs. congruent trials) of one condition by the congruency of the other condition.” 

8. “I feel like the hypothesis of a bias in the learning of the stimulus-response mappings (and the way the authors tested for this hypothesis in Experiment 2) will be hard to understand for somebody who is not expert in the field. I may be useful to detail this hypothesis a bit more, perhaps with an illustration. This illustration could also include the other hypotheses (spatial overlap and multiplicative priming).”

RESPONSE: As suggested by Reviewer A, we added figures to explain the learning of stimulus-response pairings (see Fig 2) and the multiplicative priming effect (see Figs 1 and 3). 

Reviewer B

B1 “Results of this study are interesting but the way in which they are presented is not convincing. There are many logic inconsistencies and the rationale of the experiments is weak: the proposed accounts cannot explain the critical interaction and/or the proposed experiment is not appropriate to evaluate the account.

I suggest starting from the multiplicative priming account, which appears to be the most sensible, sound and grounded account based on previous studies, and presenting feature-overlap and frequency of S-R parings as factors they may have an impact on the interaction, i.e., they may be necessary for the multiplicative priming and, consequently, the interaction to occur (feature-overlap) or they may either act in favour or counteract the under-additive interaction predicted by the multiplicative priming account.

The study may be presented as an attempt to evaluate the impact of these factors.”

RESPONSE: We followed Reviewer’s B suggestion, and we re-organized the manuscript accordingly. Thus, Experiments 3a and 3b are now Experiments 1a and 1b, respectively, and Experiments 1 and 2 are now Experiments 2 and 3, respectively. We also present all these experiments as testing whether the interaction between Simon and flanker congruency is affected by the multiplicative priming of the correct response, the spatial overlap between Simon and flanker features, and the bias in the learning of stimulus-response pairings (see p. 2, 14-15, 35-36, 41-42, and 47-48). 

B2.1 “Other accounts of the interaction (conflict-monitoring and spatial-code decay).

p.8. lines 117-125. I do not fully understand why the results of the ERP study described here rule out the conflict-monitor and spatial-code decay accounts. Please, give a thorough and clear explanation for that.“

RESPONSE: We now explained in more details the results of the ERP study and how they challenge both accounts (see p. 10): “Second, the central assumption put forward in both accounts – that is, resolving the first conflict affects the resolution of the second conflict – has been put into question by a study using event-related potentials (ERPs; Frühholz et al., 2011; see also Rey-Mermet et al., 2019). In that study, ERPs were used to track the neural time course of conflict resolution. The results showed an interaction in RTs, but a sequential conflict resolution in the ERPs. That is, the resolution of the flanker conflict was associated with an early component (i.e., N2), whereas the resolution of the Simon conflict was associated with a later component (i.e., P3b). However, no interaction was observed in the ERPs, in particular on the later component. If the first conflict affected the resolution of the second conflict, this should have resulted in an interaction in the later component because this later component is associated to the resolution of the second conflict. In contrast, the results suggest that although flanker conflict and Simon conflict are resolved at separable stages, there appears to be little influence of the earlier stage on the later one.”

B2.2 “Beyond this, the results of ONE experiment are not enough to rule out an account - especially because:

(a) as I understand it, the critical results challenging that account are ONLY the ERP data of that study ;

(b) the conflict-monitoring account is presented in the paper as one of the main accounts for the interaction. I do not actually think that the conflict-monitoring account is the main account for the interaction. In my view it is a sensible explanation for sequential effects, but it is not a sound explanation for this type of interaction (see my second minor point).

In any case, among the studies that the authors mention, there may be other studies the results of which challenge these two accounts (conflict-monitor and spatial-code decay). For example, did other studies perform distributional analyses? Did other studies find additive effects instead of interactive effects? Did other studies find other types of interactions? Are these additive effects/other types of interactions accountable for by these two hypotheses?

From what I understand, the results of the present study do not rule out these alternative accounts (both accounts are compatible with the results), thus the authors should present convincing evidence that the multiplicative priming account is the most suitable explanation for the interaction.”

RESPONSE: Our goal was not to say that the within-trial conflict-monitoring and spatial-code decay accounts are ruled out. Rather, our goal was – for the sake of completeness – to report the accounts used in previous research to explain the interaction of this kind, and then to highlight the findings which are not line with these accounts. Moreover, in case of the multiplicative-priming account, our goal was to highlight the fact that this account seems very appropriate to explain the findings when all irrelevant features are conveyed by the same object, but it is unknown whether this is still the case when a more standard design is used (i.e., when irrelevant features are conveyed by different objects). 

Overall, we think that these accounts can so far explain some but not all empirical results. Accordingly, more than one process could account for the under-additive interaction between flanker and Simon congruency. This is now clarified in the manuscript (p. 9-11).

B3 “On p. 8, the authors claim that “although flanker conflict and Simon conflict are resolved at separable stages, there appears to be little influence of the earlier stage on the later one.”

What are these different processing stages? That the flanker and Simon effects arise at different processing stage is indeed inconsistent with:

(a) the conclusions of the present study.

According to the additive factor logic, the presence of an interaction suggests that two effects arise at the same stage, or involve at least one processing stage in common (cf. Sternberg, 1969, 1984). Moreover, if, as suggested by the authors, the interaction is due to priming of the same vs. different response(s) by the two irrelevant features, then we can conclude that both effects emerge at the response-selection stage.

(b) main accounts of the flanker and Simon effects, according to which they both emerge at the response selection stage.

There are plenty of evidence suggesting that the locus of both effects is that stage (cf., e.g., literature reviews by Proctor and colleagues, Eriksen and colleagues, experimental studies like that by Hübnerand Töbel, 2019) and I think that the authors should explicitly state it. They should also underline that, given that both effects arise at the same processing sage, one may expect an interaction between the two compatibility factors.” 

RESPONSE: Reviewer B exactly points out the issue in previous research. According to the additive factor logic, the interaction in RTs observed between Simon and flanker congruency should occur because two variables influence the same processing stage (most probably the response selection or execution stage; see Steinhauser & Hübner, 2009). At the same time, when the Simon and flanker tasks were investigated separately (e.g., Ellinghaus et al., 2018; Gade et al., 2020; Hübner & Töbel, 2019; Ulrich et al., 2015; Vallesi et al., 2005) or combined together (Frühholz et al., 2011; Hommel, 1997), the results showed different temporal resolution for flanker and Simon tasks. How can we reconcile both aspects? In a tentative way, we proposed that the interaction does not emerge during conflict resolution, but appears on the level of motor responses. For example, the flankers and the irrelevant location could activate their respective motor responses via direct associations already during early stimulus processing (see Hübner et al., 2010). Thus, the flanker condition is first processed, pre-activating the correct response. Then, the Simon condition is processed, pre-activating the correct response again. These pre-activations of motor responses then result in a co-activation of the correct response at the stage of response selection or execution, which affects the duration of response selection or response execution, and in consequence performance. Please note that in addition to mentioning the additive factor logic in the revised version of the manuscript (see p. 50-51), we added a figure (Fig 10) to illustrate this explanation.

B4.1 “The feature overlap account is NOT an account for the interaction, but only a possible account for the discrepancies between studies: according to this account, the interaction can only emerge when there is an overlap between the irrelevant spatial feature producing the Simon effect and a spatial feature of the flanker object(s), thus this overlap is a prerequisite for the interaction (and this would explain why in some studies no interaction emerged while it emerged in others), but it does not specify which is the mechanism producing the interaction. Accordingly, feature overlap should not be presented as an account for the interaction, but as a possible factor influencing it.”

RESPONSE: We agree with Reviewer B, and we now present spatial feature overlap as a possible factor influencing the interaction between Simon and flanker congruency (see p. 11-12). 

B4.2 “It is worth noticing that a paper mentioned by the authors on p. 5 and p. 45 (Treccani et al., 2012) proposes that the interaction only emerges when there is massive overlap between the two irrelevant features (i.e., they are borne by the very same object) and participants process such an overlap (i.e., they “bind” the two features). Thus the “feature binding” is the prerequisite. Yet, the mechanism proposed to account for the interaction is multiplicative priming.” 

RESPONSE: We now better highlight that it is unknown whether the multiplicative priming account can still explain the interaction between Simon and flanker congruency when a more standard design is used (i.e., when irrelevant features are conveyed by different objects). We did it as follows (see p. 10-11): “The multiplicative priming effect of the correct response has been so far tested only by Treccani and colleagues (2009, 2012). In comparison to previous studies (see Table 1), they used a very particular design in which the same object – that is, the irrelevant square – conveyed the irrelevant features for both conditions. This may have changed the weight for processing the irrelevant features so that the irrelevant features could trigger the associated response. Thus, it is possible that in all other studies reported in Table 1 in which the Simon and flanker conditions were combined in a more standard way – that is, with different objects conveying the irrelevant features for both conditions – the priming effect cannot account for the interaction between Simon and flanker congruency. This might be particularly the case because there is some debate in previous research whether in standard Simon and flanker tasks, the irrelevant stimulus feature can co-activate the correct response in congruent trials at all (de Bruin & Sala, 2018; B. A. Eriksen & Eriksen, 1974; C. W. Eriksen & St. James, 1986; Simon & Craft, 1970; Simon & Small, 1969; Wild-Wall et al., 2008; Yeh & Eriksen, 1984).”

B5 “pp. 9-11. The account based on the “learning of stimulus-response mappings” does not actually account for the critical interaction, at least as it is presented (see REASONS** below).

The authors may consider presenting the frequency of presentation of the different S-R pairings simply as a factor that may influence the performance in this kind of tasks (i.e., tasks combining more than one compatibility factor). They may present Experiment 2 as an experiment aimed to investigate the possible impact of this factor and its possible contribution to the interaction (together with OTHER factors) - in principle, at least, this factor may actually counteract, rather than act in favour of the interaction (see below).

If they decide to do so, they should specify in which previous studies this factor may have had an effect, that is, in which previous studies using flanker-Simon paradigms the S-R pairings were presented with a different frequency. Indeed, the rationale of Experiment 2 would become even weaker if in other previous studies the frequencies of different S-R parings were balanced. To make a significant contribution to the understanding of the factors influencing performance in this kind of tasks, I strongly suggest comparing (by means of an ANOVA) the results of Experiment 2 with that of the similar condition of Experiment 1 in which, however, frequencies were not balanced. In this way, the authors may evaluate the specific contribution of this factor.

**REASONS WHY THIS ACCOUNT DOES NOT WORK:

(a) The authors claim that “the stimulus-response mappings for the congruent-congruent trials are the easiest to learn and then to retrieve because there are only two mappings”

This does not make any sense to me : the congruent-congruent trials are not presented in isolation but with the other trials, thus each of these two (I would say) S-R “doubly” congruent pairings are simply one of 8 possible S-R pairings. They cannot be easier to learn or retrieve.

(b) The authors probably meant something else: They probably meant that the numbers of times with which the S-R pairings are presented can explain the interaction (i.e., some S-R pairings may be practiced more). However, I do not understand how.

- When the stimulus irrelevant features are two (e.g., two positions, two flanker identities), and the study provides for the same number of trials in all the four possible categories (a. flanker congruent-Simon congruent; b. flanker incongruent-Simon incongruent, c. flanker congruent-Simon incongruent; d. flanker incongruent-Simon congruent), the number of S-R pairings for each category is exactly the same (i.e, two S-R pairings for each category), thus one would not expect any interaction. To produce the interaction, a study should provide for a larger number of S-R pairings for the categories a and b than for the categories c and d. For example, a study that considers only three categories: totally congruent trials, totally incongruent trials, partially (either flanker or Simon) incongruent trials. This last category would include more S-R pairings (i.e, 4 exemplars), that would be presented less frequently To the best of my knowledge, a similar study has not been conducted, thus this is not an account for the interaction for any two-feature study (If –instead- I am wrong, the authors should explicitly mention this study and explain that in this specific case the interaction may emerge from the addition of two compatibilities effect and the effect of the number of exemplars/frequency)

- When the stimulus irrelevant features are four, and the study provides for the same number of trials in all the four possible categories, the number of S-R pairings for each category is as proposed by the authors: 4 exemplars of the category a (congruent )< 12 exemplars of category c= 12 exemplars of category d< 36 exemplars of category b . The effect of this should be: RT (and/or errors) of aThere is no apparent reason why there should be advantage for the 4-exemplars “a” category over the 12 -exemplars categories, while there should not be an advantage of the 12-exemplars categories (“c” or “d”) over the 36-exemplars category. Quite the opposite, these different frequencies may counteract a possible under-additive interaction produced by other factors (e.g., multiplicative priming). The comparison I suggested (see above) may help to understand the real impact of this factor.

(c) The authors also state that “Explaining performance on this trial type may be sufficient to account for the interaction …The reason is that although previous research has focused on the reduction of the congruency effect of one task when the other task was incongruent, the interaction between Simon and flanker congruency mainly resulted from an increase in the congruency effect of one task when the other task was congruent, in particular from the fast RTs and high rates of correct responses in the congruent-congruent trials”

As explained above, this pattern of performance cannot be explained by the different frequency of S-R pairings given that, based on that, one should expect larger effects of a compatibility factor in both levels of the other factor.

Moreover, I do not think that the interaction mainly results from an increase of the effect of a congruency factor in the congruent level of the other factor: in some studies a congruency effect was found in the congruent level of the other factor, while this effect was no longer significant in the incongruent level of the other factor (thus, a reduction/disappearance of the effect was observed in the other-factor incongruent conditions compared to standard conditions). The authors themselves found a disappearance of the effect of a compatibility factor in the incongruent level of the other factor. E.g.,: “Similarly, the Simon congruency effect was smaller – but not significant – when the trials were flanker incongruent”” 

RESPONSE: We followed Reviewer’s B suggestion, and we presented Experiment 3 (previously Experiment 2) so that its goal was to determine whether a bias in the learning of stimulus-response pairings may explain the discrepancy in previous results (see p. 12-14): 

“A second issue is the question whether the interaction is affected by episodic memory effects, in particular a bias in the learning of stimulus-response pairings (Dreisbach et al., 2007). In all studies using a small stimulus set (i.e., two stimulus features for each Simon and flanker condition), the interaction was observed (see Table 1). However, in this case, the number of stimulus-response pairings is similar across all trial types (i.e., flanker congruent – Simon congruent, flanker incongruent – Simon congruent, flanker congruent – Simon incongruent, flanker incongruent – Simon incongruent, see Fig 2). Thus, the learning of these stimulus-response pairings should be similar across all trial types. 

In contrast, in studies using a larger stimulus set, the results are mixed (see Table 1). Whereas Akçay and Hazeltine (2011) reported no interaction between Simon and flanker congruency, Rey-Mermet and Gade (2016) did observe an interaction. One reason for this finding might be that Rey-Mermet and Gade presented all possible stimulus-response pairings in each trial type (please note that this information was not provided in Akçay & Hazeltine, 2011). However, when the stimulus set consists of four stimulus features (i.e., four locations for the Simon condition and four colors for the flanker condition), the number of stimulus-response pairings differed across trial types (see Fig 2). There are four stimulus-response pairings when both conditions are congruent (i.e., 4 congruent target/flanking colors in each response-congruent location), twelve stimulus-response pairings when one condition is congruent and the other is incongruent (i.e., 4 congruent target/flanking colors x 3 incongruent locations, or 3 incongruent target/flanking colors x 4 congruent locations), and thirty-six stimulus-response pairings when both conditions are incongruent (i.e., 4 target colors x 3 flanking colors x 3 locations). Thus, presenting equally often each trial type (i.e., congruent – congruent, congruent – incongruent, incongruent – congruent, and incongruent – incongruent) results in different frequencies for each stimulus-response pairing. In particular, the stimulus-response pairings from the congruent-congruent trials are presented more often than those of any other trial types. Thus, because this trial type is presented most frequently, there is more chance that participants learn to assign a response to each stimulus exemplar. This may encourage participants to base their performance for this trial type on direct stimulus-response pairings. Assuming that this learning is linear, it is possible that the stimulus-response pairings in trials mixing congruent and incongruent features are better learned than those in incongruent-incongruent trials. The reason is that there are more repetitions of stimulus-response pairings in trials mixing incongruent and congruent trials than those in incongruent-incongruent trials. However, there is no a priori reason to prefer a linear learning process. Therefore, if a bias in the learning of stimulus-response pairings changes the interaction between Simon and flanker congruency, it should be best observed in congruent-congruent trials. This would explain why responses were the fastest and the most correct for the congruent-congruent trial type. Critically, explaining performance on this trial type may be sufficient to account for the interaction between Simon and flanker congruency observed in Rey-Mermet and Gade (2016). In that study, the interaction was driven by fast RTs and high rates of correct responses in the congruent-congruent trials. There was no decrease in RTs (i.e., faster RTs) or increase in correct responses (i.e., higher rates of correct response) in the incongruent-incongruent trials.”

In this context, we also performed an analysis comparing Experiment 3 (previously Experiment 2) with the corresponding condition of Experiment 2 (previously Experiment 1, that is, the group with vertical flankers and horizontal Simon locations). The results revealed that the interaction between Simon and flanker congruency did not differ across both experiments (see p. 46-47). 

B6 “The rationale of Experiments 3a and 3b is not clear. Do these two experiments simply aim to verify a prediction of the multiplicative priming account? The authors claim that “the interaction should not be found when the interference (i.e., the difference between incongruent and neutral trials) was used as dependent measure”. What are the predictions based on other accounts? Could we expect other outcomes based on the other factors mentioned by the authors?” 

RESPONSE: The goal of Experiments 1a and 1b (previously Experiments 3a and 3b) was to determine whether the under-additive interaction between Simon and flanker congruency results from the multiplicative priming of the correct response in a design in which – contrary to the design used by Treccani and colleagues (2009, 2012) – the irrelevant features for both Simon and flanker conditions are conveyed by different objects (see Fig 3). With such a design, in congruent-congruent trials, the correct response is activated not only by one of the irrelevant features (e.g., the position on the screen) but also by the other irrelevant feature (e.g., the flanker; see the top-left panel of Fig 3). This may result in multiplicative effects of priming of the correct response due to co-activation (Mittelstädt & Miller, 2018; Treccani et al., 2009). Thus, this would accelerate trial processing for the congruent-congruent trials not only compared to when the response is primed by only one irrelevant feature (see all other panels of Fig 3), but its facilitative effect would be also stronger than the sum of the single priming effects. This over-additive priming would result in an under-additive RT pattern because the processing of congruent-congruent trials is selectively accelerated. 

However, the multiplicative priming account is based on the assumption that the irrelevant stimulus feature for each Simon and flanker condition co-activate the correct response in congruent trials. As there has been some debate in previous research whether this occurs (de Bruin & Sala, 2018; B. A. Eriksen & Eriksen, 1974; C. W. Eriksen & St. James, 1986; Simon & Craft, 1970; Simon & Small, 1969; Wild-Wall et al., 2008; Yeh & Eriksen, 1984), it is possible that both the irrelevant location and flanking letters do not co-activate the correct response. In this case, the interaction should not be observed in the facilitation effect. This implies that the interaction should be observed in the interference effect. The reason is that as the congruency effect is computed as the addition of the interference effect and the facilitation effect, finding an interaction in the congruency effect but not in the facilitation effect should result in finding an interaction in the interference effect. This is now clarified in the manuscript (see p. 15-17). 

MINOR POINTS

1. “From the manuscript: “Simon Task, flanker task, and their interaction” “Previous research has shown an interaction between the Simon and flanker tasks when both tasks were combined within the same trial”.

It is worth noticing that the interaction is not between two tasks. Indeed, the task is only one: participants had to indicate the colour of the central letter while ignoring the flanking letters and their location on the screen. This task had the characteristics of both typical Simon and flanker tasks.

I would say that the interaction is between the flanker-target identity congruence (the compatibility factor usually observed in flanker tasks) and the stimulus-response spatial correspondence (the compatibility factor usually observed in flanker tasks).

Speaking about two tasks may be confusing to the reader (one may think of a dual-task paradigm) and makes it harder to understand the proposed accounts for the interaction.

For example, that authors claim that “One can assume that the conflict monitoring system estimates the current levels of conflict for the task being processed first. If this task is incongruent and thus associated with a high degree of conflict, this leads to a shift of control signal. That is, response-relevant features are activated while all irrelevant features – including those for the task being processed second – are inhibited. Thus, the strength of control is also adjusted for the second task to be processed. “

What are the tasks the authors are speaking about?

I guess they are speaking about irrelevant features, instead of tasks, which are not to be processed (given that they are task-irrelevant), but they are processed nevertheless.

I would rather write: “One can assume that the conflict monitoring system estimates the current levels of conflict. If the first irrelevant feature being processes (e.g., stimulus position) is incongruent with the relevant one, and thus associated with a high degree of conflict, response-relevant features are activated while all irrelevant features (not only that already processed) are inhibited.“

RESPONSE: We agree with Reviewer B, and we changed the terminology accordingly throughout the manuscript where appropriate. Moreover, we changed the description of the within-trial conflict monitoring (see p. 7): “Here, one could assume that if the first irrelevant feature being processed (e.g., the irrelevant flanker) is incongruent with the relevant one and thus associated with a high degree of conflict, this leads to a shift of control signal. As a consequence, the response-relevant feature is activated while all irrelevant features – including the irrelevant Simon location – are inhibited. Thus, the strength of control is also adjusted for the second irrelevant feature. This reduces the impact of the irrelevant features if the second irrelevant feature is also incongruent, thus resulting in a smaller congruency effect. According to this account, control processes generalize across all irrelevant features, explaining the under-additive interaction observed in most previous studies.”

2. “Also, what is exactly inhibited according to the conflict-monitor account? Following some versions of this account (e.g., the “gate” version), when a conflict between competing response codes is detected by the conflict monitoring mechanism, the weight of the route activated by the irrelevant feature(s) is attenuated. Do the different versions of the conflict-monitoring account have different predictions?”

RESPONSE: At first glance, we would say that the different versions of this account would not differ in their predictions.

3. "Actually, as underlined above, I find this account not very convincing. If the gate closes, it should close in the first place for the direct route involving the feature for which a conflict is detected. Moreover, it should take a while for the gate to close, thus the effect of this closure should be observed in the following trial.”

RESPONSE: This account is presented because previous research have mentioned it as an explanation for such interactions.

4. “p.8 " Hommel [6] (see Table 1), suggesting that this effect could explain the observed interaction in some experiments but not in others".

What effect? Please be more explicit “suggesting that this phenomenon(i.e., the decay of the stimulus spatial code) could explain”

RESPONSE: We changed the text as follows (see p. 10): “the decay of response codes could explain the observed interaction in some experiments but not in others”.

5. “Please fix inconsistencies in lexical choices. “Stimulus-response mapping” (e.g., p.9) usually means the instructed (i.e., contained in the instructions) associations between specific stimulus values and the correct responses(i.e., the responses required by these values). I think that the authors should use instead “S-R pairings” (e.g., S-R congruent pairings). Note that the authors themselves use “stimulus-response mapping” with the standard meaning on p. 14.

RESPONSE: We thank Reviewer B for proposing such a clear differentiation. We applied it throughout the manuscript. 

6. p. 14. “The stimulus was determined randomly for each trial” Maybe better (more clear) “the stimulus colour was…”.

RESPONSE: We changed the sentence according to the suggestion (see p. 18) 

7. “p. 16. In contrast with the horizontal-key condition, the keys in the vertical-key condition are not perfectly aligned along one spatial axis, the vertical one (i.e., they are not perfectly one above the other). Can the authors justify this choice? (in many keyboards, there are numerical keys which are vertically aligned).”

RESPONSE: We opted for these keys because they are next to the keys we used for the horizontal Simon condition in this study as well as in the previous one (Rey-Mermet & Gade, 2016). In addition, if we selected keys from the numerical keyboard, we would have had a problem for the horizontal Simon condition. Here, there are no four keys in a row. 

8. “pp. 14-16. I did not understand whether stimuli were of different size in the different conditions (it is not clear from the description) and whether in the flanker pure blocks the asterisks appeared in a (central) yellow square as in the pure Simon blocks. These seem unnecessary differences between experimental conditions.”

RESPONSE: In the pure Simon and flanker blocks as well as in the mixed blocks, all stimuli were presented in 28-point Arial Bold font. In the pure flanker blocks, the stimuli were not presented in a yellow square because all stimuli were presented centrally. This is now clarified in the manuscript (see p. 19).

---

## [Decision Letter · Decision Letter 1]

16 Dec 2020

PONE-D-20-15659R1

Multiplicative priming of the correct response can explain the interaction between Simon and flanker congruency

PLOS ONE

Dear Dr. Rey-Mermet,

Thank you for submitting your revised manuscript to PLOS ONE. The previous reviewers agreed to review your revision. Whereas Reviewer 1 is completely satisfied with this revision, Reviewer 2 raised a number of concerns (see the detailed review to upload). I invite you to submit a revised version of your manuscript addressing the points raised by R2. If you disagree with some of R2’s points, please make that clear in your response letter. I will send back the revision to R2.

A rebuttal letter that responds to each point raised by the Reviewer 2. You should upload this letter as a separate file labeled 'Response to Reviewers'.A marked-up copy of your manuscript that highlights changes made to the original version. You should upload this as a separate file labeled 'Revised Manuscript with Track Changes'.An unmarked version of your revised paper without tracked changes. You should upload this as a separate file labeled 'Manuscript'.

We look forward to receiving your revised manuscript.

Kind regards,

Ludovic Ferrand, Ph.D.

Academic Editor

PLOS ONE

Reviewers' comments:

Reviewer's Responses to Questions

**Comments to the Author**

1. If the authors have adequately addressed your comments raised in a previous round of review and you feel that this manuscript is now acceptable for publication, you may indicate that here to bypass the “Comments to the Author” section, enter your conflict of interest statement in the “Confidential to Editor” section, and submit your "Accept" recommendation.

Reviewer #1: All comments have been addressed

Reviewer #2: (No Response)

2. Is the manuscript technically sound, and do the data support the conclusions?

Reviewer #1: Yes

Reviewer #2: Partly

3. Has the statistical analysis been performed appropriately and rigorously? 

Reviewer #1: Yes

Reviewer #2: Yes

4. Have the authors made all data underlying the findings in their manuscript fully available?

Reviewer #1: No

Reviewer #2: Yes

5. Is the manuscript presented in an intelligible fashion and written in standard English?

Reviewer #1: Yes

Reviewer #2: Yes

6. Review Comments to the Author

Reviewer #1: The authors have addressed my previous concerns. I find the revised version convincing, and I believe it is ready for publication.

Reviewer #2: (No Response)

7. PLOS authors have the option to publish the peer review history of their article (what does this mean?). If published, this will include your full peer review and any attached files.

Reviewer #1: No

Reviewer #2: No

---

## [Author Response · Author response to Decision Letter 1]

26 Jan 2021

Reviewer A

A1.1 “In my view, one of the most questionable points concerns the “tentative” account the authors propose at the end of the General Discussion to reconcile their results with some of the previous findings. To do so, in fact, they propose an account that is inconsistent with many other findings and with the main hypotheses about how responses are selected in this type of compatibility tasks.

Forgive me for the long explanation below, but I need to be sure that the authors and I refer to the same theoretical framework.

The interaction between two factors that likely affect the same (response selection) stage (as it is the case with Simon and flaker compatibilities) is thought to depend on (a) the overlap in time between the effect of the processing of one irrelevant attribute and the effect of the pro-cessing of the other irrelevant attribute, and (b) on the overlap of the processing of these at-tributes with the processing of the relevant attribute and the controlled-route activation of the correct response (see, e.g., Gevers, Caessens, Fias, EJCP, 2010; Kornblum, Stevens, Whipple, & Requin, 1999; Ridderinkhof et al.,2004). The overlap or the lack of overlap between these processes are not fixed properties of a given paradigm (e.g., Simon paradigm), but they de-pend on the particular structure of the specific task that is used and on the relevant and irrel-evant stimulus features that the task involves: some features (e.g., colors) can be processed and discriminated before others (e.g. letters, direction of arrows). 

Usually, in standard horizontal Simon tasks, the Simon effect is observed at the first bins of RT distributions and it is not observed when the relevant feature is relatively difficult to dis-criminate (and, thus, when responses are relatively slow). This has been explained on the ba-sis of a temporal overlap account (Hommel 1994). The spatial code in standard Simon tasks is thought to be formed quite early: immediately after stimulus onset (e.g., indeed, according to the attention-shift account of the Simon effect, stimulus spatial coding occurs because atten-tion has to move towards he stimulus and thus the irrelevant spatial code is formed even be-fore the relevant stimulus feature is analysed). Afterward, the spatial code decays. Therefore, this code can have an effect (i.e., it can speed up RTs wen the primed response corresponds to the correct one or lengthen RTs wen the primed response is wrong, and thus needs to be inhibited) only if the activation of the correct response by the relevant attribute occurs shortly after, that is, only in the case of relatively fast responses. Response priming mediated by the automatic routes, the correct-response activation mediated by the controlled route, and the conflict between them, have been shown to occur AT RESPONSE/MOTOR RELATED STAGES that can be tapped through techniques assessing either muscle or motor cortex activation: lateralized readiness potentials (De Jong et al., 1994) and EMG recordings show an initial activation of the spatially corresponding response (shortly after stimulus presentation) followed by a de-crease of such activation when this response is the incorrect one and, finally, by the activa-tion of the correct response (see, e.g., Treccani, Cona, Milanee, Umiltà, Psych. Res, 2018 for a review).

The irrelevant flanker feature (the one that the flanker has in common with the target) is thought to exert its effect on the same processing stage as the irrelevant spatial code (i.e., the response selection stage) but it usually exert its effects later – indeed, the coding of the irrel-evant flanker attribute cannot precede the processing of the target relevant feature, as hap-pens instead in the case of stimulus location. Response priming triggered by the flanker can overlap in time with the (controlled) activation of the correct response but, typically, we ob-serve an effect of the former on the latter (i.e., we observe, a flanker effect) only in later bins compared to those in which we observe the Simon effect, and typically we observe that the flanker congruency effect increases as RTs increase. Therefore, when the two tasks are com-bined in a flanker Simon tasks, one should expect that, most of the times, the effects of the two factors do not overlap in time and they do not interact with each other (see, e.g., Gevers et al., 2010, for a similar explanation for either the interaction or lack of interaction between Simon and SNARC effects). This may explain why in many studies no significant interaction between Simon and flanker effects was found.

However, this is not a fixed rule. If, for example, the task is such as to allow a very fast pro-cessing of the feature shared by the target and flanker, or to prevent a fast processing of the irrelevant spatial feature leading to the Simon effect, than an overlap between the effects of these processes (and a significant interaction) can be observed (cf., e.g., Gevers et al., 2010 for the modulation of the interaction between Simon and SNARC that either emerged or did not emerge according to the feature that participants had to judge)

The accessory-stimulus Simon task used by Treccani et al. is an example of a task in which the Simon effect increases as RTs increase, just as the flanker effect (and this is a quite sen-sible explanation of why in their case the overlap occurs and a significant interaction was found – see also my ‘other points’ below) but similar temporal patterns have been obtained with other tasks that are not the standard one. For example, stable or increasing Simon effect functions have been also obtained for vertical Simon tasks, that is, tasks in which stimuli and responses are aligned vertically rather than horizontally (Vallesi et al, Cognition, 2005; Wie-gand & Wascher, Ps. Res., 2007). By using these tasks combined with a flanker task, one should thus expect an interaction between the Simon and flanker effects. Interestingly, in some of the experimental tasks used here, stimuli and responses were aligned vertically (i.e., the task was a sort of vertical Simon task). In fact, even the “horizontal” tasks administered here were clearly not standard Simon tasks and we could expect the temporal trend of the Simon effect to be different from the typical one. Indeed, in their experiments, the authors DO NOT observe the typical decreasing Simon effect functions: they found that “the Simon con-gruency effect initially increased while the RTs increased from fast to moderate”. 

This means that, in the present study, there was indeed a temporal overlap between the Simon and flanker effects (i.e., the effect of the stimulus location and of the flanker non-spatial at-tribute), which is the prerequisite for an interaction between the two effects to occur. 

ACCORDINGLY, I THINK THAT THERE IS NO NEED TO ADD AD-HOC ASSUMPTIONS TO EXPLAIN WHY AN INTER-ACTION WAS FOUND HERE.

For example, the authors propose that: “A tentative way to explain these findings is that that the interaction does not emerge during conflict resolution, but appears on the level of motor responses”. However, the way, in which the authors explains the multiple- priming hypothesis clearly refers to a stage that previous research has identified as the stage in which motor priming AND response conflict can occur: there cannot be a stage in which response conflict occurs and a stage in which responses are primed, at least following the current models of response selection. The conflict in incongruent trials occurs because some responses are primed and these responses are not the correct ones. The authors also maintain that “…the flankers and the irrelevant location could activate their respective motor responses via direct associations already during early stimulus processing”-> This is indeed plausible and it is consistent with many models of response section: the response/motor related stage (in which responses are automatically primed by irrelevant information, while others are chosen, via controlled routes, on the basis of the relevant features) does not need to start after stimulus processing is completed, but this does not mean that there is such a thing as a stage in which response priming occurs and another stage in which response is selected.”

RESPONSE: First, we would like to thank the Reviewer to synthesize their opinion in such a precise way. Nevertheless, we have a different opinion about the cognitive processes underly-ing the interaction between Simon and flanker congruency. Our reason is that the EEG find-ings are not directly compatible with an account assuming an overlap in time. The EEG results clearly showed separate ERPs associated to the processing Simon and flanker irrelevant fea-tures, although an interaction between both congruency variables was observed for reaction times. As put forward in the previous review, the evidence is so far limited as there is only three EEG studies investigating the interaction between congruency variables (see Frühholz et al., 2011, for a Simon-flanker combination; Kałamała et al., 2020, for a Stroop-semantic com-bination; and Rey-Mermet et al., 2019, for a Stroop-flanker combination). This is now clearly acknowledged in the revised manuscript (see p. 54). However, in our opinion, simply ignoring these findings does not belong to the integrative research process we all aimed for as a scien-tific community. Therefore, in the revised manuscript, we still present the EEG findings and our tentative way to integrate these findings to the multiplicative priming account (see p. 54-56 and Fig 10). However, at the same time, we have emphasized their limitations (see p. 53-55) and the theoretical basis used to formulate the tentative integration (i.e., the dual-stage two-phase model of selective attention put forward by Hübner and colleagues (2010) in which the pre-activation of responses in a first stage results in a response conflict at a later stage; see p. 54-56). 

Importantly, we acknowledge that the point of view put forward by the Reviewer is fundamen-tal. Therefore, for the sake of consensus, we have integrated it to the description of the EEG findings and the tentative explanation as follows: “Taken together, previous research as well as the present results put forward the versatility of the interaction between Simon and flanker congruency, with some experiments showing an interaction, whereas other experiments show-ing no interaction. This emphasizes the necessity of an account in which the boundaries of when the interaction is assumed to occur and when it assumed to not occur are clearly formu-lated. Based on the additive factor logic (Sternberg, 1969, 1998) according to which an inter-action in RTs results from the fact that two variables influence the same processing stage, a starting point may be to assume that the interaction between Simon and flanker congruency should be observed when both Simon and flanker features are processed during the same stage. That is, the interaction should occur when there is an overlap in time (1) between the processing of Simon irrelevant feature and the processing of flanker irrelevant feature and (2) between the processing of these irrelevant features and the processing of the relevant target feature. Moreover, the overlap or the lack of thereof between these processes might not be fixed properties of a given paradigm (e.g., the Simon task or the flanker task). Rather, they seem to depend on the particular administration of the task combination and on the relevant and irrelevant stimulus features that the task combination involves. Thus, when the flanker and Simon congruency variables are so administered that the flanker congruency effect increases as RTs increase but the Simon congruency effect decreases as RTs increase, the processing of both irrelevant and relevant features would not overlap in time, and both congruency varia-bles should not interact. In contrast, when both congruency variables are so combined that the processing of Simon and flanker irrelevant features overlap in time with the processing of the relevant feature, the interaction should occur. The account presented here is so far very similar to the temporal overlap account proposed by Hommel (1997), except to one key difference. Here, we do not assume a spontaneous decay of the representation features. So far, this is still an option to explain the interaction between Simon and flanker congruency. However, the inhibition of irrelevant features (such as proposed in the conflict monitoring account) or the multiplicative priming of the correct response are also valid options. 

Nevertheless, assuming an overlap in time when processing Simon and flanker features has two drawbacks. First, it is not really clear from the previous research as well as from the pre-sent study when an interaction should be expected to occur and when it should not be ex-pected to occur. This makes difficult to formulate a priori the conditions under which there is an overlap in time resulting in an interaction, and the conditions under which there is no overlap in time and thus no interaction. Second, as mentioned in the introduction, the assumption that the time overlap in processing Simon and flanker irrelevant features results in an interaction in RTs is incompatible with the EEG findings (see Frühholz et al., 2011; Kałamała et al., 2020; see also Rey-Mermet et al., 2019). The studies using EEG clearly showed an interaction in RTs but separate ERPs associated to the processing of the irrelevant features for both con-gruency variables. Nevertheless, the EEG evidence is so far limited as only three studies have been performed on paradigms combining different congruency variables (see Frühholz et al., 2011, for a paradigm combining Simon and flanker congruency; and Kałamała et al., 2020, for a paradigm combining Stroop and semantic congruency; Rey-Mermet et al., 2019, for a para-digm combining Stroop and flanker congruency). This emphasizes the necessity of further research, in particular EEG research combining different congruency variables, in order to test for the robustness of these findings.

Although the EEG findings might be treated with caution due to the limited number of studies, we would like to mention that there is a way to integrate the EEG findings with the multiplica-tive priming account. This is possible if one assume that the interaction between Simon and flanker congruency does not emerge during conflict processing, but appears on the level of motor responses. For example, the flankers and the irrelevant location could activate their re-spective motor responses via direct associations already during early stimulus processing. This assumption is based on the model of selective attention and response selection put for-ward by Hübner and colleagues (Hübner et al., 2010). According to this model, in early stages, stimuli are processed by (incomplete) perceptual filtering. Thus, in addition to the pre-activation of the response associated to the relevant feature (i.e., the correct response), irrele-vant features might still pre-activate their responses. Then, in a later stage, the selected fea-ture (ideally the relevant feature) drives response selection. Thus, following this model and in line with the EEG results, one could assume in case of the interaction between flanker and Simon congruency variables that the flanker irrelevant and relevant features are first pro-cessed, pre-activating the correct response when trials are flanker congruent. Then, the Simon irrelevant and relevant features are processed, pre-activating the correct response when trials are Simon congruent. These pre-activations of motor responses then result in a co-activation of the correct response at the stage of response selection, which affects the duration of re-sponse selection or response execution, and in consequence performance. An illustration of this explanation is given in Fig 10. So far, this tentative explanation is purely speculative, thus emphasizing the necessity of further research.”

A1.2 “Thus, the flanker condition is first processed, pre-activating the correct response. Then, the Simon condition is processed, pre-activating the correct response again. These pre-activations of motor responses then result in a coactivation of the correct response at the stage of response selection” > The flanker or Simon “conditions” cannot (as a rule) activate the correct response. Maybe, the authors meant that the flanker irrelevant feature pre-activates the associated response, which, in flanker congruent trials, is the correct response and the target position pre-activates the associated response which, in Simon congruent tri-als, is the correct response. Again, this activation (response/motor priming) is part of the re-sponse selection processes and is what is thought to cause conflict or facilitation at the re-sponse selection stage.”

RESPONSE: We followed Reviewer’s suggestion by modifying the sentences about flanker and Simon features activating the correct responses, that is: “[…] the flanker irrelevant and relevant features are first processed, pre-activating the correct response when trials are flank-er congruent. Then, the Simon irrelevant and relevant features are processed, pre-activating the correct response when trials are Simon congruent” (see p. 55).

A2.1 “I still find the “bias in learning” account not very convincing. For example, the authors claim “In particular, the stimulus-response pairings from the congruent-congruent trials are presented more often than those of any other trial types. Thus, because this trial type is pre-sented most frequently, there is more chance that participants learn to assign a response to each stimulus exemplar”, which would result in this type of trials leading to the best perfor-mance. 

Following the same line of reasoning, one may claim that the stimulus-response pairings from the incongruent-incongruent trials are presented less often than those of any other trial types, which would result in this type of trials leading to the worst performance and imply additivity between flanker and Simon congruencies. 

Similarly, the authors claim that “there is no a priori reason to prefer a linear learning pro-cess”, that is, a trend that would result in additivity, rather than in an interaction. Yet, there is no reason to think that learning follows an exponential growth curve either (i.e., a trend that would result in an under-additive interaction) or, vice-versa, a logarithmic growth curve, or, even, a curve with an initial steep slope followed by a plateau (i.e., trends that would result in an over-additive interaction). It depends on the shape of the learning curve and on the part of the curve that is tapped by this particular task (the steep or flat part). I do not know whether there are studies on the learning curve with this kind of material (if yes, the authors may con-sider mentioning such studies).

Anyway, I suggest the authors claim, more simply, that when the values of the irrelevant stim-ulus 

dimensions are more than two, there are different numbers of exemplars (S-R pairings) for the different trial types, which, in turn, leads to different frequencies of presentation of these pair-ings. The resulting learning effect (better performances for more frequent, and thus more practiced, pairings) might shape the Simon-flanker congruency interaction in unpredictable ways. One of the possible ways in which learning may affect performance results in an under-additive interaction between the two congruency factors: this would occur when more practice does not lead, at first, to better performance (e.g., going from the few presentations of each of the incongruent-incongruent S-R pairings to the relatively more frequent incongruent-congruent S-R pairings) but a significant improvement can be found with much more practice (e.g., with the congruent - congruent S-R parings).”

RESPONSE: Following Reviewer’s suggestion, we modified the manuscript as follows: “This may result in a better learning for the stimulus-response pairings which are presented more frequently and thus practiced more often. It is so far unknown how this learning might shape the interaction between Simon and flanker congruency. For example, it is possible that more practice does not lead, at first, to better performance (e.g., going from the few presentations of each of the stimulus-response pairings for incongruent-incongruent trials to the relatively more frequent presentations of the stimulus-response pairings for trials mixing incongruent and con-gruent features). However, a significant improvement in performance may be found with much more practice, such as when the stimulus-response pairings are presented very frequently in congruent-congruent trials.” (see p. 14).

A2.2 “P. 10. “the central assumption put forward in both accounts [within-trial conflict moni-toring and spontaneous decay accounts] has been put into question by a study using ERPs […] the results suggest that although flanker conflict and Simon conflict are resolved at sepa-rable stages, there appears to be little influence of the earlier stage on the later one”-> This is misleading: if it were true (two different stages for the two effects), then the multiplicative priming account too should be rule out. According to this account, that the two irrelevant fea-tures of the stimulus display (the position and the feature that the flanker shares with the tar-get) affect the same processing stage (the one in which automatic response priming and con-trolled response activation occur, responses competes and the correct one is selected). In-deed, there is plenty of literature (about 50 years of studies) showing that the Simon and flanker effect emerge at the same stage (the response selection stage), thus I would not claim that the two effects emerge at different stages (this claim would require a long discussion). Actually, evidence showing that the two effect emerge at the same response selection stage is also the main evidence challenging the temporal-overlap account proposed by Hommel for the interaction between the two effects. According to Hommel (1997), the two effects emerge at different stages: (stimulus-encoding and response-selection stages).”

RESPONSE: We did not intend to claim that the two effects “emerge” on different stages. Ra-ther, we propose that initial conflict processing occurs at different early stages, and their ef-fects then combined at the final response selection stage. Nevertheless, as the sentence put forward by the Reviewer seems to be the start of the misunderstanding, we rewrote it as fol-lows: “In contrast, the results suggest that there is little influence of the first conflict on the pro-cessing of the second conflict.” (see p. 10).

A3 “P. 17 lines 327-342. This part is very difficult to read and maybe also not very useful. The whole thing can be explained in a more simple way. The authors wanted to evaluate if and how an effect can modulate the other and they want to evaluate both the facilitation and interfer-ence components, which is a quite sensible strategy here.”

RESPONSE: The goal of this paragraph is to present the hypotheses of Experiment 1. There-fore, in our opinion, it is not sufficient to simply state that we aimed to evaluate if and how an effect can modulate the other. Nevertheless, we agree with the Reviewer that the previous version of this paragraph was difficult to read. Therefore, we re-wrote it (see p. 17-18): “In both experiments, we hypothesized that if the interaction between Simon and flanker congruency is affected by the co-activation of the correct response when the trial is both Simon and flanker congruent (Mittelstädt & Miller, 2018; Treccani et al., 2009), the interaction should be observed when the analyses focused on the facilitation effect. The reason is that the facilitation effect is computed as the difference between congruent and neutral trials, and congruent trials are as-sumed to induce some facilitative priming. In contrast, the interaction should not be found when the analyses focused on the interference effect because this difference involved incon-gruent and neutral trials (i.e., trials in which no facilitative priming is assumed to occur). How-ever, these hypotheses is based on the assumption that the irrelevant stimulus feature for each Simon and flanker congruency co-activate the correct response in congruent trials. This assumption has been debated in previous research (de Bruin & Sala, 2018; B. A. Eriksen & Eriksen, 1974; C. W. Eriksen & St. James, 1986; Simon & Craft, 1970; Simon & Small, 1969; Wild-Wall et al., 2008; Yeh & Eriksen, 1984), raising the possibility that both the irrelevant lo-cation and flanking letters do not co-activate the correct response. In this case, the co-activation of the correct response would not occur when the trial is both Simon and flanker congruent. If so, no interaction should be observed when the analyses focused on the facilita-tion effect.”.

A4 “P. 31. “Together, these findings support the explanation that the interaction between Si-mon and flanker congruency results from the multiplicative priming occurring when the trials were congruent for both Simon and flanker conditions”. In my view, these results suggest that the interaction can be found even with this paradigm and that the modulation of one effect on the other mainly involve the facilitation component. I do not see these results of the first two experiments as evidence of the multiplicative priming account. Actually, I do think that neither these experiments nor the following ones directly test this account. This study simply shows that the Simon-flanker interaction is a robust phenomenon, that is, it is not a phenomenon that can only be observed in some peculiar paradigms (the authors observed the interaction in many different experiments - the evidence is strong) and it is not modulated by many of the factors that one may think of as critical factors. The multiplicative priming account appears to be a plausible account for these results (i.e., it is consistent with the pattern of results of the study). The study also offers some useful ideas about how to test this account in future stud-ies. In my view, these are the only fair and reasonable conclusions that can be drawn from the results (and they are not trivial at all).”

RESPONSE: Following Reviewer’s suggestion, we modified the text as follows: “Together, these findings suggest that the interaction between Simon and flanker congruency can be found even with a paradigm involving neutral trials and that the modulation of one congruency effect by the other congruency effect mainly involve priming or facilitation.” (see p. 32). For the sake of consistency, we also changed the further sentences referring to the results of Experi-ment 1 (see p. 36 and 50).

A5 “P. 31 lines 599-607. “It might be surprising that for the Simon condition we observed a reverse facilitation effect”. In my view, this is not surprising at all: That may happen when, in neutral trials, stimuli are presented at fixation, that is, where visual acuity is higher (cf., e.g., Umiltà, Rubichi & Nicoletti, 1999). The benefit of stimuli being presented at fixation in neutral trials may be greater than the facilitation produced by the irrelevant stimulus location in spa-tially congruent trials. 

It is also worth noting that it is not uncommon to find a reverse Simon effect in task combing the Simon paradigm with other types of compatibility paradigms (see, e.g., the many studies about the logical-recoding mechanisms in tasks combining Simon and symbolic compatibility or Keus & Schwarz, 2005, for a task combining Simon and SNARC paradigms).”

RESPONSE: Following Reviewer’s suggestion, we changed the formulation as follows: “For the Simon congruency, we also observed a reverse facilitation effect (i.e., slower performance on congruent trials than on neutral trials; see Aisenberg & Henik, 2012, for a similar finding). This finding may result from the fact that our Simon neutral stimuli were presented at the cen-ter of the display where the visual acuity is higher and where no attentional selection process is required (Umilta et al., 1999).” (see p. 32). 

We decided to not mention the study with the SNARC paradigm (i.e., Keus & Schwarz, 2005) as this paradigm was not introduced in the present manuscript. 

MINOR POINTS

1. “P. 3: “In both Simon and flanker tasks, trials are labelled incongruent when they contain features for two different response alternatives -> .maybe better “when they contain fea-tures pointing to two different response alternatives.” 

RESPONSE: We changed the sentence according to Reviewer’s suggestion (see p. 3).

2. “P. 3: “a trial is incongruent in the Simon task when the location on the screen is different from the response location” To be consistent with the sentence above, one should specify that "a trial is incongruent in the Simon task when the stimulus location on the screen ac-tivates a response that is different from the correct response".“

RESPONSE: We changed the sentence according to Reviewer’s suggestion (see p. 3).

3. “P. 3: “In the flanker task, a trial is incongruent when the color of the central letter is dif-ferent from “”Again, to be consistent with the way in which the authors describe the in-congruent condition in the two tasks, one should say that "In the flanker task, a trial is in-congruent when the correct response (i.e., the response activated by the color of the cen-tral letter) is different from that activated by the color of the flanking..." I also suggest modifying the description of the congruent conditions accordingly.“

RESPONSE: We followed Reviewer’s suggestion, but to avoid wordiness, we formulated the sentence as follows: “the response activated by the color of the central letter is the same as the response activated by the color of the flanking letters”. Moreover, we modified accordingly the description of the congruent conditions (see p. 3).

4. “P. 4: “Findings from correlational studies showed, however, small to moderate correla-tions between both congruency effects”. I would mention, however, that many studies show that there small to moderate correlations even between the congruency effects ob-served in two different administrations of the very same task (cf., e.g., Draheim, Mash-burn, Martin, & Engle, 2019).”

RESPONSE: Different administrations of a task may change the nature of the process, thus possibly explaining the small to moderate correlations (see the General Discussion of Rey-Mermet et al., 2020, for a recent discussion of this point). Moreover, test-retest reliability – which reflects the correlation between the measures of the same task with the same admin-istration – are slightly larger (from .40 to .60; see Hedge et al., 2018) than the average correla-tion between the congruency effects of different tasks (i.e., smaller than .20; see von Bastian et al., 2020). 

5. “P. 4: “indicating that the congruency effect of one condition (e.g., the flanker condition) was smaller when the trials of the other condition (in this case, the Simon condition)”. I would say: "indicating that one congruency effect (e.g., the flanker effect) was smaller in trials that were incongruent with respect to the other congruency factor (in this case, Si-mon congruency).” 

RESPONSE: We followed Reviewer’s suggestion as follows: “[…] one congruency effect (e.g., the flanker congruency effect) was smaller in trials that were incongruent with respect to the other congruency variable (in this case, Simon incongruent trials)” (see p. 4).

6. “P. 7: "Previous research has used different theoretical accounts to explain the interac-tion between Simon and flanker congruency. A first explanation is based on the conflict monitoring framework [19]" -> this gives the wrong impression that the interaction be-tween Simon and flanker congruencies has been already explained on the basis of the conflict monitoring account and that Botvinick et al. ([19]) were those who proposed this explanation of the interaction. In contrast, as I understood it, the authors themselves are those who proposed that this interaction may be explained by this account. In a previous paper, they proposed a similar account for other types of interaction between congruency factors. This should be explained more clearly. For example: "Previous research has used different theoretical accounts to explain the interaction between congruency factors. As proposed for other types of interaction [7], the underadditive interaction between Simon and flanker congruency could be accounted for by the conflict monitoring hypothesis. Following this hypothesis, the conflict monitoring.... accordingly [19]".”

RESPONSE: In our opinion, it is also important to state that the conflict monitoring framework was originally used to explain sequential modulation of conflict across trials. Therefore, we followed Reviewer’s suggestion as follows: “Previous research has used different theoretical accounts to explain the interaction between congruency variables. A first attempt is based on the conflict monitoring framework (Botvinick et al., 2001), which assumes that a conflict moni-toring system estimates the current levels of conflict and adjusts control accordingly. While conflict monitoring has originally been used to account for sequential modulations of control across trials (see Duthoo et al., 2014; Egner, 2007, for reviews), it has also been applied to explain the within-trial interactions between the different congruency variables (Boy et al., 2010; Rey-Mermet et al., 2019; Rey-Mermet & Gade, 2016). In particular, to account for the interaction between Simon and flanker congruency, one could assume […]” (see p. 7).

7. “Here [p. 7] and in the other parts of the manuscript I would not refer to Simon, flanker or Stroop congruency as "congruency conditions," bur rather "congruency factors". In gen-eral, the term "conditions" is used to refer to levels of a factor (i.e., an independent varia-ble.) For example, in the S-S and S-R compatibility literature, congruency conditions usu-ally refers to the congruent and incongruent conditions of a particular congruency fac-tor.”

RESPONSE: We thank the Reviewer for pointing out this possible misunderstanding. Never-theless, we think that the term “factor” might also be misleading because it can be associated to other types of analyses (e.g., confirmatory factor analysis). Therefore, we opted for the term “variable”, “congruency” or “congruency variables”, depending on the case (see p. 6-8, 10-19, 22-23, 25-27, 29-32, 34, 36-40, 42-45, 49-50, 52-56). 

8. “P. 7: I would write“ irrelevant stimulus location” instead of “irrelevant Simon location””

RESPONSE: We changed the sentence according to Reviewer’s suggestion (see p. 7).

9. “P. 7: "the spontaneous decay of response-irrelevant features" [6]-> this is a bit mislead-ing: stimulus features do not decay. According to Hommel (e.g., Hommel, 1994), it is the representation of these features that decays, becomes less active.” 

RESPONSE: We changed the relevant sentences according to Reviewer’s suggestion (see p. 7 and 9).

10. “P. 7: “longer opportunity for the irrelevant location to decay” -> “longer opportunity for the irrelevant location CODE to decay”.

RESPONSE: We changed the relevant sentences according to Reviewer’s suggestion (see p. 7 and 10).

11. “P. 8: "According to this account, the correct response is primed not only by the target feature but also by the irrelevant" ->"According to this account, in the (Simon) congruent- (flanker) congruent conditions, the correct response is not only activated by the target feature, but also primed .by the .. ". Only in this fully congruent condition , indeed, the values of all the features activate the same (correct) response”

RESPONSE: We changed the sentence according to Reviewer’s suggestion (see p. 8).

12. “P. 8: “conveyed the irrelevant flanker color for the flanker condition”. Again, this “flanker condition” is not clear. Maybe “conveyed the irrelevant flanker color that causes the flanker effect”?”

RESPONSE: We changed the sentence according to Reviewer’s suggestion (see p. 8).

13. “P. 8: “because it was assumed to trigger a location congruent or incongruent to the re-sponse" "The "trigger location" bit is not clear. Maybe it is better to claim "... because the square location may be either congruent or incongruent with the correct-response loca-tion"” 

RESPONSE: In order to keep the same structure as the previous sentence, we changed the sentences as follows: “At the same time, it also conveyed the irrelevant location that causes the Simon congruency effect. The reason is that the square location was either congruent or incongruent with the location of the correct response.” (see p. 8).

14. “P. 9: "trials compared to the corresponding trials mixing incongruent and congruent features"->"trials compared to trials mixing incongruent and congruent features"”

RESPONSE: We changed the sentence according to Reviewer’s suggestion (see p. 9).

15. “P. 11: some spatial overlap in the features of the Simon and flanker conditions” -> this is not clear. What about "some overlap between the stimulus irrelevant position, which causes the Simon effect, and some spatial feature of the flankers"”

RESPONSE: In this case, to avoid wordiness, we opted for a simpler solution, that is: “some spatial overlap in the features of the Simon and flanker congruency variables” (see p. 11).

16. “P. 11: “the irrelevant features for both conditions” -> better "the irrelevant features caus-ing both the Simon and flanker effect"”

RESPONSE: In this case, to avoid wordiness, we opted for a simpler solution, that is: “the irrelevant features for both congruency variables” (see p. 11).

17. “P. 11: “The Simon condition included horizontal positions and the flanking characters were presented horizontally”-> “the irrelevant location leading to the Simon effect was the horizontal position of the stimulus display (flankers + target) and the flankers were al-so presented horizontally”. I suggest changing the other examples (e.g., the one from Rey-Mermet & Gade’s study) accordingly.”

RESPONSE: We changed the sentence according to Reviewer’s suggestion (see p. 11-12). Moreover, we modified the description of the following example as follows: “That is, the stimu-lus display (flankers + target) were presented in one of the four quadrants so that the irrelevant locations leading to the Simon congruency effect were the horizontal and vertical position of the stimulus display. Thus, there was a partial overlap with the flanker congruency variable as the flanking characters were presented horizontally.” (see p. 12).

18. “P. 12: The way in which the "learning" hypothesis is explained is not clear. The authors refer to it as a bias in the learning of stimulus-response pairings. Why a “bias”? Accord-ing to this account, the interaction results from the two compatibility effects (flanker and Simon) PLUS the effect of the different numbers of S-R pairings in the four congruency conditions (i.e., the four trial types), which, in turn, results in a different frequency of presentations of such pairings, The interaction thus would result from the summation of compatibility and episodic-memory effects. This should be better explained. The reader can understand the main point only after having read the whole paragraph (only on p. 13, line 7, the authors start explaining it) I also suggest eliminating the term bias, which makes the reader think of particular tendencies, inclinations, preferences. What about simply "episodic-memory account"?”

RESPONSE: We changed the explanation regarding the learning of stimulus-pairings as fol-lows: “According to this account, the interaction would be affected by the different numbers of stimulus-response pairings in the four trial types (i.e., flanker congruent – Simon congruent, flanker incongruent – Simon congruent, flanker congruent – Simon incongruent, flanker incon-gruent – Simon incongruent). That is, in case all four trial types are presented equally often but there are different numbers of stimulus-response pairings pro trial type, some stimulus-pairings are presented more frequently than others. In this case, the stimulus-pairings that are present-ed more frequently are more practiced, explaining better performance for these stimulus-pairings and thus the trial types to which these pairings belong. It is important to note that this difference in learning stimulus-pairings cannot affect the interaction when the stimulus set is small (i.e., when there are two stimulus values per congruency variable, such as the left and right location for the Simon congruency variable, and the colors blue and green for the flanker congruency variable). In this case, the number of stimulus-response pairings is similar across all trial types (see Fig. 2, top part), and thus the learning of these stimulus-response pairings should be similar in all trial types.” (see p. 12-13). Moreover, we removed the term “bias” from the whole manuscript. 

19. “P. 12: “a small stimulus set (i.e., two stimulus features” -> this use of the term "feature" might be confusing: usually one refers to a stimulus feature as a stimulus dimen-sion/attribute (e.g., one feature is the colour, another feature is the stimulus position). In this case, however, the authors refer to the value of the feature/dimension. Indeed, two stimulus colours are two values of the same stimulus feature/dimension (i.e., color). Ac-cordingly, when the authors speak about "four features" they are not actually referring to 4 different features of the stimulus (e.g., shape, colour, position, size) but to 4 different values (e.g., red, blue, green, yellow) of the same feature (color). I suggest replacing the term feature with other terms in the manuscript when appropriate, Note that the authors use "features" with the standard meaning on, e.g., p. 15”

RESPONSE: We thank the Reviewer for putting forward this inconsistency. In their text, the Reviewer used the term “value”. Therefore, we opted for this term, and we provided some ex-amples to clarify what we meant, that is: “when there are two stimulus values per congruency variable, such as the left and right location for the Simon congruency variable, and the colors blue and green for the flanker congruency variable” or “when the stimulus set consists of four stimulus values (i.e., four locations for the Simon congruency variable and four colors for the flanker congruency variable)” (see p. 13). 

20. “P. 12: I think that the reader might not understand what the authors mean when they claim that "Thus, the learning of these stimulus-response pairings should be similar across"”

RESPONSE: We changed the sentence as follows: “the learning of these stimulus-response pairings should be similar in all trial types.” (see p. 13). 

21. “P. 13: One reason for this finding might be that Rey-Mermet and Gade presented all pos-sible stimulus-response pairings in each trial type (please note that this information was not provided in Akçay and Hazeltine -> That is not clear at all at this point of the manu-script. The authors have not mentioned yet that when there are many (e.g., 4) possible colours/positions there are different frequencies of S-R pairings in the three different cat-egories of trials and they have not explained yet the consequences in terms of learn-ing/strength of episodic memories.”

RESPONSE: We agree with the Reviewer, and we moved this information towards the end of the section (see p. 14).

22. “P. 13: "(i.e., 4 congruent target/flanking colors in each response-congruent location", -> Given that there are four possible locations that are congruent with the correct response, this is confusing. Maybe better: “(i.e., for each of the four congruent pairings of stimulus and response locations, there was one congruent paring of target and flanker colors)” OR “(i.e., for each of the four congruent pairings of target and flanker colors, there was one congruent pairing of stimulus and response locations)”.”

RESPONSE: We used the first option in the manuscript (see p. 13)

23. “P. 17: ”the interaction should be observed when the facilitation effect (i.e., the differ-ence between congruent and neutral trials) was used as dependent measure. -> The de-pendent measures were always RTs and errors. Maybe the authors meant that ”the inter-action should be observed when the analyses focused on the facilitation effect”. The au-thors use "dependent measure" with the standard meaning in other parts of the manu-script.”

RESPONSE: We modified our text according to Reviewer’s recommendation (see p. 17-18, 29-36, and 49). 

24. “P. 48: "Together, the RTs findings of Experiments 1a and 1b emphasized the difficulty of finding appropriate Simon neutral trials on which performance is slower than congruent trials but faster than incongruent trials" I suggest the authors underline that such a prob-lem concerns only this particular task. Many previous studies found appropriate neutral Simon conditions in standard Simon tasks (e.g., Umiltà, Rubichi, Nicoletti, 1999: Wuhr & Ansorge, 2005; Aisenberg and Henik, 2011; Treccani, Cona, Milanese, Umltà, 2018)”

RESPONSE: We added the required information as follows: “This seems to be specific to the present paradigm in which both Simon and flanker congruency variables were combined as previous research showed appropriate neutral Simon conditions in standard Simon tasks (Ai-senberg & Henik, 2012; Treccani et al., 2018; Umilta et al., 1999; Wühr & Ansorge, 2005).” (see p. 49). 

25. “P. 52: "Moreover, across the experiments, the delta plots showed an inversed U-curved shape for the Simon congruency effect,...., with further RT slowing, the Simon congruen-cy effect diminished and sometimes reversed. These results are rather in line with Hom-mel than with Treccani and colleagues . A reason for this might be ..." In my view, the most plausible reason why Treccani et al. found that the Simon effect increases as the RT increases is that they used an accessory-stimulus Simon task. In this kind of tasks, the ir-relevant location that causes the effect is not the position of the target. The interfering in-formation is the position of another object (i.e., the accessory stimulus). This task is thought to be quite different from the standard one. In the accessory-stimulus version of the Simon task, just as happens in other variants of the Simon task, increasing Simon ef-fect functions are usually found. In this case, as in other paradigms in which the spatial meaning of the stimulus is not conveyed by the actual physical position of the target, the irrelevant spatial attribute causing the effect takes time to be coded- indeed, in the acces-sory stimulus task, participants are forced to analyse the target first, given that it is pre-sented at fixation, and then they move attention to the irrelevant accessory stimulus.”

RESPONSE: We added Reviewer’s suggestion in the Discussion (see p. 52).

---

## [Decision Letter · Decision Letter 2]

16 Feb 2021

PONE-D-20-15659R2

Multiplicative priming of the correct response can explain the interaction between Simon and flanker congruency

PLOS ONE

Dear Dr. Rey-Mermet,

Thank you for submitting your revised manuscript to PLOS ONE. R2 thank you for the detailed responses to his/her suggestions. He/she is suggesting other minor modifications that you might wish to add into the final version of your manuscript. I will read this final version without sending it to R2.

Please submit your final manuscript by Apr 02 2021 11:59PM. If you will need more time than this to complete this finale version, please reply to this message or contact the journal office at plosone@plos.org. Please include the following items when submitting your revised manuscript:

A rebuttal letter that responds (or not) to each point raised by R2. You should upload this letter as a separate file labeled 'Response to R2'.A marked-up copy of your manuscript that highlights changes made to the original version. You should upload this as a separate file labeled 'Revised Manuscript with Track Changes'.An unmarked version of your revised paper without tracked changes. You should upload this as a separate file labeled 'Manuscript'.

Thank you again for your patience and perseverance. I look forward to receiving your final version of your work.

Kind regards,

Ludovic Ferrand, Ph.D.

Academic Editor

PLOS ONE

Reviewers' comments:

Reviewer's Responses to Questions

**Comments to the Author**

1. If the authors have adequately addressed your comments raised in a previous round of review and you feel that this manuscript is now acceptable for publication, you may indicate that here to bypass the “Comments to the Author” section, enter your conflict of interest statement in the “Confidential to Editor” section, and submit your "Accept" recommendation.

Reviewer #2: (No Response)

2. Is the manuscript technically sound, and do the data support the conclusions?

Reviewer #2: Partly

3. Has the statistical analysis been performed appropriately and rigorously? 

Reviewer #2: Yes

4. Have the authors made all data underlying the findings in their manuscript fully available?

Reviewer #2: (No Response)

5. Is the manuscript presented in an intelligible fashion and written in standard English?

Reviewer #2: Yes

6. Review Comments to the Author

Reviewer #2: I thank the authors for their detailed replies to my comments. The following are additional comments that the authors may take into consideration:

“The account presented here is so far very similar to the temporal overlap account proposed by Hommel (1997), except to one key difference. Here, we do not assume a spontaneous decay of the representation features”

I think that the account presented here is very different from the account that Hommel proposed in 1997 for the interaction between S-R correspondence and S-S congruency. According to Hommel, S-S congruency (the compatibility between the target and flanker) affects the perceptual encoding of the stimulus, stimulus identification, while S-R correspondence (the correspondence between stimulus and response positions) affects response selection. In S-S incongruent trials, stimulus identification would be delayed and thus, when response section starts, the irrelevant spatial code would not be active anymore (either because of the decay of the spatial code or because of active inhibition of this spatial code - that does not matter in this context). Accordingly, this code would not have any effect on response selection (->under-additive interaction).

In contrast, the authors here propose that both S-S congruency and S-R correspondence affect response selection: Both irrelevant features prime the congruent/correspondent response at the response selections stage. If there is an overlap in time between the effects of these priming processes (if the response primed by a feature is still activated when the second priming process occurs) we observe the interaction. Thus, in order for the interaction to be observed, the overlap does not need to occur between the processing of the two features, but between the effect of the processing of the two features: the effect of the processing of one feature (i.e, the response priming mediated by one feature) must overlap in time with the effect of the processing of the other feature (i.e. the response priming mediated by the other feature).

In this respect, in my view, the two accounts that the authors propose in the ‘Discussion’ section (pp. 53-57) are not that far from each other.

In the cover letter the authors claim “ the EEG findings are not directly compatible with an account assuming an overlap in time. The EEG results clearly showed separate ERPs associated to the processing Simon and flanker irrelevant features.”

However, the lack of an overlap in time between two ERP components reflecting the processing of two stimulus features is totally consistent with an overlap in time of the effects of the processing of these two features: one feature can be processed and its effect (i.e., response activation/priming produced by the processing of this feature) can last over time.

Finally in the Discussion section the authors claim that: “if one considers that the interaction between Simon and flanker congruency does not emerge during conflict processing, but appears on the level of motor responses.”

This is confusing: According to main models of response selection, in the case of tasks such as Simon and flanker tasks, conflict processing DOES occur at a motor-related stage, namely, response selection (thus the level of motor responses coincides with the stage in which response conflict emerges). In this kind of tasks, the possible responses are only two or four (the same 2 or 4 responses are repeated in the different trials) and they are typically button presses, thus critical variables are not likely to affect other (late) motor-related stages, that is, motor execution (such as happens instead with unpredictable, complex movements). Response selection coincides with the activation /inhibition of the motor codes of these responses.

7. PLOS authors have the option to publish the peer review history of their article (what does this mean?). If published, this will include your full peer review and any attached files.

Reviewer #2: No

---

## [Author Response · Author response to Decision Letter 2]

19 Feb 2021

Reviewer A

A1.1 ““The account presented here is so far very similar to the temporal overlap account pro-posed by Hommel (1997), except to one key difference. Here, we do not assume a spontaneous decay of the representation features”

I think that the account presented here is very different from the account that Hommel pro-posed in 1997 for the interaction between S-R correspondence and S-S congruency. According to Hommel, S-S congruency (the compatibility between the target and flanker) affects the per-ceptual encoding of the stimulus, stimulus identification, while S-R correspondence (the cor-respondence between stimulus and response positions) affects response selection. In S-S incongruent trials, stimulus identification would be delayed and thus, when response section starts, the irrelevant spatial code would not be active anymore (either because of the decay of the spatial code or because of active inhibition of this spatial code - that does not matter in this context). Accordingly, this code would not have any effect on response selection (->under-additive interaction).

In contrast, the authors here propose that both S-S congruency and S-R correspondence affect response selection: Both irrelevant features prime the congruent/correspondent response at the response selections stage. If there is an overlap in time between the effects of these prim-ing processes (if the response primed by a feature is still activated when the second priming process occurs) we observe the interaction. Thus, in order for the interaction to be observed, the overlap does not need to occur between the processing of the two features, but between the effect of the processing of the two features: the effect of the processing of one feature (i.e., the response priming mediated by one feature) must overlap in time with the effect of the processing of the other feature (i.e. the response priming mediated by the other feature).

In this respect, in my view, the two accounts that the authors propose in the ‘Discussion’ sec-tion (pp. 53-57) are not that far from each other.”

RESPONSE: Following Reviewer’s comment, we re-wrote the sentences in order to make clear the similarity (i.e., the assumption of a temporal overlap) and the key difference (i.e., no assumption about which process is responsible for this overlap) between Hommel’s (1997) account and the framework presented in the manuscript. We did it as follows (see p. 54): 

“Similar to Hommel [6], a temporal overlap is assumed in this account. However, there is at least one key difference. No spontaneous decay of the representation features is assumed. In this view, whereas the spontaneous decay of the representation features is still an option to explain the interaction between Simon and flanker congruency, the inhibition of irrelevant fea-tures (as derived from the conflict monitoring account) or the multiplicative priming of the cor-rect response are also valid options.” 

Moreover, the Reviewer suggested similarities between the temporal overlap account present-ed on p.53-54 and the account assuming the pre-activation of the correct response. These are now reported as follows (see p. 56): “It should be noted that according to this explanation [that is, the pre-activation of motor responses by relevant and irrelevant features], there is still an overlap in time. However, this temporal overlap is not between the processing of the relevant and irrelevant features, such as described above or proposed by Hommel [6]. Rather, it is be-tween the effects of processing the irrelevant features. That is, the interaction between Simon and flanker congruency occurs only if the effect of processing one irrelevant feature (i.e., the response priming mediated by one irrelevant feature) overlaps in time with the effect of pro-cessing the other irrelevant feature (i.e. the response priming mediated by the other irrelevant feature).” 

A2 “In the cover letter the authors claim “ the EEG findings are not directly compatible with an account assuming an overlap in time. The EEG results clearly showed separate ERPs associ-ated to the processing Simon and flanker irrelevant features.”

However, the lack of an overlap in time between two ERP components reflecting the pro-cessing of two stimulus features is totally consistent with an overlap in time of the effects of the processing of these two features: one feature can be processed and its effect (i.e., re-sponse activation/priming produced by the processing of this feature) can last over time.”

RESPONSE: Following Reviewer’s suggestion, we modified the manuscript as follows (see p. 56): “This assumption [that is, the effect of processing one irrelevant feature overlaps in time with the effect of processing the other irrelevant feature] is also compatible with the EEG find-ings showing separate ERPs associated with the processing of the irrelevant features for both Simon and flanker congruency variables [5, 23,24]. The reason is that the lack of a temporal overlap between the two ERP components reflecting the processing of relevant and irrelevant features is totally consistent with a temporal overlap in the effects of processing these fea-tures: One feature can be processed and its effect – that is, the response priming mediated by the processing of this feature – can last over time.”

A3 “Finally in the Discussion section the authors claim that: “if one considers that the interac-tion between Simon and flanker congruency does not emerge during conflict processing, but appears on the level of motor responses.”

This is confusing: According to main models of response selection, in the case of tasks such as Simon and flanker tasks, conflict processing DOES occur at a motor-related stage, namely, response selection (thus the level of motor responses coincides with the stage in which re-sponse conflict emerges). In this kind of tasks, the possible responses are only two or four (the same 2 or 4 responses are repeated in the different trials) and they are typically button presses, thus critical variables are not likely to affect other (late) motor-related stages, that is, motor execution (such as happens instead with unpredictable, complex movements). Re-sponse selection coincides with the activation /inhibition of the motor codes of these re-sponses.”

RESPONSE: Because of Reviewer’s comment, we removed the sentence from the revised manuscript (see p. 55).

---

## [Editor Report · Decision Letter 3]

22 Feb 2021

Multiplicative priming of the correct response can explain the interaction between Simon and flanker congruency

PONE-D-20-15659R3

Dear Dr. Rey-Mermet,

I am pleased to inform you that your manuscript has been judged scientifically suitable for publication and will be formally accepted for publication once it meets all outstanding technical requirements.

Kind regards,

Ludovic Ferrand, Ph.D.

Academic Editor

PLOS ONE

---

## [Editor Report · Acceptance letter]

26 Feb 2021

PONE-D-20-15659R3 

Multiplicative priming of the correct response can explain the interaction between Simon and flanker congruency 

Dear Dr. Rey-Mermet:

I'm pleased to inform you that your manuscript has been deemed suitable for publication in PLOS ONE. Congratulations! Your manuscript is now with our production department. 

Kind regards, 

on behalf of

Dr. Ludovic Ferrand 

Academic Editor

PLOS ONE